# Basal IFN-λ2/3 expression mediates tight junction formation in human epithelial cells

Yagmur Keser [1,4], Camila Metz-Zumaran [1,2,4], Zina M Uckeley [1], Dorothee Reuss[2,3], Patricio Doldan[2], James M Ramsden[1], Megan L Stanifer [1✉] & Steeve Boulant [1,2✉]

## Abstract

Type-III interferons (or IFNλs) play important roles in antiviral defense and intestinal epithelial barrier integrity. While interferon expression has been primarily studied in response to pathogens, basal interferon expression also occurs in pathogen-free environments. However, the mechanisms regulating basal IFN-λ expression and their functions have not yet been elucidated. Here, we show that basal IFN-λ2/3 expression is linked to the development of an intact cellular epithelium characterized by formation of tight junctions and establishment of barrier function. Our findings indicate that basal IFN-λ2/3 expression depends on cGAS-STING-mediated mitochondrial DNA detection, while it is inhibited by the Hippo mechanotransduction pathway at low cellular densities. Cells lacking basal IFN-λ2/3 expression fail to develop proper tight junctions and establish normal barrier function. Mechanistically, IFN-λ2/3 suppresses Claudin-2 expression, thereby promoting barrier formation as cells become confluent. These results demonstrate a previously unknown function of basal IFNλ expression in regulating epithelial cell junction formation and highlight their importance not only during pathogen challenges but also in maintaining epithelial cell function under steady-state conditions.

**Keywords** Epithelial Barrier Function; Basal IFNλ Signaling; Tight Junctions; Hippo Signaling Pathway; cGAS-STING Pathway
**Subject Categories** Cell Adhesion, Polarity & Cytoskeleton; Immunology; Signal Transduction

## Introduction

Mucosal surfaces, encompassing the airways, oral cavity, digestive tract, genitourinary tract, and skin, are covered by polarized epithelial cells which establish a physical, chemical, and immuno-logical semipermeable barrier between external environments and the internal milieu. The integrity of this epithelial barrier is maintained by tight junction proteins such as occludin, claudins, and zonula occludens (Buckley and Turner, 2018; Zuo et al, 2020). These junctions rely on transmembrane proteins connected to cytoplasmic adaptors and the actin cytoskeleton to hold adjacent cells together, thereby creating intercellular seals. Consequently, tight junctions influence various downstream signaling cascades such as regulating cell proliferation and differentiation, organ growth, tissue regeneration or wound healing (Díaz-Coránguez et al, 2019; Shi et al, 2018; Zihni et al, 2014). Barrier integrity must be meticulously regulated to accommodate the commensal micro-biota while remaining vigilant and responsive to invading pathogens. Dysregulation and deficiencies of this barrier have profound effects on human health and disease, as they are associated with the onset of inflammation and diseases (Chelakkot et al, 2018; Guttman and Finlay, 2009; Paradis et al, 2021; Ulluwishewa et al, 2011).

Interferons (IFNs) are key cytokines produced and released by the host in response to pathogen challenges. They ultimately enhance the ability of the immune system to recognize, respond, and fight infections. Type I and III IFNs are essential to protect against viral infection (Hoffmann et al, 2015; Kawai and Akira, 2006; Koyama et al, 2008). In humans, there are 17 different subtypes of type I IFN (13 IFNα, IFNβ, IFNε, IFNκ, and IFNω) which bind to and signal through the type I IFN receptor which is composed of two subunits (IFNAR1 and IFNAR2). The type I IFN receptor is expressed by all nucleated cells (Gibbert et al, 2013; Kotenko and Durbin, 2017; LaFleur et al, 2001; Langer et al, 2004; Pestka et al, 2004). The type III IFN family (i.e., IFNλs) is the latest discovered IFN family and is composed of four members (IFNλ1-4). IFNλ1 (IL-29) is uniquely expressed in primates and is a pseudogene in mice, while IFNλ2 (IL-28a) and IFNλ3 (IL-28b) are expressed across all mammals and share a close similarity, with 96% amino acid sequence identity. IFNλ4 expression varies across human populations and is determined by a genetic polymorphism; while some individuals carry a null allele, a substantial proportion—especially in certain ethnic groups—express functional IFNλ4.(Kotenko et al, 2003; Prokunina-Olsson et al, 2013; Sheppard et al, 2003). IFNλs signal through the type III IFN heterodimeric receptor complex composed of the IL10R and IFNLR subunit which is mostly expressed in epithelial cells and in a subset of immune cells (Mordstein et al, 2010; Sommereyns et al, 2008). Due to the restriction of IFNLR expression, IFNλs have been described as key cytokines to protect

[1]Department of Molecular Genetics and Microbiology, College of Medicine, University of Florida, Gainesville, FL, USA. [2]Department of Infectious Disease, Virology, University Hospital Heidelberg, Heidelberg, Germany. [3]Present address: Department of Infectious Diseases, Imperial College London, London, UK. [4]These authors contributed equally: Yagmur Keser, Camila Metz-Zumaran. ✉E-mail: m.stanifer@ufl.edu; s.boulant@ufl.edu

epithelial and mucosal surfaces against viral infection (Mordstein et al, 2010; Sommereyns et al, 2008).

IFNs are mostly produced and secreted by cells in response to viral infection. Upon sensing of viral infection by cellular pattern recognition receptors (PRRs) such as RIG-like receptors (RLRs), Toll-like receptors (TLRs), and cyclic GMP-AMP synthase (cGAS), signal transduction cascades are induced to ultimately activate TANK-binding kinase-1 (TBK1) which in turn will activate, through phosphorylation, the interferon regulatory factor-3 (IRF3) (Dalskov et al, 2023; Liu et al, 2015; Odendall and Kagan, 2015). Phosphorylation of IRF3 results in its dimerization and translocation into the nucleus where it acts as a transcription factor driving the expression of both type I and type III IFNs. Both IFNs are then secreted from the cell and act on their corresponding receptors to activate the Janus kinase (JAK)-Signal Transducer and Activator of Transcription Proteins (STAT) signaling pathway. Following activation of STAT1 and STAT2 via phosphorylation, these proteins associate with interferon regulatory factor 9 (IRF9) to form the interferon stimulated gene factor 3 (ISGF3) complex. ISGF3 acts as a transcription factor regulating the expression of hundreds of interferon-stimulated genes (ISGs) that act as the antiviral effectors in cells (Schindler et al, 2007; Stanifer et al, 2019).

Besides their antiviral/pathogen actions, IFNs have been described to also regulate several cellular functions. Type I IFNs are known to regulate cell proliferation (Katlinskaya et al, 2016), angiogenesis (Enomoto et al, 2017; Zheng et al, 2011), and cellular metabolism (Liu et al, 2013; Wu et al, 2016) (Fritsch and Weichhart, 2016). Thus, induction or activation of IFNs not only plays a critical role in controlling pathogen challenges but also plays a pivotal role in cell biology and in the development of disease (interferonopathies (Crow and Stetson, 2022), cancer (Crow and Stetson, 2022), autoinflammation (Raftopoulou et al, 2022), and aging (Cao, 2022)). For IFNλs much less is known about their non-antipathogen functions. Besides their antiviral properties, IFNλ signaling has been shown to regulate the barrier function of epithelial cells during pathogenic challenges. Some studies suggest that IFNλs enhance epithelial defense and barrier function against parasites, viruses, and bacterial infections (Ferguson et al, 2019; Lazear et al, 2015; Odendall et al, 2017). Conversely, other reports indicate that IFNλs may disrupt epithelial integrity during viral and bacterial infections (Broggi et al, 2020; Major et al, 2020). These observations underscore the significant interplay between IFNλs and the barrier function of epithelial cells during pathogen challenges. However, the underlying mechanisms by which IFNλs regulate tight junction formation and mucosa healing remains mostly unknown.

Although the expression of IFNs is known to be tightly regulated in response to pathogenic triggers, it has become clear that type I IFNs are also produced constitutively at low but physiologically relevant levels in non-infected sterile cells (Gough et al, 2012; Tovey et al, 1987). These low levels of type I IFNs present in cells at steady state are referred to as basal interferons. Basal expression of type I IFN (IFNβ) is known to have multiple biological roles such as maintaining the hematopoietic stem cell niche (Essers et al, 2009) and regulating immune cell function (Ganal et al, 2012) and bone remodeling (Deng et al, 2020). Perhaps the best characterized function of basal type I IFN is its role in regulating the tonicity of the IFN response itself. Constitutively expressed type I IFNs maintain homeostatic levels of key JAK/STAT signaling pathway

components, thereby priming cells for a rapid and robust response to subsequent pathogenic challenges (Gough et al, 2012). Consistent with this model, cells lacking IFNβ signaling show reduced levels of STAT1, STAT2, IRF1, and IRF7 compared to wild-type cells in a sterile environment (Fleetwood et al, 2009; Gough et al, 2010; Thomas et al, 2006). The upstream factors and pathways which are responsible for inducing basal IFN transcription remain poorly characterized. It was recently shown that the cGAS-STING pathway may be responsible for the basal type I IFN production (Thomas et al, 2006; Tu et al, 2022). Satellite DNA, mitochondria DNA, and retro-element DNA present in the cytosol of non-infected cells have been proposed to serve as potential immunogenic triggers activating the cGAS-STING ultimately leading to the production of basal type I IFN at steady state (Tu et al, 2022). For IFNλs, much less is known. We have previously reported that IFNλs are also expressed at basal levels in epithelial cells. However, which signal transduction pathways and pattern recognition receptors are involved in regulating basal IFNλ expression remains to be determined. Similarly, the function of basal IFNλ expression in epithelial cells remains unknown. Given the function of IFNλs in regulating the barrier function of epithelial cells during pathogen challenges, basal IFNλs might directly participate in regulating an important function of epithelial cells namely the establishment of a tight epithelium barrier.

In this study, we uncovered novel regulatory mechanisms controlling basal IFNλ expression in human epithelial cells and demonstrate the essential role of basal IFNλ in promoting epithelial barrier function by modulating tight junction formation. We identify a novel pathway in which epithelial cell density and Hippo signaling modulate cGAS-STING-dependent IFN-λ2/3 expression, which in turn suppresses claudin-2 to promote tight junction formation. Our findings highlight a critical interplay between mechanotransduction, basal IFN signaling, and barrier integrity, expanding the known functions of IFNλs beyond their canonical antiviral roles.

# Results

## Production of basal IFNλs in human epithelial cells is dependent on cell confluency

During cell polarization and formation of tight junctions, the transcriptional profile of epithelial cells is significantly modified (Paradis et al, 2021). As we have previously shown that intestinal epithelial cells express basal levels of IFNλs (Karlowitz et al, 2022), we designed experiments to address whether basal IFNλ expression is influenced by cell polarization and tight junction formation. We employed the colon carcinoma cell line T84 cells as a model for intestinal epithelial cells (Karlowitz et al, 2022). To confirm that T84 cells polarize and establish tight junctions, we employed standard methods that evaluate the capacity of epithelial cells to establish a barrier function that restricts the diffusion of molecules across the epithelium. We monitored the transepithelial electrical resistance (TEER), measured the restriction of FITC-dextran diffusion from the apical to the basolateral compartment, and followed the formation of tight junctions over time by immuno-fluorescence staining of the tight junction protein ZO-1 (Stanifer et al, 2016). Seven to eight days post seeding, T84 cells form a tight

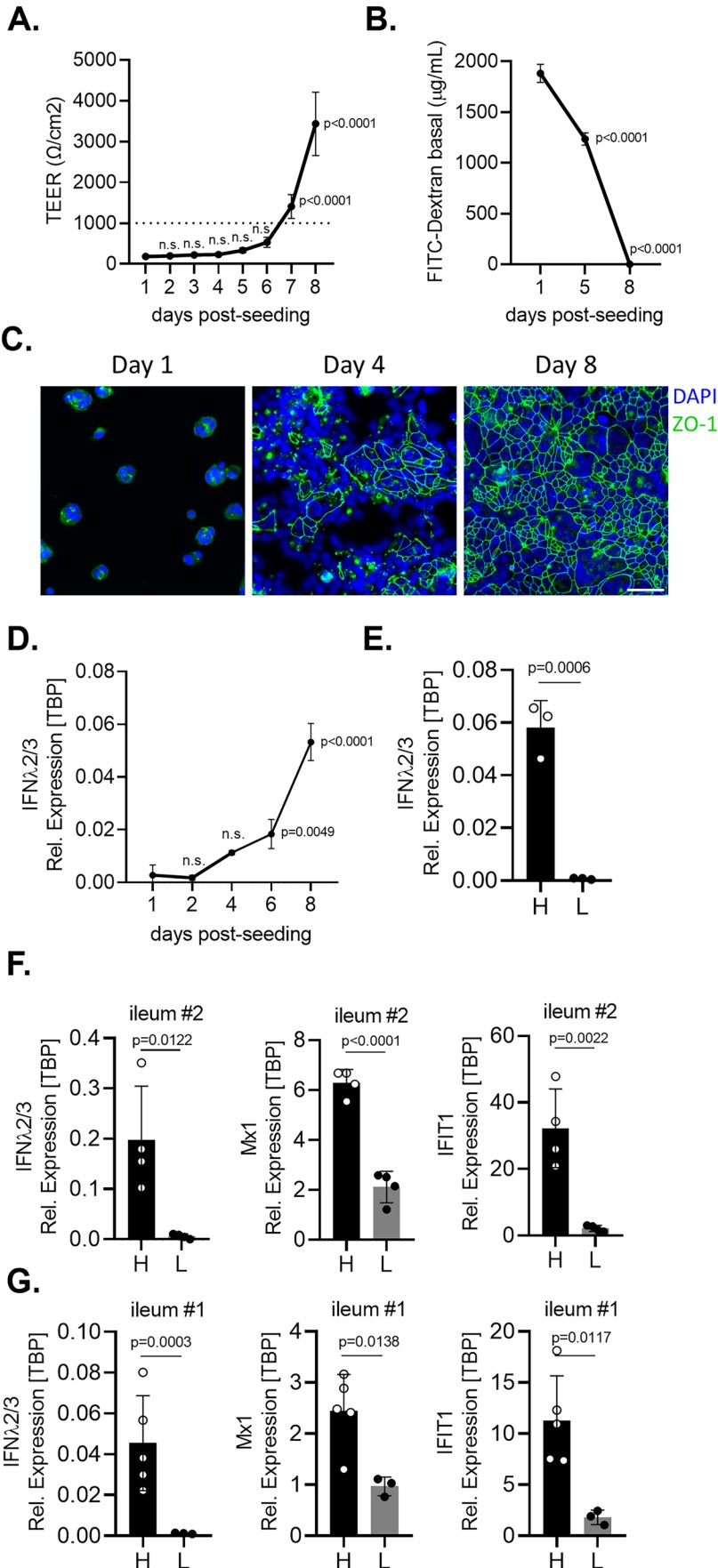

**Figure 1.   Basal IFN-λ2/3 expression is regulated in a cell density-dependent manner.**

(A–D) T84 cells were seeded on transwell inserts to allow for the formation of a polarized monolayer of cells. Formation of tight junctions and barrier function were monitored by (A) measuring the transepithelial electrical resistance (TEER) over 8 days (values > 1000 Ω/cm² (dotted line) indicates formation of efficient barrier function), (B) measuring the permeability of the T84 cell monolayer to FITC-Dextran at 1, 5 and 8 days post-seeding and (C) immunostaining of the tight junctions using an anti-ZO-1 antibody (green). Nuclei were stained with DAPI (blue). Scale bar = 50 μm. (D) T84 cells were harvested at indicated days and basal expression of IFN-λ2/3 was monitored by qRT-PCR. (E) Basal expression of IFN-λ2/3 was addressed by qRT-PCR in T84 cells seeded at high and low cellular densities. (F, G) Ileum-derived organoids from two donors (#1 and #2) were seeded at high and low cellular density, qRT-PCR was performed to monitor the basal expression of IFN-λ2/3 and of the ISGs Mx1 and IFIT1. The relative expression of IFN-λ2/3 and of the ISGs was normalized to TBP. $n \geq 3$ biological replicates. Statistical analysis was performed using (A, B, D) ordinary one-way ANOVA using the cells at 1 day post-seeding as a reference, or (E–G) unpaired $t$ test between high and low density. n.s. indicates non-significant results ($P > 0.05$). Exact $P$ values are shown on the plots when significant; otherwise, results are not significant. Error bars represent standard deviation with the mean as the center. Source data are available online for this figure.

monolayer characterized by the establishment of a transepithelial electrical resistance characteristic of polarized cells (TEER > 1000 Ω/cm²) (Stanifer et al, 2016), the ability to block dextran diffusion, and the development of a continuous tight junction belt between epithelial cells (Fig. 1A–C). To determine if expression of basal IFNλ was impacted by cell polarization and tight junction formation, we measured the expression of IFN-λ2/3 overtime as T84 cells polarize. We found that T84 cells expressed basal IFN-λ2/3, and that its expression was significantly increased overtime as cells polarized and formed tight junctions (Fig. 1D). To address whether this increased expression of basal IFN-λ2/3 correlated with an increased expression of downstream interferon stimulated genes (ISGs), we analyzed ISG expression as T84 cells polarized. Results show that similar to IFN-λ2/3, expression of representative ISGs (Mx1, IFIT1, ISG15, and Viperin) increased overtime (Fig. EV1A) as T84 cells polarize and develop tight junctions (Fig. 1A–C). To confirm these findings, we employed a reporter T84 cell line expressing the mCherry fluorescent protein under the control of the interferon-stimulated gene Mx1 promoter region (T84 pMx1-mCherry). This reporter cell line becomes fluorescent when exposed to IFN treatment (Doldan et al, 2022). Results showed that, overtime, as T84 cells become confluent and polarized, the expression of the mCherry fluorescent protein reporter gene increases (Fig. EV1B,C). Quantification revealed a positive correlation between cell confluency and the expression level of the mCherry fluorescent protein (Fig. EV1D). To explore how cell confluency influences the expression of basal IFNλ by human intestinal epithelial cells, we developed a standardized density-based seeding approach. In this system, cells were either seeded at high cell density (H) to form a confluent monolayer or seeded at low cell density (L) to obtain isolated individual cells or small cellular clusters. Comparative analysis of the relative expression level of IFN-λ2/3 in cells seeded at high versus low cellular density revealed that cells seeded at high cell density expressed significantly more basal IFN-λ2/3 than cells at low density (Fig. 1E). Comparative analysis of the relative expression levels of representative ISGs in T84 cells seeded at high *versus* low cellular density revealed that cells seeded at high cell density express significantly more ISGs (Mx1, IFIT1, ISG15, and Viperin) compared to cells seeded at low cellular density (Fig. EV1E). Similar results were observed when examining the expression of the interferon-stimulated gene ISG15 at the protein level (Fig. EV1F). To confirm these findings in primary non-transformed human intestinal epithelial cells, we employed human ileum-derived intestinal organoids (i.e., enteroids) from two different donors, ileum #1 and #2. Organoids were seeded at high and low density

and basal expression of IFN-λ2/3 and ISGs Mx1 and IFIT1 was assessed by qRT-PCR. Similar to T84 cells, cellular confluency positively correlated with the expression of basal IFN-λ2/3 and of the ISGs Mx1 and IFIT1 in ileum-derived organoids (Fig. 1F,G). To address whether this density-dependent ISG expression is conserved among other epithelial cell types, we examined the expression of the IFNλs and the interferon-stimulated genes Mx1, IFIT1, and Viperin in Calu-3 cells which is an in vitro model of human airway epithelial cells (hAECs). Results showed a similar increase in basal IFN-λ2/3 and ISG expression at high cellular density compared to low cellular density (Fig. EV1G). Together, our results show that as epithelial cells polarize and become confluent, the basal expression levels of IFN-λ2/3 and of ISGs are increased. This suggests that the basal expression of IFN-λ2/3 is regulated by cell confluency.

## Density-dependent expression of ISGs is driven by basal IFN-λ2/3 signaling

ISG expression usually depends on the activation of the JAK/STAT signaling pathway (Schindler et al, 2007). To address whether the increased expression of ISGs that we have observed in confluent cells correlates with the activation of the JAK/STAT signaling pathway, T84 cells were seeded at high *vs.* low cell densities and the activation of the JAK/STAT signaling pathway was assessed by measuring the phosphorylation status of STAT1. We found a significant increase in phosphorylated STAT1 levels in cells seeded at high cellular density compared to cells seeded at low cellular density (Fig. 2A). These findings indicate that the JAK/STAT signaling pathway activation is linked to cell confluency. The JAK/STAT pathway can be activated by various cytokines, including interferons and interleukins. To confirm the unique role of IFN-λ2/3 signaling and exclude the involvement of type I IFNs in density-dependent basal immune response, we employed our previously characterized type I IFN receptor knockout cells (IFNAR KO), type III IFN receptor KO cells (IFNLR KO), and cells depleted of both type I and type III IFN receptors (IFNR dKO) (Pervolaraki et al, 2017). Cells were seeded at high vs. low cellular density and the basal immune response of cells was monitored by qRT-PCR. Basal ISG expression levels (Mx1, IFIT1, Viperin) were similar between WT and IFNAR KO cells and importantly, the basal ISG expression was significantly increased at high cell density compared to low cell density (Fig. 2B). This strongly suggests that type I IFNs are not responsible for the increase in basal ISG expression observed in cells grown at high confluency. To confirm that type I IFNs were not modulated by density, IFNβ expression was also

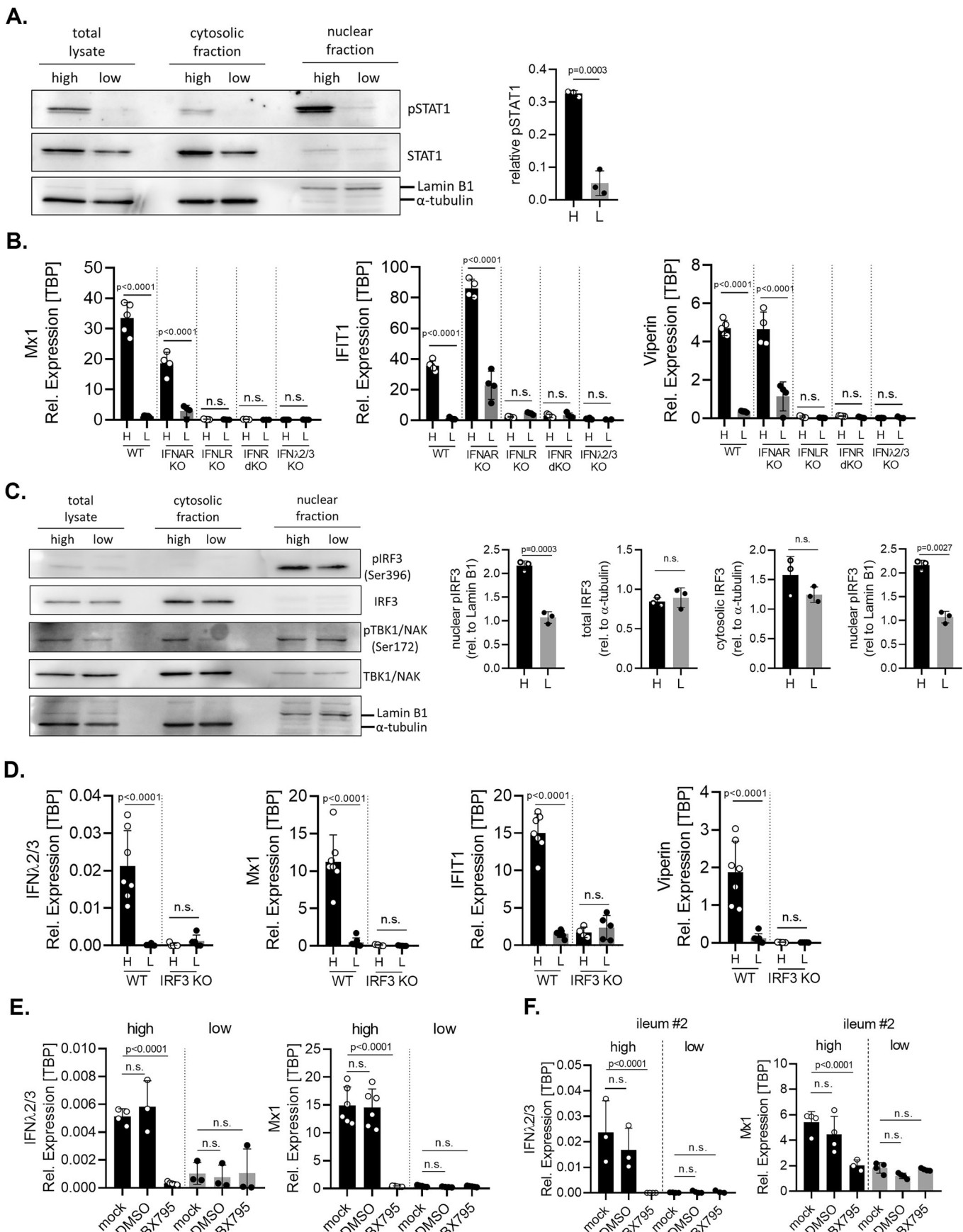

Figure 2.  Cell density-dependent basal intrinsic immune activation is driven by basal IFN-λ2/3 expression that is regulated through the TBK1-IRF3 axis.

(A) Left panel: T84 cells seeded at high and low cellular density were harvested and nuclear/cytosolic fractionation was performed to assess the relative abundance of pSTAT1 and total STAT1 using western blot analysis. α-tubulin and Lamin B1 were used as loading controls for the cytosolic and nuclear fraction, respectively. Right panel: The amount of pSTAT1 was quantified relative to the amount of Lamin B1 in the nuclear fraction. (B) T84 WT and T84 knock-out (KO) cells were seeded at high and low cellular density and the expression of the ISGs Mx1, IFIT1 and Viperin was addressed using qRT-PCR. (C) Left panel: T84 cells seeded at high and low density were harvested and nuclear/cytosolic fractionation was performed to assess the relative abundance of p-IRF3, total IRF3, p-TBK/NAK and total TBK1/NAK using Western Blot analysis. α-tubulin and Lamin B1 were used as loading controls for the cytosolic and nuclear fraction, respectively. Right panel: The amounts of nuclear p-IRF3, total and cytosolic IRF3, and cytosolic p-TBK1 and were quantified relative to α-tubulin from the cytosolic and total fraction, and Lamin B1 from the nuclear fraction. (D) T84 WT and T84 IRF3 knock-out (IRF3 KO) cells were seeded at high and low cellular density and the expression of IFN-λ2/3 and of the ISGs Mx1, IFIT1 and Viperin was addressed using qRT-PCR. (E) T84 WT cells and (F) ileum-derived organoids at high and low cell density were treated with either culture media (mock), DMSO (1 μM), or treated with 1 μM of the TBK1 inhibitor BX795. The relative expression of IFN-λ2/3 and Mx1 was addressed using qRT-PCR. (B, D–F) The relative expression was normalized to TBP. $n \geq 3$ biological replicates. Statistical analysis was performed using (A, C) unpaired $t$ test or (B, D-F) two-way ANOVA, between high and low density. n.s. indicates non-significant results ($P > 0.05$). Exact $P$ values are shown on the plots when significant; otherwise, results are not significant. Error bars represent standard deviation with the mean as the center. Source data are available online for this figure.

evaluated at high and low cellular density in WT and in IFN receptor KO cells lines. Results showed that there was no difference in IFNβ expression at high and low cellular density in all cell lines further suggesting that its expression is not modulated by cell confluency (Fig. EV2A). Interestingly, ISG expression was abolished in the absence of IFNλ signaling (IFNLR KO and IFNR dKO cells) (Fig. 2B) and no difference was found in ISG expression between high and low cellular densities (Fig. 2B). These findings strongly suggest that IFNλs are critical for the increase in basal immune response observed in intestinal epithelial cells when cells reach confluency. The critical role of IFN-λ2/3 in driving basal ISG expression was also confirmed by generating IFN-λ2/3 KO T84 cells (Appendix Fig. S1). In line with the unique regulation of basal IFN-λ2/3 expression by cellular density (Fig. 1D,E), the density-dependent basal ISG expression was abolished in cells lacking basal IFN-λ2/3 (Fig. 2B). Importantly, IFN-λ2/3 expression was not impacted by IFN signaling as IFN receptor KO cells (IFNAR KO, IFNLR KO and IFN dKO) also showed an increased expression of IFN-λ2/3 in high cell density as compared to low cell density (Fig. EV2B). To address whether other anti- and pro-inflammatory cytokines were also influenced by cell confluency, T84 cells were seeded at high and low cell densities and evaluated for the upregulation of IL-1β, IL-6, IL-18, IL-8 by qRT-PCR. Results show that IL-1β, IL-6, IL-18 were equally expressed at high and low densities while IL-8 expression was diminished at high cell density (Fig. EV2C–F). To determine if the differences in immune signaling observed between high and low cell confluency were due to differences in IFN receptor abundance, we measured IFN receptor expression by qRT-PCR in T84 WT cells grown at high and low cellular density. No difference in type III IFN receptor (IFNLR) expression was observed between high and low cellular density (Fig. EV2G). Notably, IFNAR1 expression was slightly higher in high-density cells compared to low-density cells (Fig. EV2G). However, as shown in IFNAR KO cells (Fig. 2B), type I IFNs are not responsible for basal immune signaling at high cell density. Together, our findings suggest that as epithelial cells reach cellular confluency, basal IFN-λ2/3 signaling is significantly induced, leading to downstream ISG expression via the JAK/STAT signaling pathway.

To identify the factors regulating IFN-λ2/3 induction at high cellular confluency, we dissected the signaling pathway leading to IFN production. We observed that the levels of the phosphorylated active form of IRF3 (p-IRF3) were higher in the nuclear fraction of

confluent cells compared to low confluency (sparse) cells (Fig. 2C, left and right panels). On the contrary, total and cytosolic IRF3 protein levels remain unchanged between high and low cellular density (Fig. 2C, left and right panels). To evaluate the role of IRF3 in regulating basal IFN-λ2/3 expression, we employed our previously characterized T84 IRF3 KO cell line (Triana et al, 2021). Compared to WT cells, in T84 cells lacking IRF3 there was very little to no basal IFN-λ2/3 and ISG expression and there was no difference observed between high and low cellular density (Fig. 2D). This strongly suggests that increased activation of IRF3 under high cellular confluency is responsible for the increased expression of basal IFN-λ2/3. TBK1 is a central kinase that directly phosphorylates IRF3. Although total TBK1 protein levels were similar in cells at high and low density (Fig. 2C, left panel), we observed greater amount of the active phosphorylated form of TBK1 in the cytosolic fraction of cells grown at high density compared to low density (Fig. 2C, left and right panels). Importantly, inhibition of TBK1 using the specific inhibitor BX795 (Clark et al, 2009) significantly reduced basal IFN-λ2/3 and basal ISG Mx1 expression in T84 cells and ileum-derived organoids at high cell density, but had no effect at low cell density (Fig. 2E,F). Altogether these results suggest that in confluent intestinal epithelial cells, TBK1 is activated which in turn phosphorylates IRF3 leading to the induction of basal IFN-λ2/3 expression. IFN-λ2/3 then signals through the type III IFN receptor, activating the JAK/STAT signaling cascade to induce the expression of ISGs.

## Basal expression of IFN-λ2/3 is mediated by the cGAS-STING signaling pathway

Induction of IFN expression typically occurs in response to the recognition of pathogenic elements by pattern recognition receptors (PRRs) (Dalskov et al, 2023; Liu et al, 2015; Odendall and Kagan, 2015). However, when investigating basal immune response, no exogenous pathogens are present in cells. Interestingly, it has been previously reported that the pattern recognition receptor cGAS also recognizes cellular endogenous DNA from mitochondrial or nuclear origins (Ma et al, 2020). Upon sensing of cellular-derived DNA located in the cytosol by cGAS, the second messenger 2′3′-cGAMP is produced leading to the activation of STING which in turn activates TBK1 (Motwani et al, 2019). TBK1 subsequently phosphorylates IRF3 leading to type I IFN

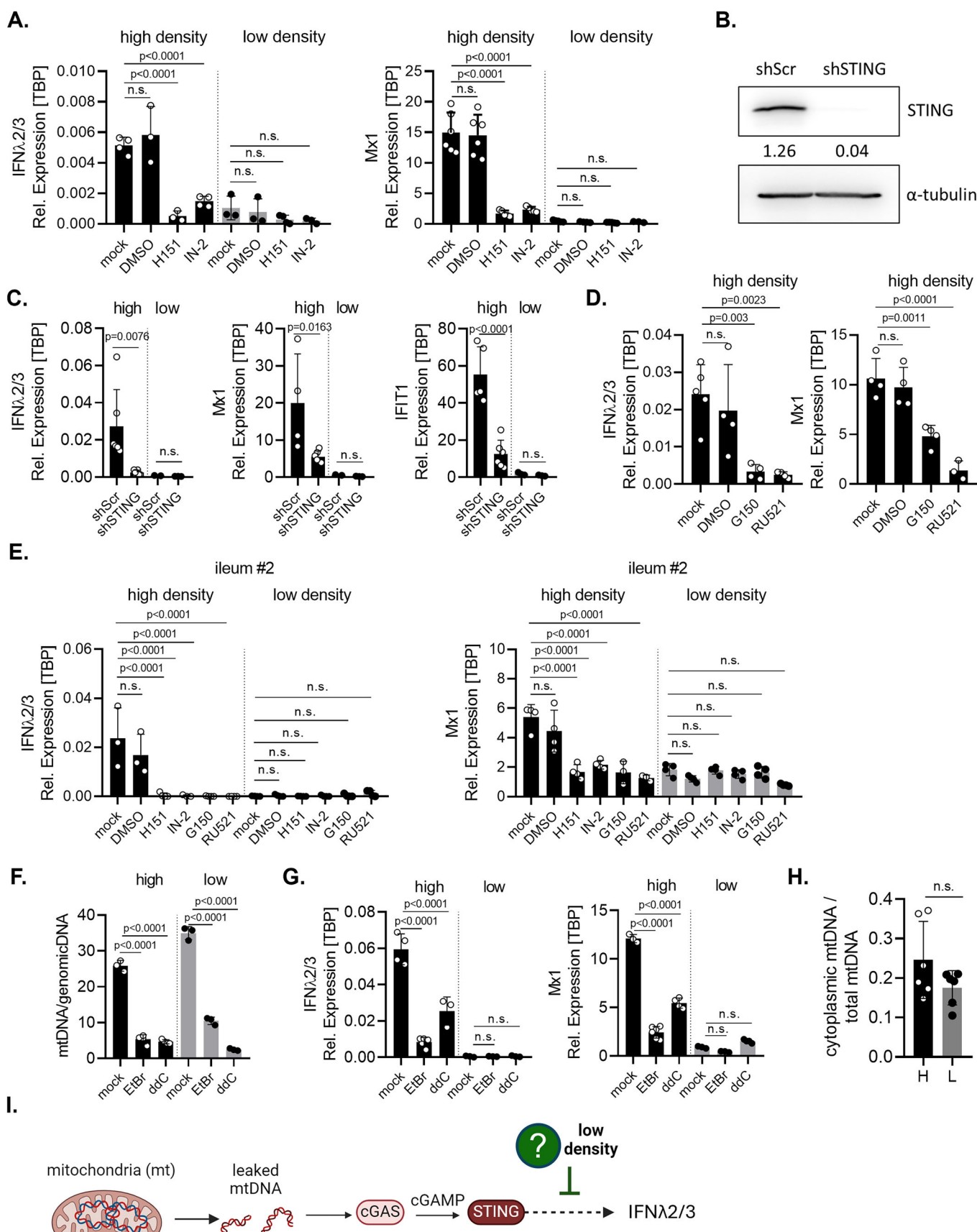

**Figure 3.  cGAS-STING signaling activated by mtDNA drives basal IFN-λ2/3 expression.**

(A) T84 WT cells at high and low cell density were treated with either culture media (mock) or DMSO (20 μM) as a control or treated with the STING-inhibitors H151 (20 μM) or IN-2 (2 μM). IFN-λ2/3 and Mx1 expression was evaluated by qRT-PCR. (B) shRNA-mediated knock-down of STING. T84 cells expressing scrambled shRNA (shScr) or shRNA targeting STING were analyzed by Western blotting. STING protein abundance was quantified relative to α-tubulin. (C) The basal expression of IFN-λ2/3, Mx1 and IFIT1 in T84 cells expressing a scrambled shRNA (shScr) or a shRNA targeting STING (shSTING) at high and low cellular density was addressed using qRT-PCR. (D) T84 WT cells at high and low cell density were treated with either culture media (mock) or DMSO (10 μM) as a control or treated with 10 μM cGAS-inhibitors G150 or RU521. Basal IFN-λ2/3 and Mx1 expression was evaluated by qRT-PCR. (E) Ileum-derived organoids #2 at high and low cell density were treated with either culture media (mock) or DMSO as a control or treated with STING-inhibitors H151 (20 μM), IN-2 (2 μM), or cGAS-inhibitors G150 (10 μM) or RU521 (10 μM). Basal IFN-λ2/3 and Mx1 expression was evaluated by qRT-PCR. (F, G) T84 WT cells seeded at high and low cellular density were depleted of mitochondrial DNA (mtDNA) by 6 days treatment with 300 ng/mL Ethidium Bromide (EtBr) or 100 μg/mL 2′-3′-dideoxycytidine (ddC). (F) Total cellular DNA was harvested and the ratio of mtDNA to genomic DNA (gDNA) was calculated by quantitative PCR. (G) Basal IFN-λ2/3 and Mx1 expression was evaluated by qRT-PCR. (H) To check the mtDNA leakage into the cytosol in high and low cell density, nuclear/cytoplasmic extraction was performed and the abundance of cytosolic mtDNA was evaluated relative to total mtDNA. (I) Schematic model depicting the putative inhibition of basal IFN-λ2/3 expression in cells seeded at low density. (A, C, D, F) The relative expression of IFN-λ2/3, Mx1 and IFIT1 was normalized to TBP. $n \geq 3$ biological replicates. Statistical analysis was performed using (A, C, E–G) two-way ANOVA, (D) one-way ANOVA or (H) unpaired $t$ test between high and low density. n.s. indicates non-significant results ($P > 0.05$). Exact $P$ values are shown on the plots when significant; otherwise, results are not significant. Error bars represent standard deviation with the mean as the center. Source data are available online for this figure.

production (Motwani et al, 2019). To investigate the potential involvement of the STING pathway in the regulation of the differential expression of basal IFN-λ2/3 in cells grown under high and low cellular density, we used two well-characterized STING inhibitors, H151 (Haag et al, 2018) and IN-2 (Hong et al, 2021). To validate the efficacies and control for the toxicities of these inhibitors, T84 WT cells were treated with the STING agonist diABZI (mimicry of 2′3′-cGAMP), both in the presence and absence of H151 and IN-2. Induction of an IFN-mediated immune response was assessed by measuring IFN-λ2/3 and the ISG Mx1 expression using qRT-PCR. As expected, treatment of cell with the STING agonist diABZI resulted in the induction of IFN-λ2/3 and of the ISG Mx1 expression (Appendix Fig. S2A). These inductions were inhibited by the STING inhibitors H151 and IN-2 (Appendix Fig. S2A) and at the concentrations used, the inhibitors were not toxic to cells (Appendix Fig. S3). To directly assess the role of the STING pathway in regulating basal IFN-λ2/3 and ISG expression, T84 WT cells were seeded at high and low cell densities and treated with H151 or IN-2. At high cellular density, STING inhibition significantly decreased basal IFN-λ2/3 and downstream ISG Mx1 expression (Fig. 3A). No major effect was observed at low cell density following STING inhibition (Fig. 3A). These findings strongly suggest that the STING pathway drives the expression of basal IFN-λ2/3 and ISG expression in intestinal epithelial cells grown at high cellular density. To validate these findings, we knocked down STING in T84 cells using a lentivirus-based shRNA approach. Efficient knock-down of STING was confirmed by Western Blot analysis (Fig. 3B). Consistent with the results obtained using the STING inhibitors H151 and IN-2 (Fig. 3A), basal IFN-λ2/3 and downstream ISG expression (Mx1 and IFIT1) at high cell density were significantly reduced in T84 cells knocked down for STING (KD) compared to control T84 expressing a scrambled (scr) shRNA (Fig. 3C).

As STING activation is orchestrated by cGAS, we next aimed to address whether basal IFN-λ2/3 and ISG expression were dependent on cGAS. T84 cells were treated with the cGAS inhibitors G150 (Lama et al, 2019) and RU521 (Vincent et al, 2017). We first confirmed that G150 and RU521 were not toxic to our T84 cells (Appendix Fig. S3). To validate, the efficacy of the cGAS inhibitors G150 and RU521, T84 cells were infected with Vaccinia virus (VV), in the absence or presence of the inhibitors. Vaccinia virus, a double-stranded DNA virus that replicates in the

cytoplasm, is sensed by cGAS, acting as an agonist for this PRR (Smith et al, 2013). Infection of T84 cells with Vaccinia virus significantly induced IFN-λ2/3 expression, which was effectively inhibited by the cGAS inhibitors G150 and RU521 validating the efficacy of these compounds (Appendix Fig. S2B). To confirm that cGAS is driving basal IFN-λ2/3 expression under high cellular confluency, T84 cells were seeded at high density and treated with G150 and RU521. Inhibition of cGAS significantly decreased basal IFN-λ2/3 and downstream ISG expression (Fig. 3D). We further confirmed these findings in ileum-derived organoids. Organoids were seeded at high and low cell densities and treated with either STING inhibitors (H151 and IN-2) or cGAS inhibitors (G150 and RU521). At high density, inhibition of STING or cGAS significantly reduced basal IFN-λ2/3 expression and the downstream ISG Mx1 expression (Fig. 3E). In contrast, at low cell density, STING and cGAS inhibition had no major effect, with the exception of RU521 treatment, which appeared to moderately decrease Mx1 expression compared to the mock-treated control (Fig. 3E).

Together, our results suggest that activation of the cGAS-STING pathway at high cellular density is responsible for basal IFN-λ2/3 and downstream ISG expression in the absence of any pathogenic challenge.

## Mitochondrial-DNA induce the expression of basal IFN-λ2/3 expression in a cGAS-STING-dependent manner

It has been previously shown that leaked cytosolic mitochondrial DNA (mtDNA) can be sensed by cGAS, initiating immune signaling in the absence of pathogenic infection (Tu et al, 2022; Wang et al, 2022). To determine if leaked mtDNA triggers the cGAS-STING pathway at high cell density, mtDNA was depleted in T84 cells with prolonged treatment with low concentrations of Ethidium Bromide (EtBr) or 2′-3′-dideoxycytidine (ddC). Low concentrations of EtBr selectively reduce mtDNA without affecting nuclear DNA (Leibowitz, 1971), and ddC inhibits mitochondrial DNA polymerase γ without affecting nuclear DNA polymerases (Kaguni, 2004). Cytotoxicity assays confirmed that prolonged EtBr and ddC were not toxic to our T84 cells (Appendix Fig. S3). Quantitative-PCR analysis of mtDNA abundance relative to genomic DNA showed efficient depletion of mtDNA after treatments of cells with either EtBr or ddC in both cells seeded at high and low cell densities (Fig. 3F). Basal IFN-λ2/3 and ISG

expression in mtDNA-depleted cells was then assessed using qRT-PCR. We found that basal IFN-λ2/3 and Mx1 expression were significantly decreased in mtDNA-depleted cells at high cellular density (Fig. 3G). These findings suggest that sensing of mtDNA by the cGAS-STING pathway at high cellular density is responsible for basal IFN-λ2/3 and ISG expression in intestinal epithelial cells. As basal IFN-λ2/3 and ISG expression is only observed under high cellular confluency, we next wanted to address whether the significantly reduced basal IFN-λ2/3 and ISG expression observed in low confluent cells correlates with the absence/reduction of cytosolic mtDNA. We measured the abundance of cytosolic mtDNA relative to total mtDNA and found no significant difference between high and low cell densities (Fig. 3H). Collectively, these findings strongly support a model in which the activation of IFN-λ2/3 signaling in a confluent epithelial monolayer is mediated by mtDNA. In this model, mtDNA is detected by cGAS, leading to the activation of STING, which then phosphorylates TBK1 and IRF3, resulting in the expression of IFN-λ2/3 and subsequent downstream induction of ISG expression (Fig. 3I). However, in low cell density growing conditions, while a similar amount of mtDNA is present in the cytosol, it fails to induce the production of IFN-λ2/3 suggesting that a cellular factor may inhibit cGAS-STING dependent signaling at low cellular density (Fig. 3I).

## Hippo signaling governs the inhibition of basal IFN-λ2/3 expression at low cellular density

The Hippo pathway is a highly conserved and essential signaling cascade that senses the cellular environment and population context (e.g. cellular density), integrating various biochemical and biomechanical cues to modulate cellular behavior (Misra and Irvine, 2018). The core components of the Hippo pathway include the protein kinases Mammalia STE20-like kinase 1/2 (MST1/2) and large tumor suppressor 1/2 (LATS1/2), and the co-activator proteins Yes-associated protein 1 (YAP) and transcriptional coactivator with PDZ-binding motif (TAZ). When the Hippo pathway is activated (Hippo ON), MST1/2 phosphorylates and activates LATS1/2, which in turn phosphorylates YAP and TAZ. This phosphorylation leads to the retention of YAP/TAZ in the cytoplasm by 14-3-3 proteins (Meng et al, 2016) (Fig. 4A, left panel). Conversely, when the Hippo pathway is inactive (Hippo OFF), YAP and TAZ translocate to the nucleus and induce the expression of genes associated with cell proliferation, survival, and migration (Fig. 4A, right panel) (Meng et al, 2016). Hippo pathway activity is coupled to cell density; high cell densities induce strong Hippo signaling, which suppresses cell proliferation by retaining YAP in the cytoplasm. In contrast, low cell densities result in weak Hippo signaling, allowing YAP to accumulate in the nucleus and promote cell proliferation (Varelas et al, 2010; Zhao et al, 2007). Previous studies have linked the Hippo pathway to IFN signaling, demonstrating that YAP/TAZ antagonize the innate antiviral response. Given our observations that cell density regulates the expression of basal IFN-λ2/3 and ISGs, we aimed to address whether the Hippo pathway could be involved in regulating the basal immune response in intestinal epithelial cells in a cell density-dependent manner. First, we confirmed that at high cell density, the Hippo pathway is ON which is characterized by the phosphorylation of YAP and by its cytosolic localization. Comparing cells

seeded at high vs. low cell density, we found that YAP was localized in the cytosol of T84 cells seeded at high density (Hippo ON) (Fig. 4B). On the contrary, YAP was observed in the nucleus in T84 cells seeded at low density (Hippo OFF) (Fig. 4B). Complementarily, we found that a greater abundance of phosphorylated YAP in cells seeded at high cellular density compared to low cellular density, while total YAP was similar between high and low density (Fig. 4C). To address whether the Hippo pathway is involved in regulating basal expression of IFN-λ2/3, we inhibited YAP phosphorylation in cells seeded at high density using the selective MST1/2 inhibitor XMU-MP-1/2 (Fan et al, 2016), thereby disengaging YAP from 14-3-3 and potentially allowing YAP to block the phosphorylation of TBK1. This approach represents a mimicry of Hippo OFF (low cell density like). Cell toxicity assays confirmed that XMU-MP-1 was not toxic to our T84 cells (Appendix Fig. S3) and Western blot analysis showed that XMU-MP-1 decreases the levels of phosphorylated YAP (p-YAP) in a dose-dependent manner (Fig. 4D). Concomitantly, we observed decreased levels of phosphorylated TBK1 (p-TBK1) (Fig. 4D) which was associated with a decreased expression of basal IFN-λ2/3 and of the ISG Mx1 (Fig. 4E). These findings strongly suggest a negative correlation between YAP phosphorylation and the inhibition of the TBK1-dependent expression of basal IFN-λ2/3. Phosphorylation of YAP at high cellular density may prevent the inhibition of TBK1 phosphorylation and the subsequently inhibition of basal IFN-λ2/3 expression.

To directly address whether YAP and TAZ are involved in negatively regulating the basal IFN-λ2/3 expression and ISG expression at low cell density (Hippo OFF, Fig. 4A), we knocked-down YAP and TAZ using an shRNA approach. YAP and TAZ knockdown efficiencies were validated by Western blot analysis (Fig. 4F). T84 cells expressing control scrambled shRNA and YAP/TAZ shRNA were cultured at both high and low cell densities, and basal IFN-λ2/3 and ISG Mx1 expression was measured using qRT-PCR (Fig. 4G). As expected, in cells expressing the scramble shRNA, we observed an increase in basal IFN-λ2/3 and ISG Mx1 expression in cells seeded at high density compared to cells seeded at low density (Fig. 4G). Interestingly, in cells knocked-down for YAP and TAZ, we observed a significant increase of basal IFN-λ2/3 and Mx1 expression in cell grown at low density compared to cells expressing the scrambled shRNA. These results strongly suggest that YAP/TAZ inhibit basal IFN-λ2/3 and ISG expression when human intestinal epithelial cells are grown at low cellular density. To determine if the increased of basal IFN-λ2/3 and Mx1 expression in cells knocked-down for YAP and TAZ was mediated in a STING-dependent manner, cells were seeded at low cell density and treated with the STING inhibitor H151. While mock-treated YAP/TAZ knocked down cells showed significantly higher basal IFN-λ2/3 and Mx1 expression compared to the control scrambled shRNA cells (Fig. 4H), this enhanced immune response was effectively suppressed in the presence of the STING inhibitor H151 (Fig. 4H). Together, our findings suggest that under high cellular density, the cGAS-STING pathway senses mtDNA leading to the activation of TBK1 and the subsequent expression of basal IFN-λ2/3. At high cellular density, YAP/TAZ are phosphorylated and cannot interfere with TBK1 activation. On the contrary, under low cellular density, YAP/TAZ are not phosphorylated and can inhibit the activation (phosphorylation) of TBK1 and, consequently, the expression of basal IFN-λ2/3.

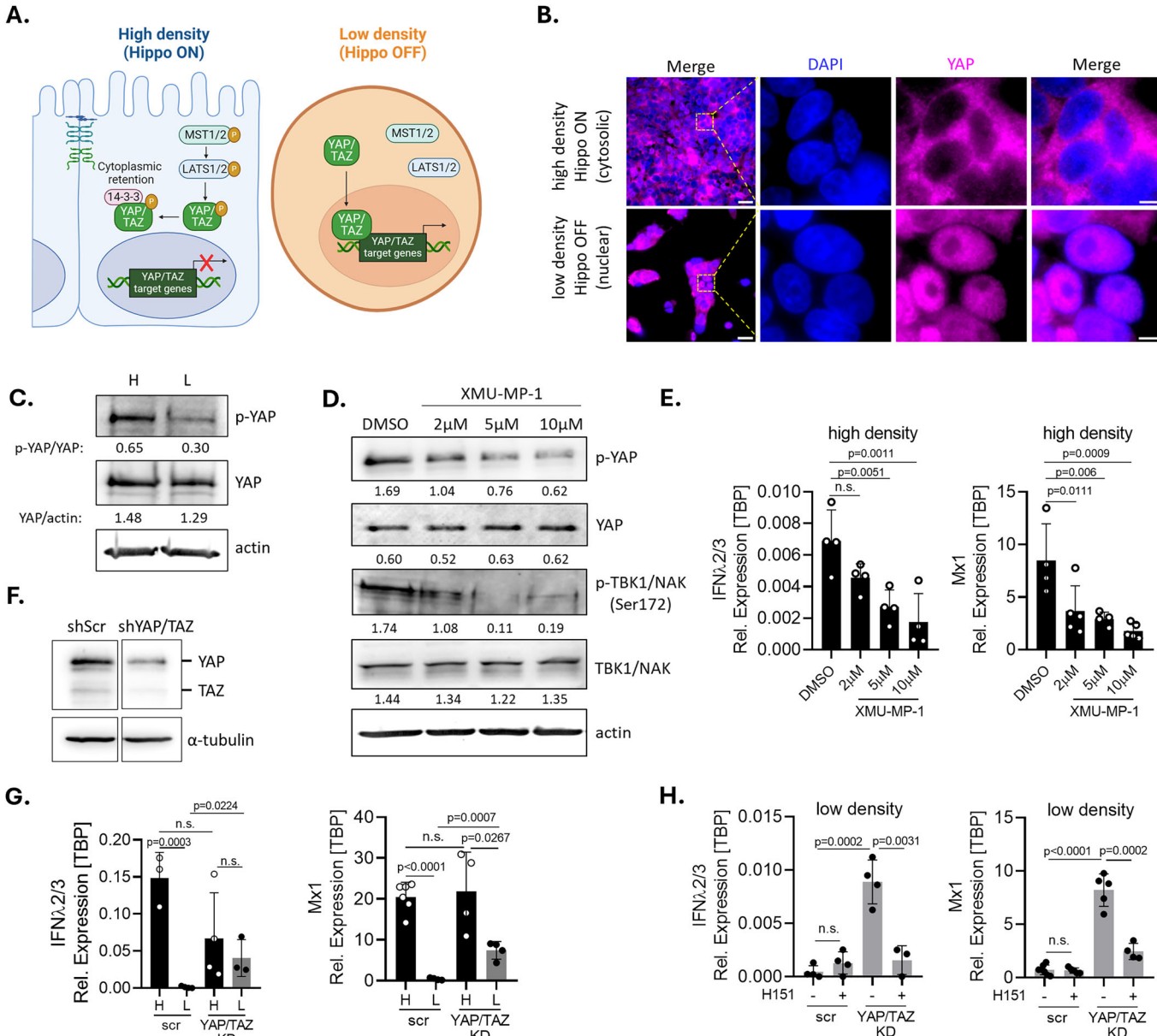

**Figure 4.   Hippo signaling governs the inhibition of basal IFN-λ2/3 expression at low cellular density through YAP-mediated blockade of TBK1 activation.**

(A) Schematic description of Hippo pathway signaling at high and low cellular density. (B) T84 cells seeded at high and low cellular density were immunostained using an antibody directed against YAP (magenta). Nuclei were stained with DAPI (blue). Scale bar = 25 μm for main view (left micrographs) and 5 μm for zoom-in regions. (C) Western blot analysis of the phopho-YAP and YAP protein abundance in T84 cells seeded at high and low density. P-YAP protein abundance was quantified relative to total YAP, and YAP protein abundance was quantified relative to actin as loading control. (D, E) T84 WT cells at high density were treated with DMSO (10 μM) or the MST1/2 inhibitor XMU-MP-1 (at 2, 5 and 10 μM), and harvested. (D) p-YAP, total YAP, p-TBK1/NAK, total TBK1/NAK protein abundances were quantified relative to actin as loading control. (E) The expression of basal IFN-λ2/3 and Mx1 was quantified using qRT-PCR. (F) Western blot analysis validating the efficient knock-down of YAP/TAZ in T84 cells. T84 cells expressing scrambled shRNA (shScr) were used as a control. (G) T84 cells expressing scrambled shRNA (shScr) or shRNA directed against YAP/TAZ (shYAP/TAZ) were seeded at high and low cellular density. Basal IFN-λ2/3 and Mx1 expression were monitored using qRT-PCR. (H) T84 cells expressing scrambled shRNA (shScr) or shRNA directed against YAP/TAZ (shYAP/TAZ) were seeded at low cellular density and mock-treated or treated with the STING inhibitor H151. Basal IFN-λ2/3 and Mx1 expression were monitored using qRT-PCR. The relative expression of IFN-λ2/3 and Mx1 were normalized to TBP. n ≥ 3 biological replicates. Statistical analysis was performed using (E) ordinary one-way ANOVA using the DMSO treated cells as a reference, or (G, H) unpaired *t* test. n.s. indicates non-significant results (*P* > 0.05). Exact *P* values are shown on the plots when significant; otherwise, results are not significant. Error bars represent standard deviation with the mean as the center. Source data are available online for this figure.

## STING-induced basal IFN-λ2/3 signaling regulates barrier formation

To address whether the expression of basal IFN-λ2/3 under high cellular density plays a key role in regulating the function of intestinal epithelial cells, we determined the transcript profiles of WT T84 cells, T84 cells depleted of IFN-λ2/3 (IFN-λ2/3 KO), and T84 cells depleted of the type III IFN receptor (IFNLR KO), grown at high and low cell density. Principal Component Analysis (PCA) revealed that samples clustered per cell lines (WT vs. IFN-λ2/3 KO vs. IFNLR KO) and per growth conditions (high *vs.* low density) (Fig. EV3A). To address the impact of growing cells at high cellular density on their transcript profiles, we first focused our analysis on WT cells. We observed significant differences in the expression profiles of WT cells seeded at high and low cellular densities, as reflected by a PC1 accounting for 95% of the variance (Fig. EV3B). Differential gene expression analysis identified 655 upregulated and 230 downregulated genes in high density conditions compared to low cellular density (Fig. EV3C). Consistent with our previous findings that growing cells at high cellular density induces the expression of basal IFN-λ2/3 (Fig. 1E–G) and ISGs (Figs. 1F,G and EV1A,E,G), we observed that many of the genes upregulated under high cellular density are interferon stimulated genes (Fig. EV3C,D). Importantly, the expression of housekeeping genes such as TBP and HPRT1 was not changed between high and low cell densities (Appendix Fig. S4). Gene Set Enrichment Analysis (GSEA) confirmed our findings that growing cells under high density induces genes related to interferon response pathways (Fig. EV3E).

Depletion of IFN-λ2/3 (IFN-λ2/3 KO) or of the type III IFN receptor (IFNLR KO) results in the loss of ISG expression (Fig. EV3F) confirming the critical role of basal IFN-λ2/3 in regulating basal ISG expression in intestinal epithelial cells. As expected, gene Set Enrichment Analysis (GSEA) between WT and IFN-λ2/3 KO cells, and between WT and IFNLR KO cells at high density revealed significant enrichment in biological processes related to antiviral defense and IFN-dependent signaling in WT cells (Fig. EV3G). Interestingly, we also observed differential regulation of processes related to the extracellular matrix and the apical/basolateral organization of the epithelial membrane in cells lacking basal IFN-λ2/3 signaling (Fig. EV3G). These findings suggest that basal IFN-λ2/3 signaling may have a role in regulating processes related to epithelial cell polarization and/or barrier formation. To address whether IFN-λ2/3 are important for the regulation of intestinal epithelial cell functions related to their polarized nature, we investigated whether lack of IFN-λ2/3 impacted the barrier function of T84 cells. WT T84, IFN-λ2/3 KO, and IFNLR KO cells were seeded on transwell inserts. Ten days post-seeding, formation of tight junctions between epithelial cells was assessed using immunofluorescence staining against the tight junction protein ZO-1. While WT T84 cells developed a tight junction network (tight junction belt) between each individual epithelial cells, T84 cells lacking either the type III IFN receptor (IFNLR KO) or IFN-λ2/3 (IFN-λ2/3 KO) displayed impaired tight junctions (Fig. 5A, top panels). Defects in tight junction formation in IFN-λ2/3 KO and IFNLR KO cells was confirmed by measuring the transepithelial electrical resistance (TEER) (Fig. 5B). Similarly, FITC-Dextran diffusion assays revealed significantly higher barrier permeability in the absence of IFN-λ2/3 signaling (Fig. 5C). This phenomenon was not due to impaired cellular growth or viability,

as fluorescence imaging (Fig. 5A), proliferation assays (Appendix Fig. S5A), and cytotoxicity assays (Appendix Fig. S5B) demonstrated that T84 IFN-λ2/3 KO and IFNLR KO cells successfully proliferated and formed a confluent monolayer as WT cells. Furthermore, gene expression analysis showed that cell viability and apoptotic pathways remained unchanged across both cell densities and cell lines, indicating that the observed phenotypic differences are not driven by alterations in cell survival or apoptosis (Appendix Fig. S5C,D).

To validate the role of IFN-λ2/3 in tight junction formation, we treated IFN-λ2/3 KO cells with increasing concentrations of recombinant IFN-λ2/3 proteins. Immunofluorescence staining of tight junctions revealed that treatment of IFN-λ2/3 KO cells with exogenous IFN-λ2/3 partially restored the integrity of the tight junctions (Fig. 5A, lower panels). Although we observed a reduction in FITC-dextran diffusion at day 5 in IFN-λ2/3 KO cells treated with recombinant IFN-λ2/3, the establishment of TEER and the impairment of FITC-dextran diffusion were not fully rescued upon exogenous treatment (Fig. 5D,E). As exogenous treatment of IFN-λ2/3 KO only partially restored tight junction formation (Fig. 5A, lower panels) with apparent missing tight junctions between cells, establishment of a TEER and impairment of FITC-dextran diffusion could not be obtained.

As we have previously shown that IFN-λ2/3 expression originates from the activation of the cGAS-STING pathway (Fig. 3), we sought to determine whether STING is important for the establishment of tight junctions in epithelial cells. T84 cells were seeded in transwell inserts and treated with the STING inhibitor H151. Interestingly, cells treated with the STING inhibitor displayed defects in tight junction formation (Fig. 5F) and failed to establish a TEER (Fig. 5G). Similarly, STING inhibitor treated cells were also impaired in limiting FITC-dextran diffusion (Fig. 5H). To confirm these findings in primary non-transformed human intestinal epithelial cells, we treated human ileum-derived intestinal organoids with H151. Similar to T84 cells, result showed that STING inhibition impaired tight junction formation in organoids (Fig. EV4A). To address whether this phenotype is conserved among other epithelial cell types, we examined the impact of inhibiting STING signaling on barrier function in airway epithelial cells (Calu-3). Similar to intestinal epithelial cells, Calu-3 cells failed to establish tight junctions in the presence of the STING inhibitor H151 (Fig. EV4B). This was concomitant with a failure of Calu-3 cells to establish a barrier function as measured by TEER and FITC-dextran diffusion assay (Fig. EV4C,D). These findings collectively suggest that expression of basal IFN-λ2/3 through STING-mediated signaling is critical for the development of tight junctions in epithelial cells.

## Claudin-2 upregulation drives barrier dysfunction in cells lacking basal IFN-λ2/3 signaling

To identify the underlying mechanisms by which STING-mediated expression of basal IFN-λ2/3 regulate tight junctions' formation in epithelial cells, we exploited our transcript profiling of WT T84, IFN-λ2/3 KO, and IFNLR KO cells grown under high and low cellular density (Fig. EV3). We compared the expression of the top 50 differentially expressed genes involved in cell–cell adhesion (GO:0098742), cell junction organization (GO:0034330), and cell–cell junction organization (GO:0045216) in T84 WT,

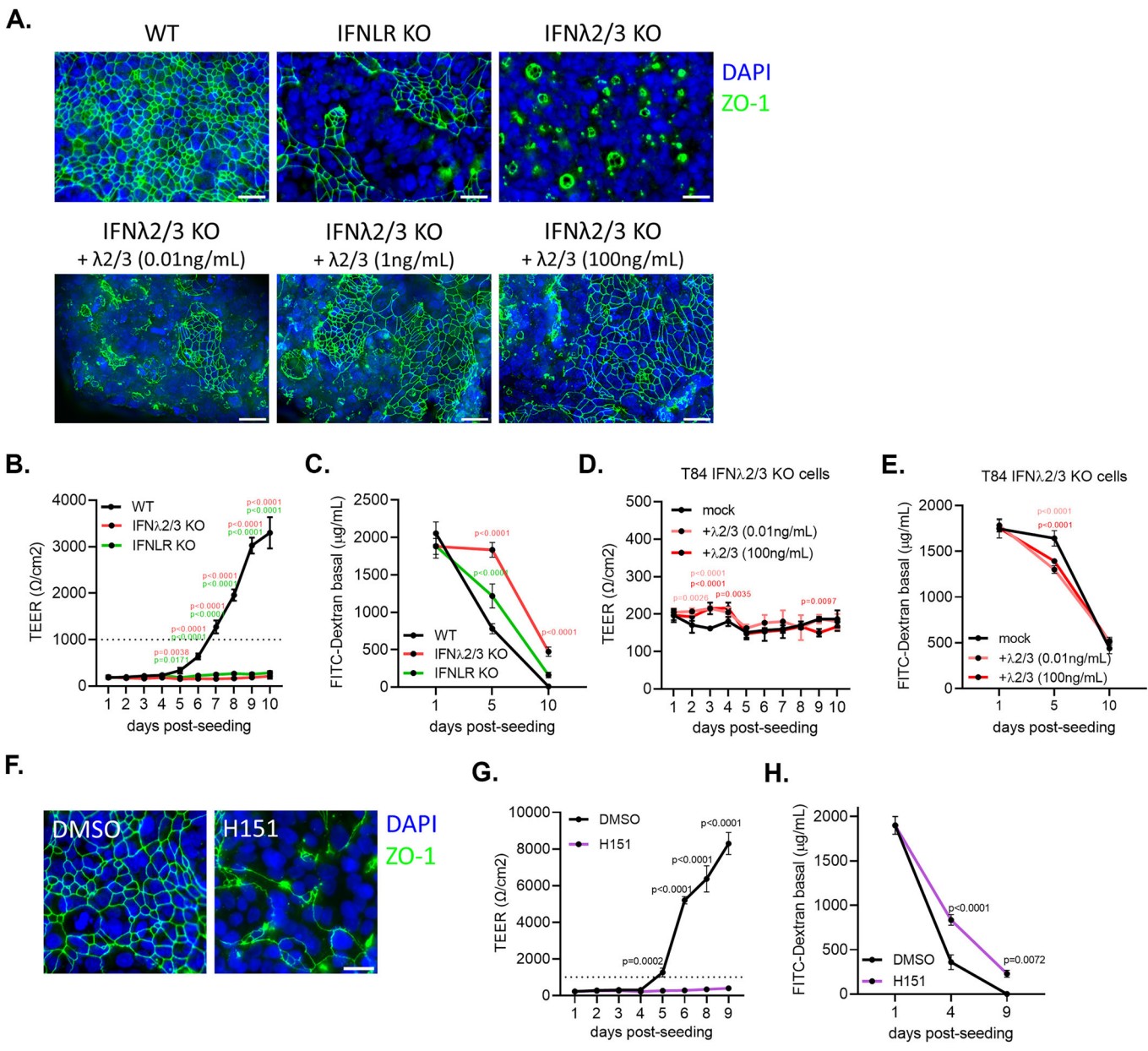

**Figure 5. STING-induced basal IFN-λ2/3 signaling regulates barrier formation in human intestinal epithelial cells.**

(A) Mock-treated T84 WT, IFN-λ2/3 KO, IFNLR KO cells, and IFN-λ2/3-treated (0.01, 1 and 100 ng/mL) T84 IFN-λ2/3 KO cells were fixed with MeOH. Tight junctions were immunostained using an anti-ZO-1 antibody (green). Nuclei were stained using DAPI (blue). Scale bar = 25 μm. (B, C) Establishment of the barrier function and permeability of the monolayer of T84 WT, IFN-λ2/3 KO and IFNLR KO cells seeded on transwell inserts were monitored by TEER measurement (B) and FITC-Dextran Assay (C). (D) TEER measurement and (E) FITC-Dextran permeability assay for T84 IFN-λ2/3 KO cells treated with exogenous IFN-λ2/3 (0.01 and 100 ng/mL). (F–H) T84 WT cells seeded on transwell were mock-treated or treated with 10 μM of the STING inhibitor H151. (F) Tight junctions were immunostained using an anti-ZO-1 antibody (green). Nuclei were stained using DAPI (blue). Scale bar = 25 μm. (G) TEER measurement and (H) FITC-Dextran permeability assay. (B, D, G) Values > 1000 Ω/cm² (dotted line) shows that cells established barrier function. $n \geq 3$ biological replicates. Statistical analysis was performed using two-way ANOVA. n.s. indicates non-significant results ($P > 0.05$). Exact $P$ values are shown on the plots when significant; otherwise, results are not significant. Error bars represent standard deviation with the mean as the center. Source data are available online for this figure.

IFN-λ2/3 KO, and IFNLR KO cells grown at high cellular density (Fig. 6A). Among the genes that were differently expressed in IFN-λ2/3 KO and IFNLR KO cells, we focused our attention on claudin-2 that was upregulated in cells defective for IFNλ signaling (Fig. 6A, red asterisk). Claudin-2 (CLDN2) is a key structural component of tight junctions that regulates epithelial barrier

function by forming paracellular channels for the transport of sodium, potassium, and water. It is typically enriched in "leaky" epithelia, such as the proximal renal tubules and intestinal crypts, where it contributes to normal fluid and electrolyte homeostasis (Luettig et al, 2015). Interestingly, claudin-2, unlike the other 27 mammalian claudins, is distinctively upregulated in most

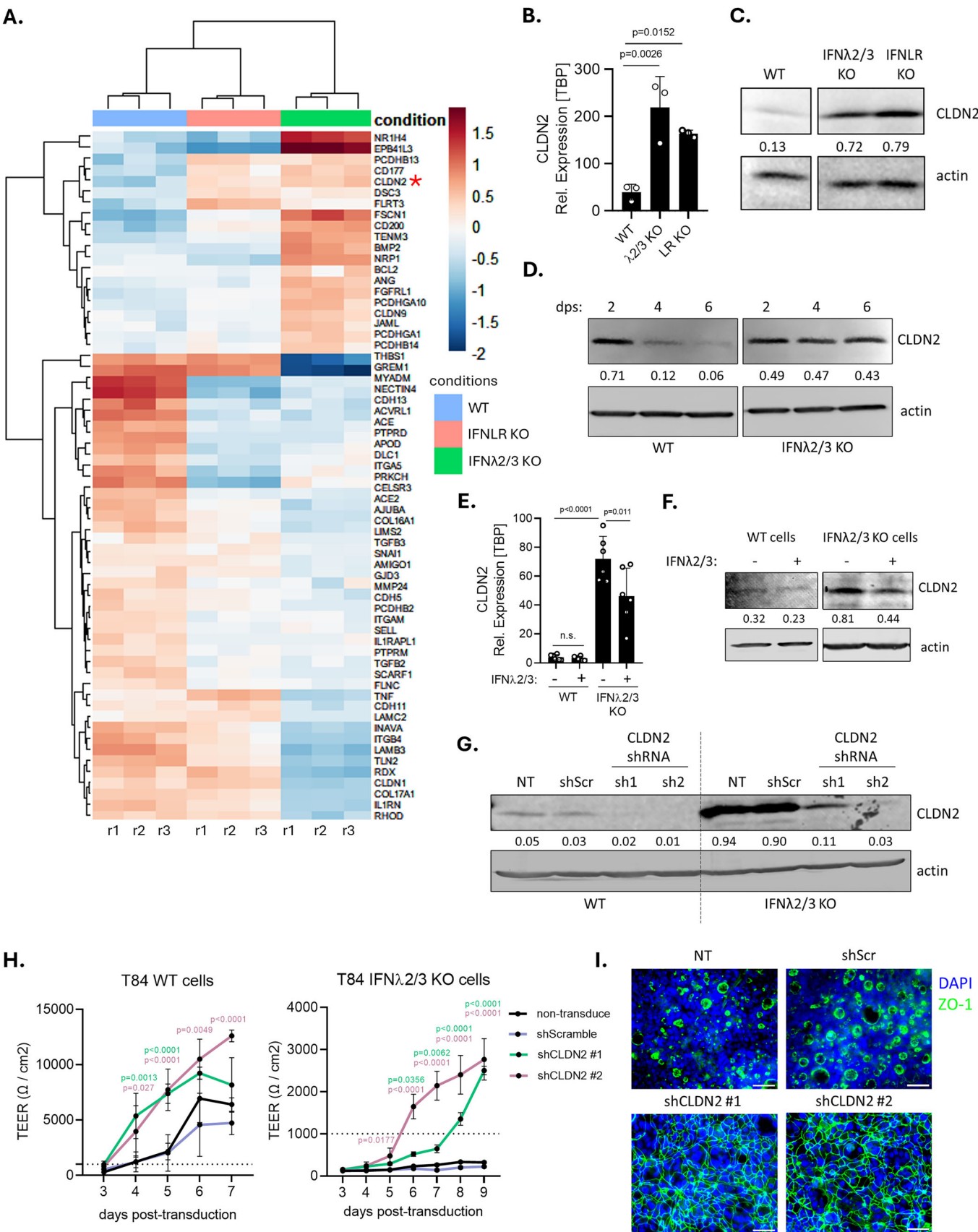

**Figure 6.   Barrier function of human intestinal epithelial cells depends on the downregulation of claudin-2 expression by IFN-λ2/3.**

(A) Heatmap illustrating the differential expression of the top 50 differentially genes involved in cell–cell adhesion (GO:0098742), cell junction organization (GO:0034330), and cell–cell junction organization (GO:0045216) in T84 WT, IFN-λ2/3 KO, and IFNLR KO cells grown at high cell density. Rows correspond to genes, and columns represent samples. Hierarchical clustering was applied to both rows and columns. Red asterisk marks claudin-2. (B) qRT-PCR analysis of claudin-2 expression (relative to the housekeeping gene TBP) in T84 WT, IFN-λ2/3 and IFNLR KO cells. (C) Same as (B) but western analysis using an anti-claudin-2 antibody. Claudin-2 protein abundance was quantified relative to actin as loading control. (D) Claudin-2 protein abundance was assessed by Western blot in T84 WT and IFN-λ2/3 KO cells at day 2, 4 and 6 post-seeding. (E) qRT-PCR analysis of claudin-2 expression (relative to the housekeeping gene TBP) of T84 WT and IFN-λ2/3 KO cells seeded at high density mock-treated or treated with recombinant IFN-λ2/3 (100 ng/mL of each) for 3 days. (F) same as (E) but western-blot analysis using an anti-claudin-2 antibody. Claudin-2 protein abundance was quantified relative to actin as loading control. (G) Knock-down of claudin-2 expression in T84 WT and IFN-λ2/3 KO cells using two distinct shRNA (sh1 and sh2). Scrambled shRNAs were used as control (shScr). Claudin-2 protein abundance was quantified relative to actin as loading control. (H) T84 WT and IFN-λ2/3 KO cells were seeded on transwell inserts and transduced with shCLDN2 #1 and shCLDN2 #2 or scramble shRNA (shScr) at 1 and 4-days post-seeding. Establishment of barrier function was monitored by measuring the TEER. (I) Same as (H) but formation of tight junctions was analysis by indirect immunofluorescence using an anti-ZO-1 antibody at 8 days post-transduction. Nuclei were stained using DAPI (blue). Scale bar = 50 µm. $n \geq 3$ biological replicates. Statistical analysis was performed using (B) ordinary one-way ANOVA or (E, H) two-way ANOVA. n.s. indicates non-significant results ($P > 0.05$). Exact $P$ values are shown on the plots when significant; otherwise, results are not significant. Error bars represent standard deviation with the mean as the center. Source data are available online for this figure.

inflammatory and infectious diseases of the intestine, such as Crohn's disease and ulcerative colitis, and contributes to diarrhea via a leak flux mechanism (Luettig et al, 2015). While CLDN2 has important physiological functions, its aberrant upregulation under pathological conditions compromises the epithelial barrier and has been implicated in the progression of intestinal disorders. Differential gene expression analysis revealed that claudin-2 expression was upregulated in both IFN-λ2/3 KO and IFNLR KO cells compared to WT cells (Fig. 6A, red asterisk). We confirmed this upregulation of claudin-2 at both the transcript levels (Fig. 6B) and at the protein level (Fig. 6C) in WT and IFN-λ2/3 KO and IFNLR KO T84 cells. To assess how claudin-2 expression is regulated during epithelial cell polarization and tight junction formation, T84 cells and IFN-λ2/3 KO cells were seeded, and claudin-2 levels were measured at days 2, 4, and 6 post-seeding. Results indicate a progressive decrease in claudin-2 expression over time in wild-type (WT) T84 cells (Fig. 6D, left panel). However, in IFN-λ2/3 KO cells, claudin-2 levels were not downregulated (Fig. 6D, right panel), suggesting a role for IFN-λ2/3 in modulating claudin-2 expression as epithelial cells become confluent and polarize.

To confirm that claudin-2 expression is regulated by IFN-λ2/3 signaling, WT T84 cells and IFN-λ2/3 KO cells at high cellular density were treated with recombinant IFN-λ2/3 proteins. Upon exogenous treatment, WT cells did not show any change in claudin-2 transcript levels (Fig. 6E), however, the elevated claudin-2 expression in IFN-λ2/3 KO cells significantly decreased at both the transcript (Fig. 6E) and protein levels (Fig. 6F). Together these findings unraveled a previously unknown link between IFN-λ2/3 and claudin-2 expression. In cells expressing IFN-λ2/3 (WT cells at high confluency), claudin-2 expression levels are low. On the contrary, in cells that express limited amount to no IFN-λ2/3 (IFN-λ2/3 KO and IFNLR KO cells), claudin-2 expression is upregulated.

To determine whether claudin-2 is responsible for the loss of barrier function in cells lacking basal IFN-λ2/3 signaling, we silenced the expression of claudin-2 in both WT and IFN-λ2/3 KO cells using two different shRNAs. After confirming that both shRNAs successfully knocked-down claudin-2 in both WT and IFN-λ2/3 KO cell lines (Fig. 6G), we measured the TEER to assess the effect of claudin-2 knock-down on barrier integrity. A significant increase in TEER was observed in T84 WT cells knocked-down for claudin-2 compared to control cells (non-transduced cells and cells expressing a scramble shRNA) (Fig. 6H, left panel). Interestingly, in IFN-λ2/3 KO cells, claudin-2 silencing

by both shRNAs restored the barrier function of T84 cells as measured by TEER (Fig. 6H, right panel). Concomitantly, formation of the ZO-1 tight junction belt was also restored in IFN-λ2/3 KO cells knocked-down for claudin-2 (Fig. 6I). These findings demonstrate that IFN-λ2/3 negatively control the expression of claudin-2 in T84 epithelial cells. In cells lacking IFN-λ2/3 expression, claudin-2 expression is upregulated and in turn negatively impacts formation of tight junction and establishment of a barrier function in intestinal epithelial cells.

Altogether, our study has revealed the complex interplay of basal IFN-λ2/3 signaling in epithelial cells. As epithelial cells polarize and form a tight barrier, YAP/TAZ are inactivated and retained in the cytosol. This retention allows for mtDNA to be sensed by the STING pathway and for IRF3 to drive the expression of IFN-λ2/3. In turn, IFN-λ2/3 negatively regulates the expression of claudin-2. This decrease in claudin-2 levels, increases the barrier function of epithelial cells preventing a "leaky" barrier (Fig. 7).

## Discussion

In this study, we uncovered a novel function of basal IFNλ in controlling the barrier function of human intestinal epithelial cells by regulating formation of tight junctions between epithelial cells. We discovered that expression of basal IFN-λ2/3 is driven by the cGAS-STING signaling pathway through sensing of mtDNA, which subsequently activates TBK1 and IRF3 that ultimately leads to the expression of IFN-λ2/3. Expression of basal IFN-λ2/3 is regulated in a cell density-dependent manner with cells expressing more IFN-λ2/3 as they grow as a confluent cell monolayer. We found that the Hippo signaling pathway inhibits basal IFN-λ2/3 expression through YAP-mediated suppression of TBK1 activation at low cellular density (Fig. 7). On the contrary, when cells reach high cellular confluency, the Hippo pathway is tuned ON and YAP is no longer able to interfere with the STING-dependent activation of TBK1 (Fig. 7). Importantly, preventing basal IFN-λ2/3 expression/ signaling through genetic ablation of either the type III IFN receptor or IFN-λ2/3 or through inhibition of the STING signaling pathway results in impaired tight junction formation and loss of barrier function in human epithelial cells. We functionally could show that this loss of barrier function was due to the increased expression of the tight junction protein claudin-2 in cells lacking basal IFN-λ2/3 expression/signaling (Fig. 7). Collectively, our study

## High density

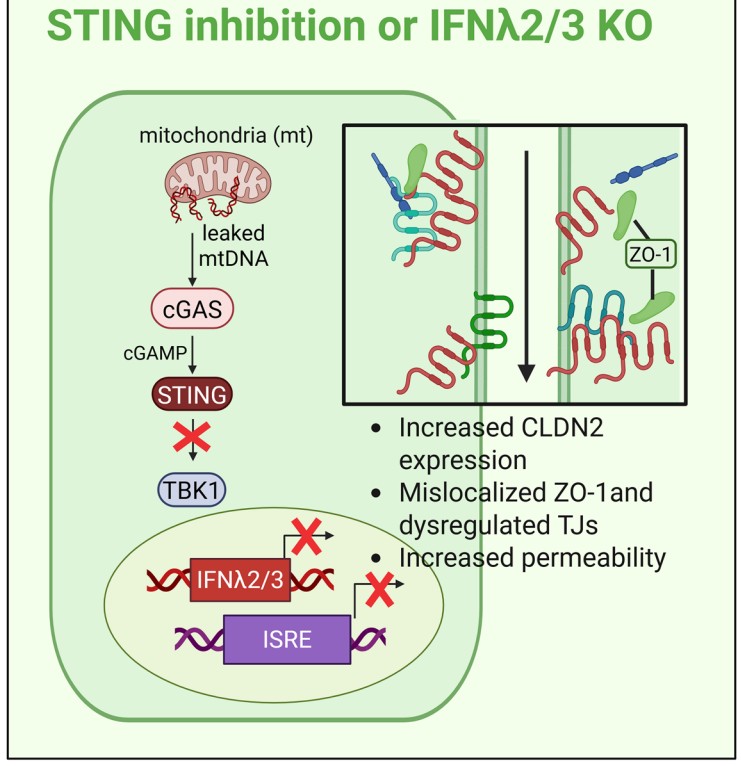

## Low density

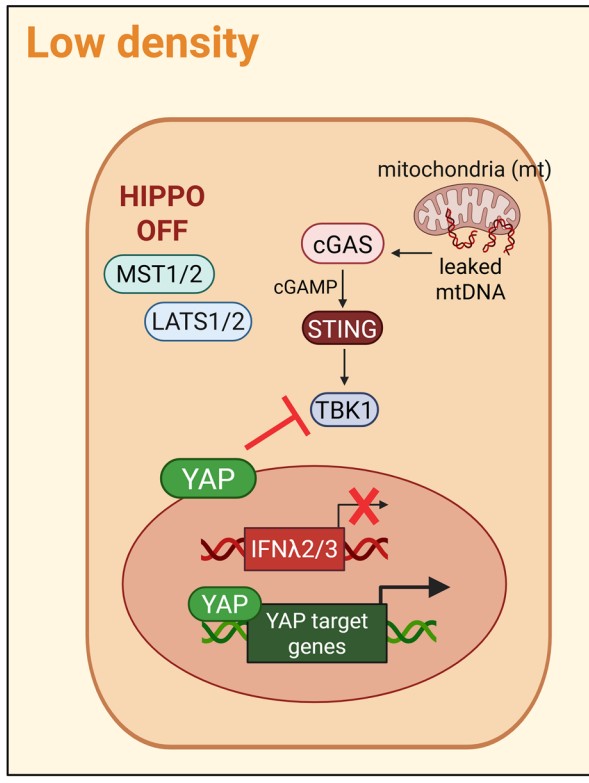

## STING inhibition or IFNλ2/3 KO

**Figure 7. Density-dependent STING-induced IFN-λ2/3 signaling regulates epithelial barrier integrity, permeability and polarization.**

Top panel: At high density, cGAS-STING pathway activation by mtDNA induces basal IFN-λ2/3 expression via TBK1-IRF3 axis. Secreted IFN-λ2/3 binds its heterodimeric receptor (IFNLR/IL10RB) and initiates JAK/STAT-dependent ISG expression. IFN-λ2/3-driven ISGs decrease CLDN2 expression, which promotes the epithelial barrier integrity. Bottom left panel: At low density Hippo pathway component YAP acts as a negative regulator of IFN-λ2/3 signaling by inhibiting TBK1 activation. This leads to lack of IFN-λ2/3 induction. Bottom right panel: Without STING-induced IFN-λ2/3 signaling, cells fail to establish intact barrier due to upregulated CLDN2 level, which causes the mislocalization and dysregulation of tight junctions (TJs). Source data are available online for this figure.

revealed that basal IFN-λ2/3 expression in epithelial cells is critical for the establishment of epithelial barrier function, illustrating that basal IFN-λ2/3 signaling serves roles beyond its antiviral functions.

IFNs are cytokines that are induced when cells encounter a pathogen and sense pathogen-associated molecular patterns (PAMPs) through their pattern recognition receptors (PRRs). Additionally, IFNs can be induced upon cellular stress where pattern recognition receptors sense damage-associated molecular patterns (DAMPs). Despite the well-controlled signaling that leads to IFN production upon activation of pattern recognition receptors, cells also produce basal IFN levels at steady state, without pathogen triggers. In murine models, intestinal basal IFN levels are defined as the IFNs that are produced in homeostatic conditions in the absence of pathogens. Immune cells and to a lesser extent intestinal epithelial cells are responsible for the production of IFNs through sensing of the intestinal luminal microbial content. This microbiota-dependent induction of IFN leads to the expression of IFN stimulated genes in discrete villus structures throughout the intestinal tract ("hot pockets") (Erttmann et al, 2022; Guttman and Finlay, 2009; Van Winkle et al, 2022). Interestingly, we have previously shown that in the absence of pathogens or even microbiota, isolated intestinal epithelial cells produce basal levels of type III IFNs (Karlowitz et al, 2022). We here show that this basal expression of IFN-λ2/3, in a sterile environment, is due to the sensing of the damage-associated molecular pattern mitochondrial DNA (mtDNA) by the DNA sensor cGAS-STING pathway (Fig. 3). Prior studies using murine cells have shown that the cGAS-STING pathway is active at steady state, leading to basal type I IFN production through the sensing of mtDNA (Tu et al, 2022; Wang et al, 2022). While our findings support a central role for cGAS in this response, it is important to note that depletion of mtDNA may also impact cellular metabolism and reduce the MAVS signaling platform, potentially affecting other innate immune pathways. Together, these findings underscore the essential role of the cGAS-STING pathway in generating a basal innate immune state in cells in a PAMP-free environment through sensing of DAMPs.

The cGAS-STING signaling pathway requires tight regulation to maintain cellular and organismal homeostasis; its dysregulation is associated with various autoinflammatory pathologies including ulcerative colitis (UC), Crohn's disease (CD), psoriasis, systemic lupus erythematosus (SLE) (Liu and Pu, 2023). Importantly, these diseases are linked to the dysregulation of mucosal barriers. Here we describe, for the first time, that the STING-dependent basal IFN-λ2/3 expression is critical for the formation of tight junctions and establishment of barrier function in epithelial cells (in intestinal cell lines (Fig. 5), primary intestinal organoids (Fig. EV4A) and airway epithelial cells (Fig. EV4B–D)). In the absence of STING and/or IFN-λ2/3 signaling, cells cannot form an intact barrier. However, how dysregulation of basal IFN-λ2/3 expression in diseases that display aberrant cGAS-STING signaling

contribute to disease development (loss of barrier function) remains unknown.

We show that as cells become confluent, basal IFN-λ2/3 expression increases. Interestingly, the DAMP mtDNA levels in cells grown at low vs. high confluency does not correlate with the amount of basal IFN-λ2/3 expression (Fig. 3F–H). Cells grown at low confluency contain higher amounts of mtDNA than cells at high confluency (Fig. 3F). We identified that the Hippo pathway through YAP/TAZ regulates basal IFN-λ2/3 expression through the cGAS-STING dependent sensing of mtDNA (Fig. 4). The Hippo pathway is a key sensor of the cellular environment and population context, and it integrates diverse biochemical and biomechanical cues to modulate behavior (Misra and Irvine, 2018). Activation of the Hippo pathway (Hippo ON) induces a kinase cascade in which MST1/2 phosphorylates and activates LATS1/2. Activated LATS1/2 phosphorylates YAP and TAZ. This leads to 14-3-3 mediated YAP/TAZ cytoplasmic retention (Meng et al, 2016). When the Hippo pathway is inactive (Hippo OFF), YAP and TAZ are active and translocate to the nucleus (Meng et al, 2016). There they interact with the TEAD transcription factor family, inducing the expression of a wide set of genes associated with cell proliferation, survival, and migration (Meng et al, 2016). Interestingly, previous studies established a link between the Hippo pathway and IFN signaling, in which YAP/TAZ antagonize the innate antiviral response. Zhang et al (Zhang et al, 2017) demonstrated that at low cellular density, YAP/TAZ hijack TBK1, preventing its activation upon pathogen detection by the cells. In contrast, they proved that at high density the Hippo pathway is activated (Hippo ON), leading to the degradation of YAP/TAZ and facilitating unhindered TBK1 signaling upon pathogenic infection (Zhang et al, 2017). Additionally, it was demonstrated that YAP can associate with IRF3 to repress its dimerization, thereby also inhibiting immune signal transduction (Wang et al, 2017). We here show that YAP/TAZ not only inhibit IFN expression during pathogenic challenges but can also inhibit the cGAS-STING mediated basal IFN-λ2/3 expression upon sensing of the DAMP mtDNA.

In addition to their antiviral effects, IFNs act on epithelial surfaces by supporting physical barrier formation, thereby indirectly hindering infections by preventing pathogen intrusion. In a mechanism independent of STAT1 signaling, both type I and III IFNs induce cell junction tightening in brain microvasculature endothelial cells (Daniels et al, 2014; Lazear et al, 2015). This in turn reduces the blood brain barrier permeability, protecting mice from virus neuroinvasion (Daniels et al, 2014; Douam et al, 2017; Lazear et al, 2015). The effect of IFNs on epithelial barrier function is not unique to the central nervous system, but has also been reported for the respiratory and gastrointestinal tract. Type I IFNs in the airway epithelium induced the production of tight junctions thereby preventing bacteria transmigration (LeMessurier et al, 2013), and counteracted tight junction dissociation during rhinovirus infection (Douam et al, 2017).

Similarly, type III IFNs enhanced intestinal barrier formation, protecting human intestinal epithelial cells from bacteria (Odendall et al, 2017) and *Cryptosporidium parvum* (Ferguson et al, 2019) infection. Contrary to this, recent studies demonstrated that sustained IFNλ treatment disrupts the airway epithelial barrier, thereby predisposing the host to bacterial superinfections (Ahn et al, 2019; Broggi et al, 2020). Similarly, it was reported that both type I and III IFNs disrupt airway epithelial repair during recovery from viral infections (Major et al, 2020) and during recovery of the intestinal epithelium in a DSS-induced colitis model (Broggi et al, 2020). Here we show that basal IFN-λ2/3 regulates tight junction formation and barrier function in human intestinal epithelial cells in the absence of pathogenic or microbial stimulation. Interestingly, in IFNLR KO cells, we observe a partial but non-uniform ZO-1 localization at the membrane, suggesting that while some tight junctions can form, they may be unstable or improperly organized. In IFNLR KO cells none of the IFNλs can signal while in the IFN-λ2/3 KO cells IFNλ1 is still present. As both IFNλ1 and IFN-λ2/3 signal on the same receptor, further work is necessary to address whether IFNλ1 and IFN-λ2/3 impact tight junction functions differently. Similarly, IFNλ4 is expressed in only a small portion of the human population, especially in certain ethnic groups, and further studies need to be done to investigate the role of IFNλ4 in the tight junction formation. Interestingly, in the IFN-λ2/3 KO cells, exogenous IFN-λ2/3 treatment partially rescues ZO-1 localization, but does not fully restore a WT-like phenotype. This incomplete rescue may reflect the need for sustained or temporally coordinated IFNλ signaling during key stages of epithelial polarization that transient treatment cannot fully replicate.

Claudin-2 is expressed in the tight junctions of leaky epithelia, and the upregulation of Claudin-2 is a key intermediate contributing to dysbiosis, intestinal damage, inflammation, and ineffective pathogen control (Luettig et al, 2015). Importantly, strong evidence now also supports a significant correlation between disease stage and claudin-2 abundance (Luettig et al, 2015). In intestinal inflammation (Crohn's disease, ulcerative colitis) and immune-mediated diseases (celiac disease), claudin-2 is upregulated in the small and large intestine and contributes to diarrhea via a leak flux mechanism (Luettig et al, 2015; Prasad et al, 2005). Consistent with previous literature (Prasad et al, 2005), we found that as cells polarize and establish a monolayer, claudin-2 levels decrease. Importantly, in cells lacking IFN-λ2/3 expression and signaling, claudin-2 expression is not downregulated upon reaching cell confluency. Silencing of claudin-2 in these cells restores barrier function and tight junction formation, further demonstrating the importance of claudin-2 in regulating tight junctions. Most importantly, these findings strongly suggest a model where negative regulation of claudin-2 expression by IFN-λ2/3 is responsible for formation of a tight barrier. This model is supported by our finding that exogenous treatment of IFN-λ2/3 KO cells with IFN-λ2/3 reduces claudin-2 transcript and protein levels. We recently identified two patients with very early onset IBD that express mutations in their IFNλ2 and IFNλ3 genes (Ouahed et al, 2022). Further work is necessary to determine whether mutations in the IFN-λ2/3 loci are responsible for an impaired IFN-λ2/3 mediated downregulation of claudin-2 expression leading to a defective barrier function of the intestinal epithelium as such constituting a trigger for the onset of disease development.

In conclusion, in this work we identified that YAP-TAZ-mediated mechanotransduction regulates basal IFN-λ2/3 expression in human epithelial cells of the airways and intestinal tract. This expression of IFN-λ2/3 is fundamental to negatively regulate the expression of claudin-2 to allow for the development of tight junctions and the establishment of barrier function. Further research is necessary to determine whether aberrant basal IFN-λ2/3 expression can be responsible for onset of pathology where epithelial barrier functions are impaired. A better understanding of the function of basal IFN-λ2/3 might provide novel opportunities to develop both prophylactic and therapeutic interventions.

# Methods

**Reagents and tools table**

| Reagent/resource | Reference or source | Identifier or catalog number |
|---|---|---|
| **Experimental models** | | |
| T84 cells | ATCC | Cat #CCL-248 |
| Calu-3 cells | ATCC | Cat #HTB-55 |
| HEK293T cells | ATCC | Cat #CRL-3216 |
| Human ileum-derived organoids | Triana et al, 2021 | |
| **Recombinant DNA** | | |
| plentiCRISPR V2 blasti | Addgene | Cat #52961 |
| pMDG.2 | Addgene | Cat #12259 |
| psPAX | Addgene | Cat #12260 |
| pLKO.1 neo | Addgene | Cat #13425 |
| pLKO.1 puro | Addgene | Cat #8453 |
| **Antibodies** | | |
| Alpha-Tubulin | Sigma | Cat #T9026 |
| β-actin | Sigma | Cat #A5441 |
| Lamin-B1 | Santa Cruz | Cat #sc-374015 |
| IRF3 | Cell Signaling | Cat #11904 T |
| phospho-IRF3 | NEB/Cell Signaling | Cat #4947 |
| ISG15 | Santa Cruz | Cat #166755 |
| Claudin-2 | Invitrogen | Cat #51-6100 |
| ZO-1 | Invitrogen | Cat #40-2200 |
| ZO-1 | Invitrogen | Cat #33-9100 |
| STAT1 | BD Biosciences | Cat #610115 |
| phospho-STAT1 | BD Biosciences | Cat #612233 |
| STING | Cell Signaling | Cat #13647 |
| TBK1/NAK | Cell Signaling | Cat #3013S |
| phospho-TBK1/NAK | Cell Signaling | Cat #5483S |
| YAP/TAZ | Santa Cruz | Cat #sc-101199 |
| YAP/TAZ | Cell Signaling | Cat #8418 |
| phospho-YAP | Cell Signaling | Cat #4911S |
| anti-mouse-IgG HRP | Abcam | Cat #ab6789 |
| anti-rabbit-IgG HRP | Abcam | Cat #ab97051 |
| anti-mouse-IgG IRDye® 680RD | Licor | Cat #926-68073 |
| anti-rabbit-IgG IRDye® 800CW | Licor | Cat #926-32210 |

| Reagent/resource | Reference or source | Identifier or catalog number |
| --- | --- | --- |
| Alexa Fluor® conjugated secondary antibodies | Invitrogen | Cat #P36941 |
| **Oligonucleotides and other sequence-based reagents** | | |
| qPCR primers | This study | Appendix Table S2 |
| sgRNAs for IFNλ2 and IFNλ3 knock out | This study | Methods: Generation of T84 knock-out cell lines |
| shRNAs for YAP/TAZ, STING and CLDN2 | This study | Methods: Generation of T84 knock-down cell lines |
| IFN-λ2/3 PCR primers | This study | Methods: Genomic DNA isolation, PCR and gel extraction |
| **Chemicals, enzymes and other reagents** | | |
| DMEM:F12 (1:1 mix) | Gibco | Cat #11320033 |
| DMEM | Gibco | Cat #31965 |
| Fetal Bovine Serum (FBS) | Sigma-Aldrich | Cat #12306C |
| Penicillin-Streptomycin | Gibco | Cat #15140122 |
| Rat tail collagen (0.01 mg/mL in 60% EtOH) | Sigma-Aldrich | Cat #C7661 |
| Human collagen (0.04 mg/mL in water) | Sigma-Aldrich | Cat #C5533 |
| 0.25% Trypsin-EDTA | Gibco | Cat #25200056 |
| 0.05% Trypsin-EDTA | Gibco | Cat #25200054 |
| Polyethylenimine (PEI) | Polysciences | Cat #23966-100 |
| Polybrene | Sigma-Aldrich | Cat #TR-1003-G |
| Blasticidin | Invivogen | Cat #ant-bl-1 |
| Puromycin | Invivogen | Cat #ant-pr-1 |
| Neomycin (G418) | Corning | Cat #30-234-CI |
| Opti-MEM | Gibco | Cat #31985062 |
| Matrigel | BioCorning | Cat #354230 |
| Compounds and concentrations for human organoid basal and differentiation media | This study | Appendix Table S1 |
| Advanced DMEM/F12 | Gibco | Cat #12634010 |
| GlutaMAX | Gibco | Cat #35050061 |
| HEPES | Sigma-Aldrich | Cat #H7523 |
| Penicillin-Streptomycin | Gibco | Cat #15140122 |
| L-WRN cell conditioned supernatant | ATCC | Cat # CRL-3276 |
| R-Spondin cell conditioned supernatant | Kind gift from Calvin Kuo, Stanford University | — |
| B-27 Supplement | Thermo Fisher | Cat #17504001 |
| EGF (recombinant mouse) | Gibco | Cat #PMG8041 |
| A83-01 | Millipore Sigma | Cat #SML0788 |
| IGF-1 (recombinant human) | Thermo Fisher | Cat #590908 |
| FGF-basic (recombinant human) | Peprotech | Cat #100-18B |
| Noggin (recombinant mouse) | Peprotech | Cat #250-38-100UG |

| Reagent/resource | Reference or source | Identifier or catalog number |
| --- | --- | --- |
| Gastrin | Sigma-Aldrich | Cat #G9145 |
| N-acetyl-cysteine | Sigma-Aldrich | Cat #A9165 |
| Y27632 hydrochloride | Cayman Chemical | Cat #10005583 |
| 0.05% Trypsin-EDTA | Gibco | Cat #25200054 |
| BX795 | MedChemExpress | Cat #HY-10514 |
| H151 | MedChemExpress | Cat #HY-112693 |
| STING IN-2 | MedChemExpress | Cat #HY-138682 |
| diABZI | Selleckchem | Cat #S8796 |
| G150 | MedChemExpress | Cat #HY-128583 |
| RU521 | MedChemExpress | Cat #HY-114180 |
| Etidium Bromide (EtBr) | MP Biomedicals | Cat #ETBC100 |
| 2′-3′-dideoxycytidine (ddC) | Sigma-Aldrich | Cat #D5782 |
| XMU-MP-1/2 | MedChemExpress | Cat #HY-100526 |
| DMSO | Sigma-Aldrich | Cat #D8418 |
| FITC-Dextran (4 kDa) | Sigma-Aldrich | Cat #46944 |
| iSCRIPT Reverse Transcription Supermix | Bio-Rad | Cat #1708890 |
| iTaq Universal SYBR Green Supermix | Bio-Rad | Cat #1725124 |
| Protease inhibitors (cOmplete™ Mini EDTA-free) | Sigma-Aldrich | Cat #11836170001 |
| Phosphatase inhibitors (PhosSTOP) | Sigma-Aldrich | Cat #4906837001 |
| NP40 | Thermo Scientific | Cat #85124 |
| InterOpt® (TBS) Blocking Buffer | Licor | Cat #927-60001 |
| 0.2 μm Nitrocellulose membrane | Bio-Rad | Cat #1704158 |
| Pierce ECL Western Blotting Substrate | Thermo Fisher | Cat #32209 |
| Pierce BCA Protein Assay Kit | Thermo Fisher | Cat #23225 |
| Paraformaldehyde (PFA) | Roth | Cat #P6148-1KG |
| Bovine Serum Albumin (BSA) | Sigma-Aldrich | Cat #A3294 |
| DAPI | Invitrogen | Cat #P36941 |
| Phusion Hot Start II DNA Polymerase | Thermo Fisher | Cat #F518L |
| Phusion HF Buffer | Thermo Fisher | Cat #F-549S |
| Prime iTaq SYBR green | Bio-Rad | Cat #1725124 |
| PPMP | Cayman Chemicals | Cat #17236 |
| **Software** | | |
| Fiji ImageJ | https://imagej.net/software/fiji/ (Schindelin et al, 2012) | — |
| Live cell imaging software | ilastik 1.2.0 | — |
| CellProfiler 3.1.9 | Open-source | — |
| BioRender | Open-Source | — |
| BLAST | Altschul et al, 1990 | — |

| Reagent/resource | Reference or source | Identifier or catalog number |
|---|---|---|
| R | https://www.r-project.org/ (R Core Team, 2023) | — |
| Prism | GraphPad | — |
| **Other** | | |
| 0.45-µm Syringe Filter | Lab Unlimited | Cat #W10462100 |
| 8-well chamber slide | Ibidi | Cat #80827-90 |
| Transwell Inserts | Thermo Fisher | Cat #3415 |
| 48-well plate | Fisher | Cat #3548 |
| EVOM3 Epithelial Volt/Ohm Meter | World Precision Instruments | — |
| STX2-PLUS Electrode | World Precision Instruments | — |
| Real-Time PCR System CFX Opus 96 | Bio-Rad | Cat #12011319 |
| 800TS Microplate Reader | BioTek | — |
| Trans-Blot® Turbo™ Transfer System | Bio-Rad | — |
| ImageQuant™ LAS 4000 | GE Healthcare | — |
| Odyssey M Imaging System | Licor | — |
| ZEISS Celldiscoverer-7 (CD7) Widefield microscope | ZEISS | — |
| Monarch® Genomic DNA Purification Kits | NEB | Cat #T3010S |
| Monarch DNA Gel Extraction Kit | NEB | Cat #T1020S |
| LDH-Glo™ Cytotoxicity Assay | Promega | Cat #J2380 |
| Sanger sequencing | GENEWIZ (Azenta Life Sciences) | — |
| PureLink™ Genomic DNA Mini Kit | Thermo Fisher | Cat #K182002 |
| RNeasy Plus Mini Kit | Qiagen | Cat #74134 |
| Illumina NovaSeq 6000 | Illumina | — |

## Cell lines and cell culture

Wild-type (WT) T84 (ATCC CCL-248) as well as T84 reporter and knock-out (KO) cells were cultured in a 50:50 mixture of Dulbecco's Modified Eagle's Medium (DMEM) and F12 (Gibco #11320033). T84 cells expressing mCherry under the control of the Mx1 promoter region (pMx1-mCherry) were previously generated in our laboratory (Doldan et al, 2022). Calu-3 (ATCC #HTB-55) and HEK293T (ATCC #CRL-3216) cells were grown in DMEM (Gibco #31965). All media were supplemented with 10% fetal bovine serum (FBS) (Sigma-Aldrich #12306C) and 100 U/mL penicillin and 100 µg/mL streptomycin (Gibco #15140122). All cell lines were authenticated by STR profiling. All cell lines were tested for mycoplasma contamination biweekly. T84 and Calu-3 cells were cultured on collagen coated surfaces. Plastic surfaces (e.g. culturing flasks, multi-well plates, and transwell inserts) were coated with 0.01 mg/mL rat tail collagen (Sigma-Aldrich #C7661) diluted in

60% EtOH for 1 h at 37 °C. Collagen was removed and surfaces were washed 2× in PBS prior to seeding cells. Glass surfaces (including glass coverslips and 8-well chamber slide (Ibidi #80827-90)) were coated with 0.04 mg/mL human collagen (Sigma-Aldrich #C5533) diluted in water for 1 h at 37 °C. Collagen was removed and surfaces were washed 2× in PBS prior to seeding cells. All cells were kept in a constant humid atmosphere at 37 °C, 5% $CO_2$ and 21% oxygen. For splitting, 0.25% Trypsin-EDTA (Gibco #25200056) was used for T84 and Calu-3 cells, and 0.05% Trypsin-EDTA (Gibco #25200054) was used for HEK293T.

## Generation of T84 knock-out cell lines

IFNAR1 KO, IFNLR KO, and IFNAR/IFNLR double knock-out (dKO) (Pervolaraki et al, 2017) and IRF3 KO (Triana et al, 2021) cell lines were previously generated in our laboratory using CRISPR-Cas9 gene editing approach. The T84 IFN-λ2/3 KO cell line was generated using a lentivirus-based CRISPR-Cas9 gene editing approach. Briefly, single-guide RNAs (sgRNAs) targeting the coding region of IFNλ2 and IFNλ3 were inserted at BamHI cloning site in the lentiviral vector lentiCRISPR v2 (Addgene #52961) containing a blasticidin resistance gene. The following sgRNAs were used: fw: 5′-**CACC**GTGGGG ACTGCACGCCAGTGC-3′, rev: 5′-**AAAC**GCACTGGCGTGCAGT CCCCAC-3′ for IFNλ2, and fw: 5′-**CACC**GCTGGAGCAGTTCCTG TCGCC-3′, rev: 5′-**AAAC**GGCGACAGGAACTGCTCCAGC-3′ for IFNλ3. To generate the lentiviruses, HEK293T cells seeded at 80% confluency in a 10 cm² dish were transfected using the transfection reagent Polyethylenimine (PEI) (Polysciences #23966-100) at a PEI:DNA ratio of 4:1. 8 µg of the lentiCRISPR v2 containing the sgRNA targeting IFNλ2 or IFNλ3, 4 µg pMDG.2 plasmid (Addgene #12259) and 4 µg psPAX (Addgene #12260) plasmid were used for the transfection of each 10-cm² dish. Three days post transfection, the supernatant was collected, spun down at 4000 rcf for 10 min, and filtered through a 0.45 µm syringe filter (Lab Unlimited #W10462100). To pellet the lentiviruses, the supernatant was spun down at 125,000 rcf for 1:30 h using a SW40 Ti rotor. The lentivirus pellet was resuspended in 100 µL of OptiMem (Gibco #31985062) (per yield of one 10-cm² dish). For the lentiviral transduction, $3 \times 10^5$ T84 WT cells were seeded per well of a six-well plate. 16 h post-seeding, cells were transduced with 20 µL of the concentrated lentiviruses supplemented with 3 µL Polybrene infection/transfection reagent (Sigma-Aldrich #TR-1003-G) diluted in 3 mL of DMEM:F12 media. Three days post-transduction, transduced cells were selected with Blasticidin at 0.1 mg/mL (Invivogen #ant-bl-1). Single cell cloning was performed using a limited serial dilution approach in 96-well plate. KO of IFNλ2 and IFNλ3 genes was confirmed by Sanger sequencing (Appendix Fig. 3).

## Generation of T84 knock-down cell lines

T84 knockdown cells were generated using a lentivirus shRNA knockdown approach. Briefly, short hairpin RNAs (shRNAs) targeting STING and YAP/TAZ were inserted into the lentiviral vector pLKO.1 puro(Addgene #8453) containing a puromycin resistance gene; shRNAs targeting claudin-2 were inserted into the lentiviral vector pLKO.1 neo (Addgene #13425) containing a neomycin resistance gene. The following shRNA sequences were used: STING 5′-CAGAGCTATTTCCTTCCACA-3′; YAP/TAZ (shRNA#1) 5′-GCCACCAAGCTAGATAAAGAA-3′, (shRNA #2)

5′-GCGTTCTTGTGACAGATTATA-3′; claudin-2 (shRNA #1) 5′-GTGCCTGACAGCATGAAATTT-3′, (shRNA #2) 5′-GCTCTTT ACTTGGGCATTATT-3′. To generate the lentiviruses, HEK 293T cells seeded at 80% confluency in a 10 cm² dish. Next day, media was replenished with 10 mL antibiotics-free media, and cells were transfected using the transfection reagent PEI at a PEI:DNA ratio of 4:1. 8 μg of the pLKO.1 plasmid containing shRNA, 4 μg pMDG.2 plasmid and 4 μg psPAX plasmid were used for the transfection of each 10 cm² dish. Next day, the media was replenished with 10 mL culture media. Three days post transfection, the supernatant was collected, spun down at 4000 rcf for 10 min, and filtered through a 0.45 μm syringe filter (Lab Unlimited #W10462100).

For YAP/TAZ and STING KD, lentiviruses were concentrated by spinning down at 125,000 rcf for 1:30 h using a SW40 Ti rotor. The lentivirus pellet was resuspended in 100 μL of OptiMem (Gibco #31985062) (per yield of one 10 cm² dish). For the lentiviral transduction, $3 \times 10^5$ T84 WT cells were seeded per well of a six-well plate. Sixteen hours post-seeding, cells were transduced with 20 μL of the concentrated lentiviruses supplemented with 3 μL Polybrene infection/transfection reagent diluted in 3 mL of DMEM:F12 media. Three days post-transduction, transduced cells were selected with 10 μg/mL puromycin (Invivogen #ant-pr-1).

For CLDN2 KD, lentiviruses were not concentrated. T84 WT and IFN-λ2/3 KO cells were seeded at a density of $1.5 \times 10^5$ cells/cm² into transwell inserts (Thermo Fisher #3415). One day post-seeding, 80 μL of lentiviruses were diluted in 720 μL of DMEM:F12 media in the presence of 0.8 μL Polybrene infection/transfection reagent. For transduction, 200 μL of the mixture were added to the top compartment of the transwell and 600 μL to the bottom compartment. Three days post-transduction, transduced cells were selected using 50 μg/ml Neomycin (Corning, 30-234-CI).

## Cell seeding at different cellular density

Seeding on plastic or glass surfaces: for high cell density, $2.2 \times 10^5$ cells/cm² were seeded. One-day post-seeding, medium was exchanged with fresh medium, and 2 days post-seeding, cells were treated or harvested for the experiments. For low cell density, $2.7 \times 10^4$ cells/cm² were seeded. One-day post-seeding cells were treated or harvested for the experiments. For the seeding on transwell inserts: $1.5 \times 10^5$ cells in 200 μL medium were seeded in the top compartment of the transwell insert, and 600 μL media was added to the bottom compartment of the transwell. Media was exchanged every second day with fresh media.

## Organoid culture and seeding

Human ileum-derived organoids were received in accordance with the recommendations of the University Hospital Heidelberg with written informed consent from all subjects in accordance with the Declaration of Helsinki. All samples were received and maintained in an anonymized manner. The protocol was approved by the "Ethics Commission of the University Hospital Heidelberg" under the protocol S-443/2017. Organoids were grown as three-dimensional structures in Matrigel (Corning #354230) and maintained at 37 °C, 5% $CO_2$ in basal or differentiation media (compounds specified in Appendix Table S1). Organoid culturing and passaging were previously described (Stanifer et al, 2020).

Briefly, media changes were performed every second day and organoids were passaged weekly using 0.05% trypsin-EDTA. Immediately after each split, basal media was supplemented with 10 μM Y27632 hydrochloride (Cayman Chemical #10005583).

To seed ileum organoids at different cellular densities, three-dimensional organoids were digested with 0.05% Trypsin-EDTA (Gibco #25200054) for 5 min and seeded in basal media into 48-well plates coated with 0.04 mg/mL human collagen (Sigma-Aldrich, C5533). To obtain different densities, different amounts of wells (one or four wells containing three-dimensional organoids in Matrigel per new well) were pooled and seeded. 1-day post-seeding, media was exchanged for differentiation media and on day 2 organoids were harvested for qRT-PCR.

To explore tight junction formation in organoids after STING inhibition, organoids were trypsinized with 0.05% Trypsin-EDTA for 5 min. Ileum-derived organoids were seeded into eight-well glass chamber slides (Ibidi #80827-90) coated with 0.04 mg/mL human collagen (Sigma-Aldrich, C5533). On the next day, organoids were treated with DMSO or H151 at the indicated concentrations. Three days post-treatment, indirect immunofluorescence staining was performed as described below.

## TEER measurement and FITC-Dextran permeability assay

Establishment of barrier function by epithelial cells was assessed by both measurement of the Transepithelial electrical resistance (TEER) and by FITC-Dextran permeability assay. TEER measurement was performed daily starting at day one post-seeding using the EVOM3 Epithelial Volt/Ohm Meter with STX2-PLUS (Word Precision Instruments). A measurement of 1000 $\Omega/cm^2$ and above indicates that T84 cells have established a barrier function. FITC-Dextran permeability assay was performed to assess the capacity of a cell monolayer to restrict the paracellular diffusion of small molecules. Cells were seeded on transwell inserts. At indicated days post-seeding, media was removed from the apical (top) compartment of the transwell and replaced by 200 μL of fresh medium containing 2 mg/mL fluorescein isothiocyanate (FITC)-labelled dextran (4 kDa) (Sigma-Aldrich #46944). As a negative control, culture media alone was used. For the positive control, 200 μL of 2 mg/mL of FITC-Dextran was added to the apical side of a cell-free collagen coated insert and 600 μL culture media were added to the basolateral compartment. Cells and controls were incubated for 3 h at 37 °C and media was collected from the basal compartment. Fluorescent signal was measured using an 800TS Microplate Reader (BioTek) at an excitation wavelength of 495 nm. A standard curve by serial dilution of the FITC-Dextran in culture media was done to assess the basolateral FITC-Dextran concentration.

## RNA isolation, cDNA synthesis, and qRT-PCR

RNA was isolated using the RNAeasy RNA extraction kit (Qiagen #74136), DNA was synthesized using iSCRIPT reverse transcriptase (Bio-Rad #1708890), and quantitative RT-PCR (qRT-PCR) assay was performed using iTaq SYBR green (Bio-Rad #1725124) according to the manufacturer's instructions. qRT-PCRs were performed using the Real-Time PCR System CFX Opus 96 (Bio-Rad #12011319). The expression of target gene was normalized to the housekeeping gene TaTa box binding protein (TBP). Primer sequences are listed in Appendix Table S2.

## Nuclear and cytoplasmic fractionation of cells for SDS-PAGE and Western Blot analysis

Cells were seeded into a 12-well plate at either high or low density as described above depending on the experiments. Cells were harvested with trypsin, washed with cold PBS and resuspended in 1 mL cold PBS. For the total lysate fraction (TL-fraction), 200 μL was removed, spun down at 2000 rpm for 5 min at 4 °C. The cell pellet was lysed in 60 μL RIPA buffer containing protease and phosphatase inhibitors for 20 min on ice, followed by centrifugation at 13,000 rpm for 10 min at 4 °C. The supernatant corresponds to the total lysate fraction. RIPA buffer composition: 1× RIPA buffer (150 mM sodium chloride, 1.0% Triton X-100, 0.5% sodium deoxycholate, 0.1% sodium dodecyl sulfate (SDS), 50 mM Tris, pH 8.0) supplemented with the protease inhibitors cOmplete™ Mini EDTA-free Protease Inhibitor Cocktail (Sigma-Aldrich #11836170001) and phosphatase inhibitor PhosSTOP (Millipore Sigma #PHOSS-RO). The remaining 800 μL cell suspension was spun down at 2000 rpm for 5 min at 4 °C, and the pellet was resuspended in 100 μL cytosolic extraction buffer (0.1% NP40 in PBS containing the protease inhibitors cOmplete™ Mini EDTA-free Protease Inhibitor Cocktail and phosphatase inhibitor PhosSTOP). After 2 min incubation on ice, the resuspended pellet was centrifuged at 6500 rpm for 1 min at 4 °C. The supernatant was stored as the cytosolic fraction (C-fraction). The pellet was then washed twice with 500 μL cytosolic extraction buffer and centrifuged at 2000 rpm for 5 min at 4 °C. The pellet was resuspended in 40 μL RIPA buffer containing protease and phosphatase inhibitors and centrifuged at 13,000 rpm for 10 min at 4 °C. The supernatant corresponds to the Nuclear fraction (N-fraction).

## SDS-PAGE and western blot

Adherent cells were lysed with 1× RIPA buffer with cOmplete™ Mini EDTA-free Protease Inhibitor Cocktail and phosphatase inhibitor PhosSTOP for 5 min at 37 °C. Lysates were collected and protein concentration was measured using the Pierce BCA Protein Assay Kit assay (Thermo Scientific #23225) according to the manufacturer's instructions. In total, 5–10 μg protein per condition were separated by SDS-PAGE and blotted onto a 0.2 μm nitrocellulose membrane (Bio-Rad, #1704158) using a Trans-Blot® Turbo™ Transfer System (Bio-Rad). Membranes were blocked with Tris Buffer saline (TBS)-Tween (0.5% Tween in TBS) containing 5% bovine serum albumin (BSA) or containing 50% Intercept® (TBS) Blocking Buffer (Licor #927-60001) for 1–2 h at room temperature (RT). Membranes were incubated with primary antibodies diluted 1:1000 against alpha-Tubulin (Sigma #T9026), β-actin (Sigma #A5441), Lamin-B1 (Santa Cruz #sc-374015), IRF3 (Cell Signaling #11904 T), phospho-IRF3 (NEB/Cell Signaling #4947), ISG15 (Santa Cruz #166755), Claudin-2 (Invitrogen #51-6100), ZO-1 (Invitrogen #33-9100 and #40-2200), STAT1 (BD Biosciences #610115), phospho-STAT1 (BD Biosciences #612233), STING (Cell Signaling #13647), TBK1/NAK (Cell Signaling #3013S), phospho-TBK1/NAK (Cell Signaling #5483S), YAP/TAZ (Santa Cruz #sc-101199 and Cell Signaling #8418), phospho-YAP (Cell Signaling #4911S) diluted in blocking buffer overnight at 4 °C. Anti-mouse-IgG (Abcam #ab6789) and anti-rabbit-IgG (Abcam #ab97051) antibodies coupled with horseradish peroxidase (HRP) (GE Healthcare #NA934V) or IRDye® 680RD/800CW (Licor 926-68073/ 926-32210) were used as secondary antibodies. Membranes were washed three times with TBS-T for 5 min at RT after each step. The Pierce ECL Western Blotting Substrate (Thermo Fisher #32209) was used for detection of HRP-conjugated antibodies according to manufacturer instructions. The membranes were imaged with the Image-Quant™ LAS 4000 (GE Healthcare) or Odyssey M imaging system (Licor). Quantification was done using the open image analysis software ImageJ. Relative abundance of target protein was normalized to the loading control housekeeping protein.

## Indirect immunofluorescence staining

Cells and ileum-derived organoids were washed with PBS and fixed in 2% Paraformaldehyde (PFA) (Roth #0335.3) (diluted in PBS) for 20 min at RT for all antibodies or MeOH for 7 min at −20 °C for ZO-1 (Invitrogen #33-9100) antibody. Cells and organoids were washed in PBS three times and permeabilized in 0.5% Triton-X-100 (Sigma-Aldrich #X-100-500ML) diluted in PBS for 15 mins at RT. Cells were blocked using 3% Bovine Serum Albumin (BSA) (Sigma-Aldrich #A3294) for all antibodies or 10% FBS (Sigma-Aldrich #12306 C) in PBS for 30 min at RT for ZO-1 (Invitrogen #33-9100) antibody. Primary antibodies against ZO-1 (Invitrogen #33-9100) diluted in 10% FBS (in PBS), YAP/TAZ (Santa Cruz #sc-101199) were diluted in 1% BSA (in PBS) and incubated for 1 h at RT. Cells and organoids were incubated with Alexa Fluor® conjugated secondary antibodies with DAPI (Invitrogen #P36941), both diluted in 1% BSA in PBS for 1 h at RT. Cells were washed in PBS three times after each step.

## Fluorescence imaging and image analysis

Live cell imaging and images of fixed samples were acquired using the epifluorescent ZEISS Celldiscoverer-7 (CD7) Widefield microscope. For live cell fluorescence microscopy, cells were kept at 37 °C and 5% $CO_2$ during the whole procedure. Assessment of basal expression of ISG using live cell microscopy: T84 pMx1-mCherry cells were seeded at low density on glass bottom 8-well chamber slides (Ibidi #80827-90). Cells were imaged every 3 h for a period of 10 days. Expression of mCherry was performed as follows: ilastik 1.2.0 was used for masking individual cells from the Brightfield images. CellProfiler 3.1.9 was used to measure the mean fluorescence intensity (MFI) within the mask and the area of the mask (area occupied by the cells). The mean fluorescence intensity was then plotted against time and the area occupied by the cells.

## Genomic DNA isolation, PCR and gel extraction

T84 WT and IFN-λ2/3 KO cells were harvested, and genomic DNA (gDNA) was isolated by using Monarch® Genomic DNA Purification Kits (NEB #T3010S) according to manufacturers' protocol. To amplify the genomic loci of IFNλ2 and IFNλ3, PCR amplification was performed by using 5× Phusion HF Buffer (Thermo Fisher #F-549S) and Phusion Hot Start II DNA Poly (Thermo Fisher #F518L) with following primers: IFN-λ2/3 forward: 5′-CTAGGTGAGTCC-CACATCTCTGTCCGTGCTCAG-3′, IFNλ2 reverse: 5′-CCTGGA GGTGAGTTGGATTTACACACAC-3′, IFNλ3 reverse: 5′-GCGAC TGGGTGACAATAAATTAAGCCAAGTGGC-3′. The PCR products were subjected to electrophoresis on 1% Agarose (Sigma-Aldrich #A6013) in 1X TBE (1.1 M Tris-base, 900 mM Boric Acid, 25 mM EDTA). Specific amplicon bands (IFNλ2: 1555 bp, IFNλ3:

1516 bp) were extracted using the Monarch DNA Gel Extraction Kit (NEB #T1020S). Extracted DNAs were sequenced using Sanger sequencing (GENEWIZ, Azenta Life Sciences).

## Mitochondrial DNA depletion

Mitochondrial DNA (mtDNA) was depleted either using Ethidium bromide (EtBr) (protocol adjusted from King and Attardi, 1989) or using 2′-3′-dideoxycytidine (ddC) (protocol adjusted from Rongvaux et al, 2014 (White et al, 2014)). Cells were incubated with 300 ng/mL EtBr (MP Biomedicals #ETBC100) or 100 µg/mL ddC (Sigma-Aldrich #D5782) for 6 days. For the depletion of mtDNA at high cellular density: $2.7 \times 10^4$ cells/cm$^2$ cells were seeded with the media supplemented with EtBr or ddC. Media was replenished every 2 days starting 1 day post-seeding, and cells were harvested at 6th day post-seeding, when they formed a confluent monolayer. For the depletion of mtDNA at low cellular density: cells were grown in media supplemented with EtBr or ddC as described above. At day 5 post-seeding, cells were split and re-seeded at low density ($2.7 \times 10^4$ cells/cm$^2$) in medium supplemented with EtBr or ddC. One day after re-seeding (equivalent to 6 days of treatment), cells were harvested for further processing.

For assessment of mitochondrial to nuclear DNA ratio in the cells: Cells were washed, and total DNA was purified from using the PureLink™ Genomic DNA Mini Kit (Thermo Fisher #K182002) following the manufacturer's instructions. 50 ng total DNA was used for qRT-PCR analysis using the iTaq SYBR green (Bio-Rad #1725124) according to the manufacturer's instructions. qRT-PCRs were performed using the Real-Time PCR System CFX Opus 96 (Bio-Rad #12011319). Primers targeting mitochondrial gene nd2 and nuclear gene S18 are described in Appendix Table S2. The amount of mitochondrial DNA was determined relative to the amount of nuclear DNA ($2^{-Cq}$[nd2]/$2^{-Cq}$[18S]).

## Cellular fractioning to assess total and cytoplasmic mitochondrial DNA

Cells were seeded into a 12-well plate at either high or low density as described above. Cells were harvested with trypsin, washed with cold PBS, and resuspended in 1 mL cold PBS. For assessment of total mitochondrial DNA in the cell: 200 µL was removed, spun down at 2000 rpm for 5 min at 4 °C. Cells were washed, and total DNA was purified using the PureLink™ Genomic DNA Mini Kit following the manufacturer's instructions. For assessment of cytosolic mitochondrial DNA: The remaining 800 µL cell suspension was spun down at 2000 rpm for 5 min at 4 °C, and the pellet was resuspended in 100 µL cytosolic extraction buffer (0.1% NP40 in PBS containing the protease inhibitors cOmplete™ Mini EDTA-free Protease Inhibitor Cocktail and phosphatase inhibitor PhosSTOP). After 2 min incubation on ice, the resuspended pellet was centrifuged at 6500 rpm for 1 min at 4 °C. This spinning step pellets cellular organelles including mitochondria to ensure that only leaked cytosolic mtDNA are evaluated. The supernatant corresponds to the cytosolic fraction and was purified using the PureLink™ Genomic DNA Mini Kit following the manufacturer's instructions. All steps are performed on ice or at 4 °C. After DNA extraction, 50 ng total DNA was used for qRT-PCR analysis using the iTaq SYBR green according to the manufacturer's instructions. qRT-PCRs were performed using the Real-Time PCR System CFX Opus 96. Only primers targeting mitochondrial gene nd2 were used and are

described in Appendix Table S2. The amount of cytoplasmic mitochondrial DNA was relativized to the amount of total mitochondrial DNA (cytoplasmic(2-Cq[nd2])/nuclear(2-Cq[nd2])) for each condition and compared between high and low density.

## Cell growth assay

Cells were seeded into a 12-well plate at a density of $3.6 \times 10^4$ cells per well. The following day, the media was replaced. Every second day, the cells were detached using trypsin, and cell suspension was diluted 1:1 in a 0.4% Trypan Blue Solution (Gibco #15250061). The live cell count was determined using a TC20 Automated Cell Counter (Bio-Rad).

## Cytotoxicity assay

Cytotoxicity was assessed using the LDH-Glo™ Cytotoxicity Assay (Promega #J2380) following the manufacturer's protocol. Cells were seeded in 96-well plates and treated with 100 µL inhibitors at indicated concentrations. PPMP (50 µM; D-threo-1-Phenyl-2-hexadecanoylamino-3-morpholino-1-propanol hydrochloride) (Cayman Chemicals #17236) was included as a positive control for cytotoxicity. To quantify extracellular (released) LDH, 50 µL of culture supernatant was transferred to a new clear, flat-bottom 96-well plate at the indicated time point. To assess intracellular LDH, 15 µL of 10×Lysis Buffer was added to the remaining cells in the original wells and incubated for 1 h at 37 °C. Then, 50 µL of the resulting cell lysate was transferred to a separate clear 96-well plate. For both supernatant and lysate samples, 50 µL of LDH Substrate Mix was added to each well, mixed gently, and incubated for 30 min at room temperature in the dark. The reaction was stopped by adding 50 µL of Stop Solution per well. Luminescence was measured at 490 nm using an 800TS Microplate Reader (BioTek). The percentage of cytotoxicity was calculated by normalizing released LDH to total intracellular LDH.

## Bulk RNA-sequencing

T84 WT, IFN-λ2/3 KO, and IFNLR KO cells were seeded at high and low cellular density in a 48-well plate as described above. For each condition, three biological replicates were prepared. For high cell density, three wells were pooled; for low cell density eight wells were pooled for each biological replicate. RNA extraction was performed using the RNeasy Plus Mini Kit following the manufacturer's instruction. RNA sequencing was performed by GENEWIZ (Azenta Life Sciences). Briefly, cDNA libraries were prepared using the TruSeq Stranded mRNA Library Prep Kit (Illumina) and sequenced on an Illumina NovaSeq 6000 platform, generating 150 bp paired-end reads. Data preprocessing together with quality control report (FAST QC Report) was provided by Genewiz. The quality control of raw sequencing reads was checked using FastQC to ensure high-quality sequencing data. Low-quality reads and adapter sequences were trimmed using Trimmomatic. The cleaned reads were then aligned to the human reference genome (GRCh38) using STAR aligner. Gene expression levels were quantified using featureCounts, and differential gene expression analysis was conducted with DESeq2. Genes with an adjusted $P$ value < 0.05 and a fold change ≥2 were considered significantly differentially expressed. Gene Ontology (GO) enrichment analysis

was performed on the differentially expressed genes using the clusterProfiler package in R.

## Data plotting and statistics

If not specified otherwise, data plotting and all statistical analyses were performed with the GraphPad Prism 5.0 software. The number of biological replicates and statistical tests used are specified in the figure legends. Statistical tests are listed in figure legends. If not specified otherwise, all schematics and illustrations were created with BioRender.com. All figures were assembled with the Affinity Designer 1.10.0 software.

## Data availability

The RNA-seq data generated in this study have been deposited in the NCBI Gene Expression Omnibus (GEO) under accession number GSE279172.

The source data of this paper are collected in the following database record: biostudies:S-SCDT-10_1038-S44318-025-00539-5.

## Peer review information

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

## Acknowledgements

SB and MS are supported by the UF College of Medicine start-up package funds. Research reported in this publication was supported by the National Institute of Allergy and Infectious Diseases (NIAID) of the National Institutes of Health under award number 1R01AI185510 and 1R01AI189780. Additionally, this project was supported by the Deutsche Forschungsgemeinschaft (DFG) by the projects SFB 1129 (DFG 240245660). PD was supported by Deutscher Akademischer Austauschdienst (DAAD). ZU is supported by Walter-Benjamin funding by the German Research Foundation (Deutsche Forschungsgemeinschaft, DFG, project number 521582111). We would like to thank the Boulant and Stanifer lab members for the constructive discussions and for proofreading this manuscript.

## Author contributions

**Yagmur Keser**: Conceptualization; Data curation; Formal analysis; Validation; Visualization; Methodology; Writing—original draft; Writing—review and editing. **Camila Metz-Zumaran**: Conceptualization; Data curation; Formal analysis; Visualization; Methodology; Writing—original draft. **Zina M Uckeley**: Data curation; Formal analysis. **Dorothee Reuss**: Data curation. **Patricio Doldan**: Data curation. **James M Ramsden**: Data curation. **Megan L Stanifer**: Conceptualization; Data curation; Supervision; Funding acquisition; Visualization; Methodology; Writing—original draft; Writing—review and editing. **Steeve Boulant**: Conceptualization; Supervision; Funding acquisition; Writing—original draft; Project administration; Writing—review and editing.

Source data underlying figure panels in this paper may have individual authorship assigned. Where available, figure panel/source data authorship is listed in the following database record: biostudies:S-SCDT-10_1038-S44318-025-00539-5.

## Disclosure and competing interests statement
The authors declare no competing interests.

# Expanded View Figures

**Figure EV1. Cell density regulates the basal immune response in human intestinal and airway epithelial cells.**

(A) T84 cells seeded on transwells were harvested at indicated days post-seeding and the expression of the ISGs Mx1, IFIT1, ISG15 and Viperin were assessed by qRT-PCR. (B–D) T84 pMx1-mCherry cells were seeded at low cellular density. One-day post-seeding cells were imaged over a period of ten days to visualize the expression of the mCherry reporter whose expression is under the control of the ISG Mx1 promoter region. (B) Representative image, Scale bar = 50 μm. (C) Quantification of the pMx1-mCherry mean fluorescence intensity over time. (D) Positive correlation between cell density and expression of the pMx1-mCherry reporter. Linear regression and coefficient of correlation (r) were calculated using the mean fluorescence intensity from (C) and the area occupied by cells. (E, F) T84 cells were seeded at high and low cellular density. (E) Expression of the ISGs Mx1, IFIT1 and Viperin was addressed using qRT-PCR. The relative expression was normalized to TBP. $n \geq 3$ biological replicates. (F) ISG15 expression was assessed by Western Blot. ISG15 abundance was quantified relative to the loading control actin. (G) The expression of IFN-λ2/3 and of the ISGs Mx1, IFIT1 and Viperin was addressed by qRT-PCR in Calu-3 cells seeded at high and low cellular densities. The relative expression was normalized to TBP. $n \geq 3$ biological replicates. Statistical analysis was performed using ordinary one-way ANOVA using the cells at one day post-seeding as a reference (A, C) and unpaired $t$ test between high and low density (E, G). n.s. indicates non-significant results ($P > 0.05$). Exact $P$ values are shown on the plots when significant; otherwise, results are not significant. Error bars represent standard deviation with the mean as the center. Source data are available online for this figure.

                                                        

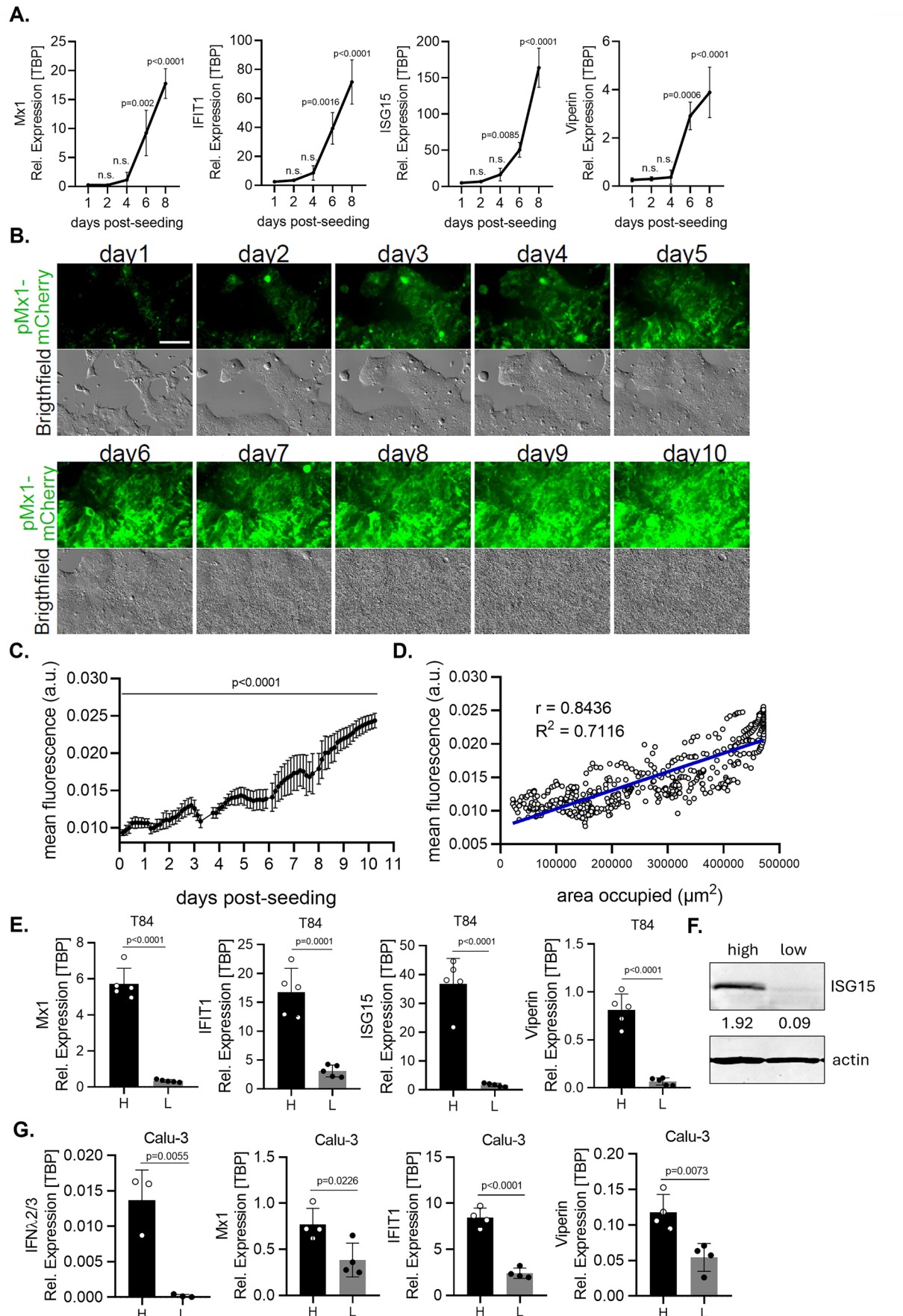

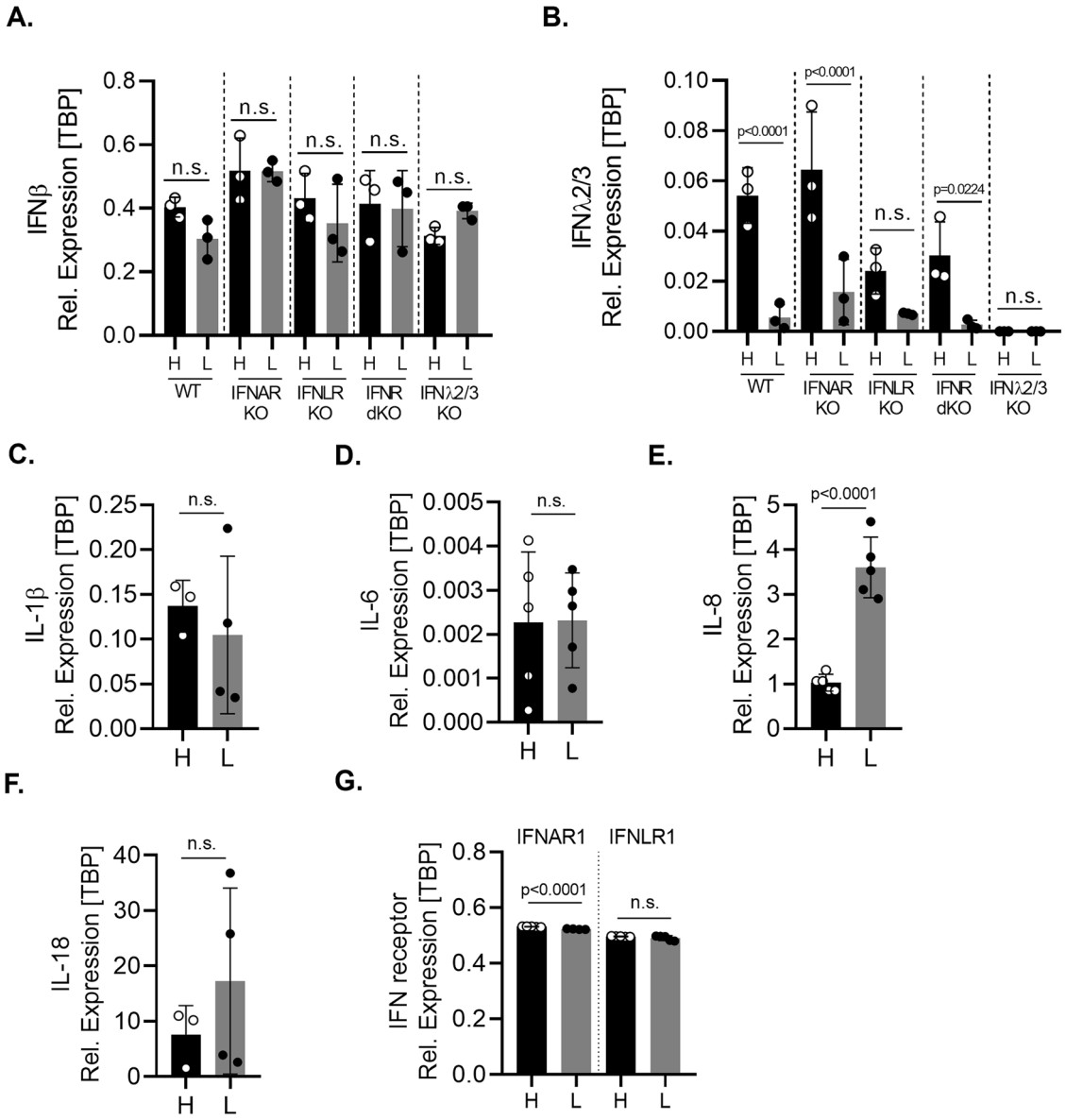

**Figure EV2. Basal expression of type I IFNs and anti- and pro-inflammatory cytokines are not dependent on cell confluency.**

T84 WT and T84 knock-out (KO) cells were seeded at high and low cellular density and the expression of the (**A**) IFNβ and (**B**) IFN-λ2/3 was addressed using qRT-PCR. T84 WT cells were seeded at high and low cellular densities and the relative expression of the cytokines IFNβ, IL-1β, IL-6, IL-18 and IL-8 was addressed using qRT-PCR. (**C**) IL-1β, (**D**) IL-6, (**E**) IL-8 and (**F**) IL-18. (**G**) The expression level of the type I IFN and type III IFN receptors IFNAR and IFNLR was assessed by qRT-PCR analysis of T84 cells grown at high and low cellular density. Relative expression normalized to TBP. $n \geq 3$ biological replicates. Statistical analysis was performed using unpaired *t* test between high and low density. n.s. indicates non-significant results ($P > 0.05$). Exact *P* values are shown on the plots when significant; otherwise, results are not significant. Error bars represent standard deviation with the mean as the center. Source data are available online for this figure.

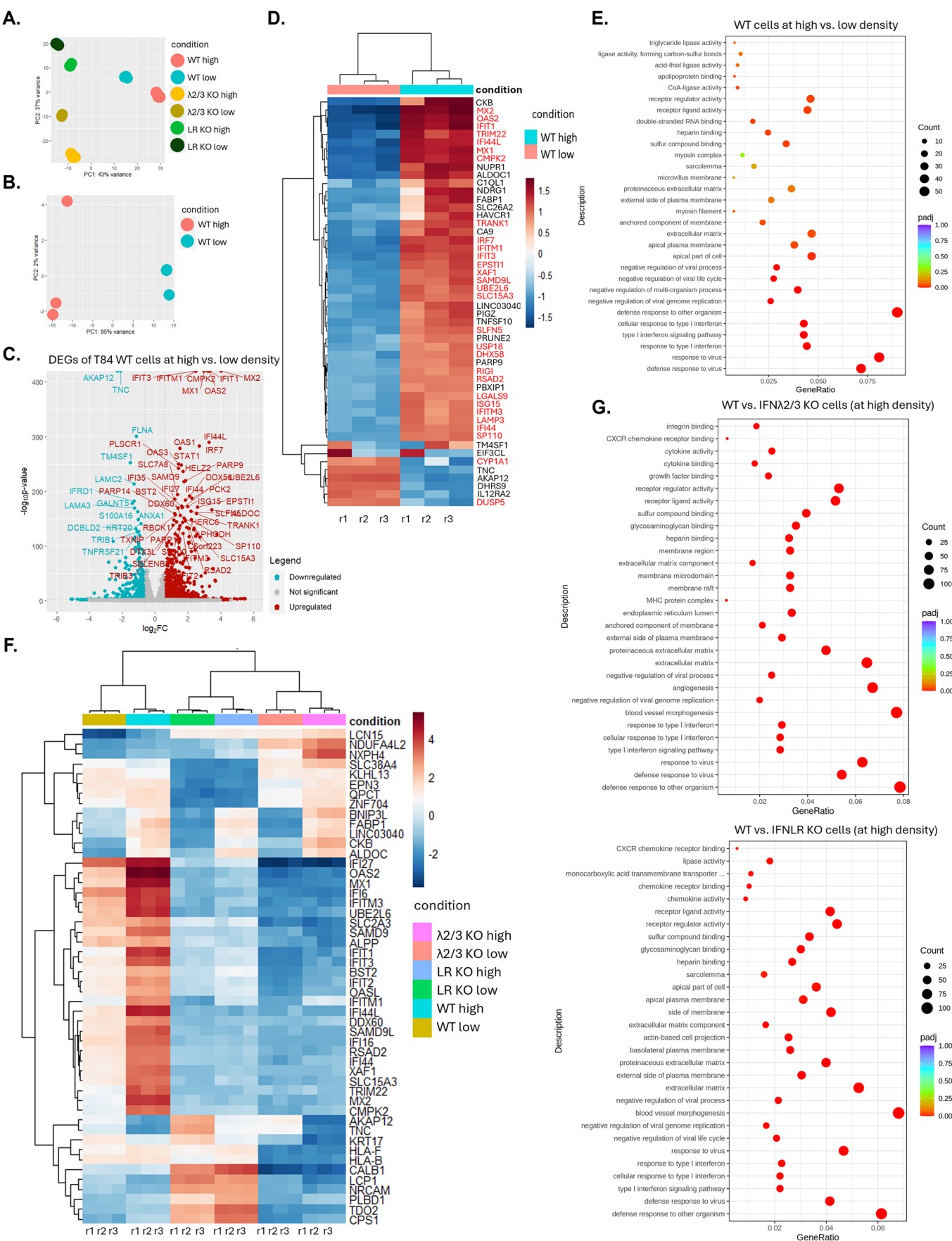

**Figure EV3.  RNA-sequencing revealing the role of basal IFN-λ2/3 signaling is beyond its antiviral function.**

T84 WT, IFN-λ2/3 KO and IFNLR KO cells were seeded at high and low density and subjected to RNA sequencing. (A, B) The PCA plot displays the distribution of (A) T84 WT, IFN-λ2/3 KO and IFNLR KO cells at high and low cell density and (B) only T84 WT cells at high and low density based on their gene expression profiles. Each point represents an individual sample, colored according to the experimental group. (C) The volcano plot illustrates the differential expression analysis results between T84 WT cells at high and low cell density. Each point represents a gene, plotted by its fold-change (x axis) and statistical significance (−log10 P value, y axis). Genes with significant differential expression (P < 0.05) are highlighted in red (upregulated) and blue (downregulated). Key genes of interest are labeled. (D) Heatmap displaying the expression levels of the top 50 differentially expressed genes across T84 WT cells at high and low density. ISGs were highlighted in red. (E) The enrichment of GO terms associated with the differentially expressed genes between T84 WT cells at high and low density. (F) Heatmap displaying the expression levels of the top 50 differentially expressed genes across T84 WT, IFN-λ2/3 KO and IFNLR KO cells at high and low density. (G) The enrichment of GO terms associated with the differentially expressed genes between (upper panel) T84 WT and IFN-λ2/3 and (lower panel) T84 WT and IFNLR KO cells at high density. (D, F) Rows represent genes, and columns represent samples, with hierarchical clustering applied to both dimensions. The color scale indicates relative expression levels, with red representing high expression and blue representing low expression. Clusters of co-expressed genes and samples are clearly visible, indicating distinct transcriptional signatures associated with the treatment. (E, G) GSEA plots were generated using a Kolmogorov–Smirnov-like statistic, as previously described (Subramanian et al, 2005).

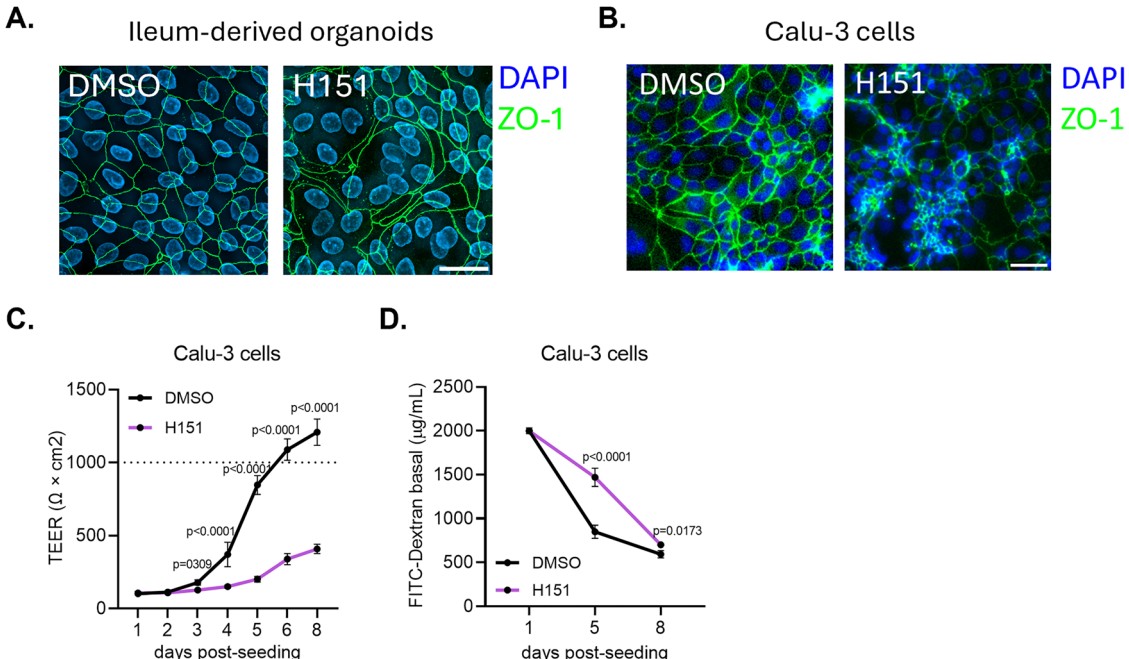

**Figure EV4. Inhibition of STING impairs barrier formation in human primary intestinal epithelial cells and in human airway epithelial cells.**

(A) Human ileum-derived organoids were mock-treated or treated with the STING inhibitor H151. Tight junctions were immunostained using an anti-ZO-1 antibody (green). Nuclei were stained using DAPI (blue). Scale bar = 25 μm. (B) Same as (A) but in the airway epithelial cells Calu-3 cells. Scale bar = 50 μm. (C) TEER measurement and (D) FITC-Dextran permeability assay for Calu-3 cells mock-treated or treated with the STING inhibitor H151. (C) Values > 1000 Ω/cm² (dotted line) shows that cells established barrier function. $n \geq 3$ biological replicates. Statistical analysis was performed using two-way ANOVA. n.s. indicates non-significant results ($P > 0.05$). Exact $P$ values are shown on the plots when significant; otherwise, results are not significant. Error bars represent standard deviation with the mean as the center. Source data are available online for this figure.

