## [Peer Review File · The EMBO Journal]

Basal IFN- λ 2/3 expression mediates tight junction formation in human epithelial cells

Yagmur Keser, Camila Metz-Zumaran, Zina Uckeley, Dorothee Reuss, Patricio Doldan, James Ramsden, Megan Stanifer, and Steeve Boulant

Corresponding author(s): Steeve Boulant (s.boulant@ufl.edu) , Megan Stanifer (m.stanifer@ufl.edu)

Review Timeline:

Submission Date:	3rd Feb 25
Editorial Decision:	7th Feb 25
Appeal Received:	12th Feb 25
Editorial Decision:	14th Mar 25
Revision Received:	5th Jun 25
Editorial Decision:	4th Jul 25
Revision Received:	8th Jul 25
Accepted:	25th Jul 25

Editor: Ioannis Papaioannou

Transaction Report:

Dear Steeve,

Thank you again for submitting your manuscript (EMBOJ-2025-120363) to The EMBO Journal. I have now read it in detail, looked at the relevant literature, and also discussed the findings reported in your manuscript with the other members of our editorial team. I am sorry to say that we cannot pursue publication of the manuscript in The EMBO Journal, but I would like to encourage you to consider transferring it to our sister journal Life Science Alliance, for which it would be an excellent candidate.

In this manuscript you show basal type-III IFN λ expression in epithelial cells to correlate with the formation of confluent epithelial cell monolayers. You demonstrate that this basal expression is driven by cGAS-STING signaling, which is further triggered by cytosolic mitochondrial DNA. Your results further provide both regulatory and mechanistic insights: you provide evidence indicating that the Hippo signaling inhibits cGAS-STING and suppresses basal IFN λ expression at low cellular levels, and that the reported STING-dependent basal IFN λ 2/3 signaling regulates barrier formation through the tight junction network by regulating claudin-2 expression. We recognize that these findings will be interesting for the interferon signaling and epithelial integrity communities.

Although we appreciate the novelty of some of the observations and the reported mechanism for IFN λ , we also note that there are several precedents in the literature that limit, to some extent, the conceptual novelty provided by the study, in our view. In particular, type-III IFNs are already known to be expressed at basal levels in epithelial cells. In addition, they have already been linked to epithelial integrity and barrier function in the context of microbial infection, and an earlier study reported that IFN λ signaling modulates tight junction protein localization, independently of STAT1, in the blood-brain barrier. Furthermore, a recent study uncovered another function of IFN λ in tissue repair of intestinal mucosa in contexts other than microbial infection - after inflammatory or DNA-related damage, in particular. Moreover, other interferons have previously been reported to down-regulate claudin-1 and claudin-2 expression, thereby impairing the epithelial barrier function in cultured cells. Apart from the above lines of previously reported evidence, mitochondrial DNA is a well-known relevant trigger of cGAS-STING signaling in different contexts, and STING signaling itself is already known to drive type-I IFN basal production. We also note that the study relies on cell culture experiments, which leaves the question of the *in vivo* relevance of the findings open, while there are also some unaddressed mechanistic questions, such as the exact molecular mechanism by which IFN λ 2/3 signaling down-regulates claudin-2 expression.

Taking into account the above considerations, I regret to say that we did not find the manuscript a suitable candidate for The EMBO Journal, and we have therefore decided not to proceed with in-depth peer review at our journal. Please be assured that this decision does not reflect a judgment of the relevance of the topic or the quality of the investigation; it is only based on our considerations regarding the degree of conceptual novelty and depth of new mechanistic insights expected from papers published in The EMBO Journal.

That said, considering the novel parts of the study with regard to IFN λ signaling, as well as the potential *in vivo* relevance of your findings, we think that the manuscript is an excellent candidate for our partner journal Life Science Alliance (<http://www.life-science-alliance.org/>; our broad scope Open Access journal published in partnership between the EMBO-, Rockefeller University-, and Cold Spring Harbor Laboratory Presses). Should you be interested in transferring your manuscript to Life Science Alliance, its editorial team will be pleased to send it out for peer review. Please follow the transfer link below if you are interested in this option.

For The EMBO Journal, I am sorry we could not be more positive on this occasion, and I hope that this unfavorable decision will not prevent you from considering our journal again for the publication of your results in the future. In the interest of your manuscript and your time, I am providing you with an editorial decision on your manuscript that will allow you to submit it elsewhere without further delay, and I very much hope that you will find the transfer option to Life Science Alliance worthwhile.

Best regards,

Ioannis

** As a service to authors, EMBO Press provides authors with the possibility to transfer a manuscript that one journal cannot offer to publish to another EMBO publication or the open access journal Life Science Alliance launched in partnership between

EMBO Press, Rockefeller University Press and Cold Spring Harbor Laboratory Press. The full manuscript and if applicable, reviewers' reports, are automatically sent to the receiving journal to allow for fast handling and a prompt decision on your manuscript. For more details of this service, and to transfer your manuscript please click on Link Not Available. **

Dear Ioannis,

Thank you for taking the time to evaluate our manuscript. We appreciate your comments and while I usually do not dispute editorial decisions, I would like to address them while highlighting some key aspects of our work that we believe demonstrate its significance and novelty.

Maintenance of a tight barrier in the gut under homeostasis is fundamental to prevent intestinal inflammation in humans and pathology such as IBD arise when this barrier is altered. While extensive research has characterized barrier disruption during inflammation and infection, a critical knowledge gap exists in our understanding of the molecular signals that establish and maintain a healthy barrier in the gut under normal physiological conditions. This gap is particularly evident in our limited understanding of how immune mediators might contribute to barrier maintenance during homeostasis. The present work represents a paradigm shift in our understanding of both interferon biology and epithelial barrier regulation as we describe that type III interferons, specifically IFNL2/3, are key regulators of gut homeostasis by being essential for regulating/promoting barrier function maintenance in epithelial cells.

While we acknowledge the significant prior research in this field that you have highlighted, we would like to clarify how our findings represent a fundamental advance in understanding type III IFN biology at epithelial surfaces, particularly regarding their previously unrecognized role in homeostatic barrier maintenance. We would like to specifically address each point you raised to demonstrate how our findings constitute a significant departure from and expansion of the current paradigm.

1. In particular, type-III IFNs are already known to be expressed at basal levels in epithelial cells. Mitochondrial DNA is a well-known relevant trigger of cGAS-STING signaling in different contexts, and STING signaling itself is already known to drive type-I IFN basal production.

While our lab has previously described basal type III IFNs, our study focused exclusively on their role in antiviral defense and cellular priming before infection. Our current work reveals fundamental advances: We define for the first time the cellular mechanisms leading to basal type III IFN production. Similar to type I interferon, we demonstrate that the mitochondrial DNA-STING pathway regulates basal type III IFNs. Most significantly, we challenge the classical view of basal interferons as static mediators. Instead of being maintained at constant levels, we show that basal type III IFNs are dynamically regulated during epithelial cell confluence through the mechano-sensing HIPPO pathway to orchestrate tight junction formation. Not only, these findings establish type III IFNs as essential regulators of epithelial barrier formation during normal tissue development and homeostasis, they reveal for the first time the active cellular regulation of basal type III interferon expression by epithelial cells to acquire one of their most important cellular function (i.e. barrier function)

2. In addition, they have already been linked to epithelial integrity and barrier function in the context of microbial infection, and an earlier study reported that IFN λ signaling modulates tight junction protein localization, independently of STAT1, in the blood-brain barrier. Furthermore, a recent study uncovered another function of IFN λ in tissue repair of

intestinal mucosa in contexts other than microbial infection - after inflammatory or DNA-related damage, in particular.

While previous studies have shown that type III IFNs enhance barrier function during pathogen infection, the underlying mechanisms remained unknown. Our work provides two crucial advances: we reveal the molecular mechanism by which type III IFNs regulate barrier function, and more importantly, we demonstrate that this regulation occurs during homeostasis, independent of infection. This discovery fundamentally shifts our understanding of type III IFNs from purely infection-response mediators to essential regulators of basic epithelial cell biology, specifically in maintaining barrier integrity under steady-state conditions.

3. Moreover, other interferons have previously been reported to down-regulate claudin-1 and claudin-2 expression, thereby impairing the epithelial barrier function in cultured cells.

Multiple reports have highlighted that treatment of cells with IFN gamma regulates tight junctions by altering claudin expression. However, treatment of cells with IFN gamma does not favor but actually disrupts tight junctions. These findings are contrary to our findings. Our work can show that loss of IFNL2/3 leads to an increase of claudin-2 and loss of barrier integrity. These results show that IFNL2/3 is pro-barrier (which we demonstrate by rescuing barrier function through exogenous treatment of IFNL2/3) while IFN gamma is anti-barrier. Studies of IFN gamma have shown the IFN gamma disrupts the barrier through inducing the internalization of multiple tight junction proteins. As such our data are fundamentally different as both IFN have opposite effects on tight junctions by manipulating claudin expression by distinct mechanism.

4. We also note that the study relies on cell culture experiments, which leaves the question of the in vivo relevance of the findings open,

We have validated our findings through multiple complementary cellular systems. Beyond our initial studies in T84 cells, we confirmed our observations in primary human intestinal organoids, providing validation in a physiologically relevant intestinal epithelial system. Additionally, we further strengthened our findings by demonstrating that these mechanisms are conserved in respiratory epithelial cells, highlighting that the function of basal type III interferon in regulating homeostatic barrier function of epithelial cells is conserved across different barrier tissues. Concerning the lack of in-vivo relevance. Analysis of IFNL2/3 KO mice reveal no apparent defect in barrier function in the intestinal epithelium. However, while mice express IFNL2/3 they lack IFNL1. We show that IFNL2/3 promotes tight junction formation but not IFNL1. Interestingly, we have gathered evidence that IFNL1 might have the opposite effect by destabilizing tight junction formation. As such IFNL2/3 may have acquired different functions in mice and human due to the absence of IFNL1 in mice. Consequently, we believe our findings cannot be adequately modeled in mice. Finally, our collaboration with Boston Children's Hospital has identified early-onset IBD patients carrying IFNL2/3 mutations which interfere with IFNL2/3 mediated signaling through the IFNL receptor. This provides direct clinical evidence for the essential role of these cytokines in maintaining barrier integrity.

5. While there are also some unaddressed mechanistic questions, such as the exact molecular mechanism by which INFLambda2/3 signaling down-regulates claudin-2 expression.

While we acknowledge that additional mechanistic details remain to be explored, our current work provides significant advances in understanding basal type III IFN regulation and function. We have established the fundamental link between the HIPPO pathway and basal type III IFN regulation, and demonstrated basal IFNL2/3's specific control of claudin-2 expression as a direct driver of tight junction formation. These findings represent major conceptual advances in the field. The precise molecular mechanisms downstream of IFNL2/3 signaling that regulate claudin-2 expression are the focus of ongoing studies, but do not diminish the significance of our current discoveries in establishing this previously unknown regulatory axis.

Looking forward to hearing whether you would reconsider your editorial decision

Once again, I would like to thank you for the thorough evaluation of our work

Best regards

steeve

Steeve Boulant, PhD

Associate professor, Department of Molecular Genetics & Microbiology
University of Florida, College of Medicine
1200 Newell drive, Room R1-295
Gainesville, FL 32610
T: +1 352-273-7534
M: +1 352 451 8088
s.boulant@ufl.edu
<https://boulantlab.com/>
Boulant, Steeve

Dear Steeve,

Thank you again for submitting your manuscript EMBOJ-2025-120363R-Q for consideration by The EMBO Journal, and for your patience during peer review. Your manuscript has now been seen by three experts in the field, and we have received the full set of their comments, which you can find below.

I am glad to say that all three referees recognize that this is an interesting and comprehensive study reporting the results from well-performed and appropriately controlled experiments, and that it will be significant for the field. They also identify, however, a number of limitations and provide several reasonable and constructive suggestions for strengthening the manuscript further. In particular, I would like to emphasize the first major comment of referee #1 (also briefly mentioned by referee #2) on the reliance of the study on the T84 cell line and the need for further validation in primary cells as a requirement for any conclusions to be made about normal epithelial cell biology. We agree with the referee that this validation would be an important addition to the study and increase its impact on a broader field. We also find the second major point of referee #2 on the levels of TBP expression particularly relevant for the normalization of the results. In addition, there are several other points and suggestions that could realistically be addressed either experimentally or by appropriately revising the text.

Given the referees' positive comments and recommendations, I would like to invite you to submit a revised version of your manuscript along with a detailed point-by-point response addressing all referees' comments. I should add that it is The EMBO Journal policy to allow only a single round of major revision, and acceptance of your manuscript will therefore depend on the completeness of your responses in this revised version. Please let me know if you have any questions or comments that you would like to discuss with me. If there are any major points you do not agree with or cannot address during your revision, I would encourage you to share them with me as early as possible to discuss how to proceed further in the most efficient way.

We generally allow three months as standard revision time (June 13, 2025). As a matter of policy, competing manuscripts published during this period will not negatively impact our assessment of the conceptual advance presented by your study. However, we request that you contact us as soon as possible upon publication of any related work, to discuss how to proceed. Should you foresee a problem in meeting this three-month deadline, please let us know in advance and we may be able to grant an extension.

Thank you for the opportunity to consider your work for publication in The EMBO Journal. I look forward to your revision.

Best regards,

Ioannis

Instructions for preparing your revised manuscript

1. When you are ready to submit the revision, please upload:

- A Word file of the manuscript text (including legends of main Figures, EV Figures and Tables). Please make sure that changes are highlighted (or "tracked") to be clearly visible.

- Individual production-quality figure files (one file per figure). When assembling your figures, please refer to our figure preparation guidelines in order to ensure proper formatting and readability in print as well as on screen:

If the data shown in a figure are obtained from n {less than or equal to} 2, please use scatter plots showing the individual data points.

- i. the name of the statistical test used to generate error bars and P values
- ii. the number (n) of independent experiments (please specify technical or biological replicates) underlying each data point (discussion of statistical methodology can be reported in the Materials and Methods section, but figure legends should contain a basic description of n , P , and the test applied)

iii. the nature of the bars and error bars (s.d., s.e.m.).

- A point-by-point response to the referees' comments, with a detailed description of the changes made (as a word file). All referees' concerns must be fully addressed and their suggestions taken on board. When preparing your letter of response to the referees' comments, please bear in mind that this will form part of the Review Process File and will therefore be available online to the community. Please note that you have the possibility to opt out of the transparent process at any stage prior to publication by letting the editorial office know (contact@embojournal.org); if you do opt out, the Review Process File link will point to the following statement: "No Review Process File is available with this article, as the authors have chosen not to make the review process public in this case.". For more details on our Transparent Editorial Process, please visit our website: <https://www.embopress.org/page/journal/14602075/authorguide#transparentprocess>

- Expanded View (EV) files (replacing Supplementary Information) that are collapsible/expandable online. A maximum of 5 EV Figures can be typeset. EV Figures should be cited as "Figure EV1, Figure EV2" etc. in the text, and their respective legends should be included in the manuscript file after the legends of regular figures. See detailed instructions regarding Expanded View files here:

- For the figures that you do NOT wish to display as Expanded View figures, they should be bundled together with their legends in a single PDF file called "Appendix", which should start with a short Table of Contents (including page numbers). Appendix figures should be referred to in the main text as: "Appendix Figure S1, Appendix Figure S2" etc. Please see detailed instructions here: <https://www.embopress.org/page/journal/14602075/authorguide#expandedview>

- A complete author checklist, which you can download from our author guidelines (<https://www.embopress.org/page/journal/14602075/authorguide>). Please note that the checklist will also be part of the Review Process File.

2. Please note that no statistics should be calculated and shown in Figures if $n=2$. Please also note that each p value should be reported as an exact value.

3. Before submitting your revision, primary datasets (and computer code, where appropriate) produced in this study need to be deposited in appropriate public databases (see <https://www.embopress.org/page/journal/14602075/authorguide#dataavailability>). In particular, we kindly request you to list the accession numbers, database, and the specific URLs (links) for your deposited RNA sequencing data in a formal "Data availability" section (placed after Methods), following the example below:

"The RNA-seq datasets produced in this study are available in the following database:

Gene Expression Omnibus GSE46843 (<https://www.ncbi.nlm.nih.gov/geo/query/acc.cgi?acc=GSE46843>)"

*** All links should resolve to a page where the data can be accessed. ***

*** Please remember to provide in the Data availability section of your revised manuscript reviewer passwords if the datasets are not yet public. ***

*** The Data Availability Section is restricted to new primary data that are part of this study. In case you have no data that require deposition in a public database, please state so instead of referring to the database: "Our study includes no data deposited in public repositories." under the heading "Data availability". ***

4. The materials and methods need to be described in the manuscript using our structured methods format, which is now required for all research articles. According to this format, the Methods section includes a single "Reagents and Tools Table" - listing key reagents, experimental models, software and relevant equipment including their sources and relevant identifiers - followed by a "Methods and Protocols" section describing the methods. Please download and fill our Reagents and Tools Table template (.docx), which you can find in our author guide:

<https://www.embopress.org/page/journal/14602075/authorguide#structuredmethods>. When submitting your revised manuscript, please do not include the Reagents and Tools Table in the Methods section of the manuscript but instead upload it as a separate file choosing the file type "Reagent Table".

5. Please check that the title and the abstract of the manuscript are brief, yet explicit, even to non-specialists. The length of the title should not exceed 100 characters, and the abstract should be a single paragraph not exceeding 175 words.

6. Please also note our reference format: <https://www.embopress.org/page/journal/14602075/authorguide#referencesformat>.

8. Please remember: digital image enhancement is acceptable practice, as long as it accurately represents the original data and conforms to community standards. If a figure has been subjected to significant electronic manipulation, this must be noted in the figure legend or in the "Materials and Methods" section. The editors reserve the right to request original versions of figures and the original images that were used to assemble the figure.

9. Our journal encourages inclusion of data citations in the reference list to directly cite datasets that were obtained from public databases. Data citations in the article text are distinct from normal bibliographical citations and should directly link to the database records from which the data can be accessed. In the main text, data citations are formatted as follows: "Data ref: Smith et al, 2001" or "Data ref: NCBI Sequence Read Archive PRJNA342805, 2017". In the Reference list, data citations must be labeled with "[DATASET]". A data reference must provide the database name, accession number/identifiers, and a resolvable link to the landing page from which the data can be accessed at the end of the reference. Further instructions are available at: <https://www.embopress.org/page/journal/14602075/authorguide#referencesformat>.

10. We request authors to consider both actual and perceived competing interests. Please review our policy (<https://www.embopress.org/page/journal/14602075/authorguide#conflictsofinterest>) and update your competing interests statement if necessary. Please name this section 'Disclosure and competing interests statement' and place it after the Acknowledgements section.

11. Please note that all corresponding authors are required to provide an ORCID ID upon submission of a revised manuscript (<https://orcid.org/>). Please find instructions on how to link your ORCID ID to your account in our manuscript tracking system in our Author guidelines (<https://www.embopress.org/page/journal/14602075/authorguide#authorshipguidelines>).

12. We use CRediT to specify the contributions of each author in the journal submission system. CRediT replaces the author contribution section, which should be removed from the manuscript. Please use the free text box to provide more detailed descriptions. See also guide to authors: <https://www.embopress.org/page/journal/14602075/authorguide#authorshipguidelines>.

14. We would also welcome the submission of cover suggestions or motifs to be used by our Graphics Illustrator in designing a cover.

15. Please use the link below to submit your revision:
<https://emboj.msubmit.net/cgi-bin/main.plex>

Referee #1:

Interferon lambdas are a family of interferon cytokines with particular potent signaling to epithelial cells. The study by Keser et al. defines mechanisms of basal IFNL production by a colonic epithelial cell line and shows that basal IFNL is regulated by cell density and impacts cell junctions. Using genetic knockouts and pharmacological inhibitors, the authors convincingly show that IFNL production is dependent on cGAS/STING/TBK1 signaling, which is countered by density-dependent hippo pathway signaling. A consequence of the basal IFNL signals is to promote barrier integrity through downregulation of "bad" pore-forming claudin 2. Strengths of the manuscript include well-conceived experiments with appropriate controls and orthogonal genetic and pharmacological approaches to define mechanisms of action. Most of the experiments are quantified with convincing differences that support the model. The major weakness is the use of a metastatic colon carcinoma cell line (T84) for most experiments and using these cells to draw conclusions about normal basal epithelial activities. This doesn't invalidate any of the data but is an important context for implications about normal epithelial cell biology. Specific points follow.

Major points

1. Validation in primary cells is essential for conclusions that these pathways are operational in normal tissues. I appreciate that there are several supportive experiments in normal ileum epithelial cells showing density-dependant expression of IFNL by qPCR (Fig. 1F-G). Also, there is one image shown in sup. Fig. 7A with STING inhibitor H151 but it is not quantified and not clear to my eye that the phenotype shown is the same as the one shown in T84 cells. Additional experiments with the inhibitors that test the mechanistic model would significantly bolster the conclusions. Alternatively, the authors could acknowledge this as a limitation and discuss the role that IFNL production may play in colon cancer.
2. Depletion of mtDNA would be expected to have wider effects beyond cGAS activation as fewer mitochondria would alter the

metabolic state of the cell and reduce the signaling platform for MAVS. This possibility should be acknowledged and/or discussed.

3. Why does IFNL not fully rescue the barrier defect of IFNL ko cells when added back in figure 5? This is not sufficiently explained and raises some question about the direct role of IFNL in this phenotype.
4. Related to the prior point, It is not entirely accurate to claim that CLDN2 knockdown in IFNL ko cells 'restored' the barrier function because TEER remained substantially lower than WT in figure 6H. So, there must be other unknown factors that contribute to the phenotype.
5. Does IFNL treatment of T84 at low density result in loss of CLDN2? Or is this only a phenotype that manifests at high density?

Minor points

1. The introductory statement "IFNL λ 4 is expressed in only a small portion of the human population" should be modified because, depending on ethnicity, the proportion of individuals expressing functional IFNL4 can be substantial.
2. Sup. Fig. 1B-D legend states that cells were plated at high and low density, but it appears that only one condition is shown. Please clarify
3. For the experiment quantifying mitochondrial DNA in Fig. 3G, it would be helpful to clarify whether the cytoplasmic extraction method is removing intact mitochondria.
4. Discussion states that "We recently identified two patients with very early onset IBD that express mutations in their IFNL λ 2 and IFNL λ 3 genes." Does this refer to the studies by Ouahed et al. <https://doi.org/10.1101/2022.03.17.22271929>? Or some other study? The study should be cited

Referee #2:

Keser et. al. examine the influence of basal interferon lambda 2/3 expression on the integrity of tight junctions between intestinal epithelial cells during conditions of homeostasis. Their goal is to further understand how mucosal surfaces maintain barrier integrity that preserves homeostatic interactions with commensal bacteria while impairing the capacity of pathogenic microbes to invade, hypothesizing interferon lambdas play a role. They primarily use a colorectal cell model (T84 cells) that forms an epithelial monolayer that demonstrates barrier function followed by genetic/pharmacologic approaches to dissect pathway activity, with work completed using ileum-derived organoids and a human airway epithelial cell line. The primary findings suggest basal IFNL2/3 expression is mediated by detection of mitochondrial DNA by the cGAS/STING pathway that only occurs at high cellular density due to lack of inhibition by an active Hippo pathway that features cytosolic retention of phosphorylated YAP. They find this IFNL2/3 expression and subsequent pathway activation are required for the capacity of epithelial cells to form tight junctions and maintain barrier function through negative regulation of claudin-2, which impairs these processes. While the work was primarily conducted in an immortalized colorectal cell line, the work is comprehensive, mechanistic, and well-controlled and provides important novel insights into links between mucosal barrier function and innate immunity and is felt to be of high significance.

Major critiques:

1. There is limited data provided on how cell viability was monitored and quantitated, particularly during states of cellular confluence, which is the essential variable influencing the observed phenotypes, and whether there was an impact on viability of the multiple KO lines and chemical inhibitors that were used. It would be helpful to more explicitly include this information to help readers interpret the findings, in addition to what is provided in Supp Fig.6, which hints at a possible difference between lines but which presents proliferation data without commenting on viability. Likewise, comment on presence or absence of genes/pathways associated with cell viability, apoptosis, etc could be further commented on for the RNA-seq data provided in Supp. Fig. 5.
2. TBP expression can be downregulated during epithelial cell differentiation and throughout the manuscript gene expression results are expressed relative to TBP. As much of the data and interpretation relies on this comparison, it would be helpful to include data or comment showing TBP expression itself is not influenced by the experimental conditions (i.e. confluency and the various knockouts and inhibitors) to improve confidence TBP can indeed serve as a reliable "housekeeping" gene to normalize the genes of interest.

Minor comments:

1. The introduction is long and includes information repeated in the discussion, could consider trimming (i.e. most of last 2 paragraphs recap the major findings).
2. Consider an alternative way of expressing that claudin-2 expression is "bad" (page 15), with the assumption that this protein likely serves a physiologic purpose in some context.
3. In Fig. 2C nuclear pIRF3 is evaluated by western blot and then quantitated to conclude there is an increase in high density cells. It would be helpful to provide similar graphical, quantitative data for total IRF3 in total and cytosolic fractions to help assure the reader there are no differences in overall levels using the same technique used to quantitate pIRF3.
4. Comment in the discussion refers to patients with mutations in IFNL2/3. No data or references are provided to provide context for this. Consider add data or references or indicate data not shown (or work in progress, etc.).

5. In the introduction it is noted that IFNL4 is only made by a subset of the population. It would be interesting to speculate in the discussion how this protein may or may not influence the observed phenotypes (at the least, could comment if T84 cell lines are capable of making IFNL4 or not).
6. There is a degree of residual MX1 and IFIT1 induction in high density conditions in spite of STING abrogation and use of cGas inhibitors (Fig.3C-D); it would be interesting to speculate the reason (i.e. incomplete inhibition vs. alternate pathway).
7. Fig. 4C it would be helpful to show (or comment) that total YAP levels are similar at high and low densities.
8. For Fig. 5, it would be helpful to comment on the markedly different nature of the pattern of ZO-1 staining in IFNLR KO and IFNL2/3 lines. In addition, it would be helpful to speculate on why there is partial but not uniform ZO-1 staining noted in the IFNLR KO line and why exogenous IFNL2/3 treatment only partially rescues the phenotype in IFNL2/3 KO cells.
9. Fig 6E-F legend should specify how long after IFNL2/3 treatment RNA and protein were harvested, and if these experiments were performed, could comment on whether repeated or prolonged IFNL treatment led to more significant decreases in claudin2 transcript and protein.
10. Although select results were validated in patient-derived organoids composed of primary cells, which enhances the work significantly, the discussion section should include a paragraph on the limitations of the immortalized cell model that was used for the bulk of the analyses (T84 colorectal cell line) with respect to interpreting/extrapolating the results.

Referee #3:

The manuscript by Keser et al describes a role for basal interferon lambda expression in epithelial tight junction formation and barrier integrity, through the negative regulation of claudin-2, an epithelial protein that normally interferes with barrier integrity. Basal interferon lambda expression is induced by cGAS/STING-mediated mitochondrial DNA detection but inhibited at low cell density by the Hippo mechanotransduction pathway. This is an excellent study and well written manuscript, with clearly presented data and well supported conclusions. I have only minor comments.

1. Page 7-8 and Fig 2B - what was the effect of IFNAR KO on IFN lambda expression? Conversely, what was the effect of IFNLR KO on type I IFN expression? These data would provide a more complete picture of the relevant effects of these KOs.
2. Page 14, 2nd paragraph and Fig 5E - Looks like a significant reduction of FITC-dextran diffusion at day 5 by IFN lambda treatment
3. Sup Fig 7A - I'm confused by this image - the ZO1 network does not seem to outline cells as marked by nuclei - is this just an effect of organoids?
4. Fig 6I - which day post seeding of cells were these images taken?
5. Fig 7 - In the top and bottom right panels, it is not clear what the transmembrane proteins/structures at the cell junctions are - you have CLDN2 right next to them but I don't think they are claudin-2 protein
6. Discussion page 19, 2nd paragraph - is "basal" IFN lambda expression ever relevant if the intestinal tract is seeded with microbiota immediately after birth? How would the presence of microbiota affect your proposed mechanism?
7. Page 12 first line - confirmed instead of controlled?

Dear Ioannis Papaioannou,

Please find below our point-by-point response to the reviewer's comments. We have addressed every point raised by the reviewers and added novel data to answer their questions. We have reorganized the figures as main figures, extended view figures, and appendix figures, as requested.

In summary, we **validated our key findings in primary human ileum-derived organoids** by testing critical pathway inhibitors (TBK1, STING, cGAS) and confirming density-dependent IFN λ 2/3 regulation. We **performed comprehensive cell viability assessments** using cytotoxicity assays to demonstrate that the observed phenotypes were not due to cell death and **enhanced our rescue experiments** with improved claudin-2 knockdown protocols that achieved better functional restoration of barrier function. Additionally, we **addressed technical concerns** by validating housekeeping gene stability, quantifying control proteins, and clarifying methodological details throughout the manuscript.

Our responses to the reviewer comments are organized as follows: reviewer comments are in **black**, and our responses are in **blue**.

We would like to thank the editorial board of EMBO and all three reviewers for this very constructive review. We sincerely feel that addressing the reviewer's comments has significantly increased the impact and relevance of our work.

Looking forward to hearing from you and the reviewers.

All the best,

Steeve Boulant and Megan Stanifer

Referee #1:

Interferon lambdas are a family of interferon cytokines with particular potent signaling to epithelial cells. The study by Keser et al. defines mechanisms of basal IFNL production by a colonic epithelial cell line and shows that basal IFNL is regulated by cell density and impacts cell junctions. Using genetic knockouts and pharmacological inhibitors, the authors convincingly show that IFNL production is dependent on cGAS/STING/TBK1 signaling, which is countered by density-dependent hippo pathway signaling. A consequence of the basal IFNL signals is to promote barrier integrity through downregulation of "bad" pore-forming claudin 2. Strengths of the manuscript include well-conceived experiments with appropriate controls and orthogonal genetic and pharmacological approaches to define mechanisms of action. Most of the experiments are quantified with convincing differences that support the model. The major weakness is the use of a metastatic colon carcinoma cell line (T84) for most experiments and using these cells to draw conclusions about normal basal epithelial activities. This doesn't invalidate any of the data but is an important context for implications about normal epithelial cell biology. Specific points follow.

We want to thank the reviewers for their assessment of our study on interferon lambda signaling in epithelial cells. We appreciate their recognition of our methodological strengths in using both genetic and pharmacological approaches to demonstrate the cGAS/STING/TBK1 dependency of basal IFNL production and its counter regulation by density-dependent Hippo signaling, as well as the functional importance of this pathway in promoting barrier integrity through claudin-2 regulation. We agree that the work needs to be extended to primary cells and have updated the manuscript to include this critical point.

Major points

1. Validation in primary cells is essential for conclusions that these pathways are operational in normal tissues. I appreciate that there are several supportive experiments in normal ileum epithelial cells showing density-dependent expression of IFNL by qPCR (Fig. 1F-G). Also, there is one image shown in sup. Fig. 7A with STING inhibitor H151 but it is not quantified and not clear to my eye that the phenotype shown is the same as the one shown in T84 cells. Additional experiments with the inhibitors that test the mechanistic model would significantly bolster the conclusions. Alternatively, the authors could acknowledge this as a limitation and discuss the role that IFNL production may play in colon cancer.

We thank the reviewer for this thoughtful and constructive comment. We fully agree that validation in primary epithelial models is critical for establishing the physiological relevance of our findings. To address this, we have performed additional experiments using ileum-derived human organoids. Specifically, we tested the effect of key inhibitors—including the TBK1 inhibitor (BX795), STING inhibitors (H151, IN-2), and cGAS inhibitors (G150, RU.521)—in ileum-derived organoids cultured at high and low densities and assessed their impact on IFN λ 2/3 and ISG (Mx1) expression by qRT-PCR and included these in Figure 2F and 3E. Our results show that treatment with these inhibitors significantly reduced basal IFN λ 2/3 and Mx1 expression at high density, while expression levels at low density remained largely unaffected. These findings are consistent with our observations in T84 cells, and further support a density-dependent, cGAS-STING-mediated regulation of IFN λ 2/3 signaling in primary human epithelial tissues. These new data

validate the involvement of the cGAS-STING pathway and downstream IFN λ 2/3 signaling in primary intestinal epithelium, supporting the broader relevance of our model. We have updated the Results and Discussion sections accordingly.

Organoids inherently grow as three-dimensional structures and when seeded as monolayer in 2D they stay associated as clumps. Upon treatment with the STING inhibitor H-151, ZO-1 expression is notably disrupted; however, H-151 may not fully dismantle pre-established tight junctions that are already formed during the initial seeding of organoids. As a result, some organoid regions may still display ZO-1 staining, while others do not. This mosaic pattern reflects variability in inhibitor accessibility and tight junction dynamics. Due to the non-uniform nature of ZO-1 staining in organoids, quantification is technically challenging. To address this, quantified the ZO-1-positive areas relative to DAPI-labeled nuclei in T84 cells, ileum-derived organoids, and Calu-3 cells. Additionally, below please find additional representative ZO-1 staining images of ileum-derived organoids treated with H-151 to better illustrate the observed phenotype.

2. Depletion of mtDNA would be expected to have wider effects beyond cGAS activation as fewer mitochondria would alter the metabolic state of the cell and reduce the signaling platform for MAVS. This possibility should be acknowledged and/or discussed.

We appreciate this insightful comment. We agree that depletion of mtDNA could have broader consequences beyond cGAS activation, particularly by impacting mitochondrial metabolism and the availability of MAVS as a signaling scaffold. In the revised discussion, we now acknowledge these additional effects and note that future work will be necessary to disentangle direct cGAS-driven responses from those secondary to changes in mitochondrial function and MAVS availability (Line 495-498).

3. Why does IFNL not fully rescue the barrier defect of IFNL ko cells when added back in figure 5? This is not sufficiently explained and raises some question about the direct role of IFNL in this phenotype.

We agree that the incomplete rescue of the barrier defect upon IFN λ reconstitution in IFN λ 2/3 KO cells raises additional questions about how IFN λ s help to regulate barrier formation. It is possible that IFN λ signaling must be sustained over time or act in concert with other cell-intrinsic programs, or polarization-dependent pathways to fully support tight junction formation. Supporting this model, our immunofluorescence analysis of tight junction formation in IFN λ 2/3 KO cells treated with exogenous IFN λ 2/3 show that tight junction reform at defined patches within the cells, suggesting that proper mechanotransduction between cells might also be necessary to rescue tight junction formation. Given the link between IFN λ 2/3 and the YAP/TAZ mechanotransduction pathway, there is a possibility that some cells depleted of IFN λ 2/3 have now impaired YAP/TAZ pathway which would in turn impair the rescue of tight junction formation. Alternatively, developmental changes in KO cells may render them less responsive to exogenous IFN λ , even when added back. These possibilities warrant further investigation to clarify the direct versus indirect contributions of IFN λ to epithelial barrier function. We have added additional discussions about this in the text (Line 550-553)

4. Related to the prior point, it is not entirely accurate to claim that CLDN2 knockdown in IFNL ko cells 'restored' the barrier function because TEER remained substantially lower than WT in figure 6H. So, there must be other unknown factors that contribute to the phenotype.

We appreciate this comment and agree that the term "restored" may overstate the extent of the rescue compared to wild-type levels in the original version of our manuscript. In the previous version, cells were transduced with lentivirus expressing an shRNA against claudin-2 (or non-coding control) as they were seeded onto transwells, and polarization was followed directly after transduction. While we observed an increased TEER using this method, this was not a complete rescue. To address this concern and more accurately assess the contribution of CLDN2, we repeated this experiment but first transduced the cells and allowed them to express the shRNA prior to polarization to achieve a more prolonged and stable knockdown of CLDN2. With this enhanced knockdown approach, we observed a more pronounced improvement in TEER, with values exceeding 1000 Ω /cm², indicating a stronger functional rescue (updated Figure 6H). In addition, we also see a restoration of the ZO-1 belt using this new method (Figure 6I). These results further support the role of CLDN2 in mediating the barrier defect. The figures and corresponding text have been updated in this new version of our manuscript to reflect these new data.

5. Does IFNL treatment of T84 at low density result in loss of CLDN2? Or is this only a phenotype that manifests at high density?

For WT cells we see small reduction in claudin-2 levels at high density, as there is only a small amount of claudin-2 expressed in these cells. At low density when there is greater claudin 2 expression we see a larger decrease with addition of IFN λ 2/3 (see below). In IFN λ 2/3 KO cells we see a large decrease in claudin-2 expression at both low and high density following exogenous treatment (see below) For the sake of narrative clarity, we chose to present the high-density condition only in Figure 6F.

Minor points

1. The introductory statement "IFNλ4 is expressed in only a small portion of the human population" should be modified because, depending on ethnicity, the proportion of individuals expressing functional IFNλ4 can be substantial.

We have updated the text to more accurately reflect the population-level variation in IFNλ4 expression, which is indeed influenced by genetic background. Specifically, we now state that IFNλ4 expression varies across human populations and can be substantial depending on ethnicity, rather than describing it as present in only a small portion of the population (Line 50-53).

2. Sup. Fig. 1B-D legend states that cells were plated at high and low density, but it appears that only one condition is shown. Please clarify

Thank you for catching this — in Supplementary Figure 1B–D (now EV1 B-D), cells were seeded at low cellular density, then imaged over days. We have corrected the figure legend to reflect this accurately (Line 1002).

3. For the experiment quantifying mitochondrial DNA in Fig. 3G, it would be helpful to clarify whether the cytoplasmic extraction method is removing intact mitochondria.

Our "Cellular fractioning to assess total and cytoplasmic mitochondrial DNA" protocol includes a brief permeabilization step using 0.1% NP-40 in PBS, followed by a low-speed centrifugation (6500 rpm for 1 min at 4°C) to pellet cellular organelles, including intact mitochondria. This method is commonly used to isolate cytosolic fractions while minimizing mitochondrial contamination. All steps were performed on ice to preserve subcellular integrity and limit artifactual mitochondrial rupture. Therefore, the mitochondrial DNA detected in the supernatant is likely to represent mtDNA that was released into the cytosol, rather than contamination from intact mitochondria. A sentence clarifying this has been added to our methods (Line 802-803).

4. Discussion states that "We recently identified two patients with very early onset IBD that express mutations in their IFNλ2 and IFNλ3 genes." Does this refer to the studies by Ouahed et al. <https://doi.org/10.1101/2022.03.17.22271929>? Or some other study? The study should be cited

Yes, the statement refers to the study by Ouahed et al. (<https://doi.org/10.1101/2022.03.17.22271929>), which we have now properly cited in the revised manuscript. Since this original preprint we have worked with the authors and added additional function details. This work is current under review.

Referee #2:

Keser et. al. examine the influence of basal interferon lambda 2/3 expression on the integrity of tight junctions between intestinal epithelial cells during conditions of homeostasis. Their goal is to further understand how mucosal surfaces maintain barrier integrity that preserves homeostatic interactions with commensal bacteria while impairing the capacity of pathogenic microbes to invade, hypothesizing interferon lambdas play a role. They primarily use a colorectal cell model (T84 cells) that forms an epithelial monolayer that demonstrates barrier function followed by genetic/pharmacologic approaches to dissect pathway activity, with work completed using ileum-derived organoids and a human airway epithelial cell line. The primary findings suggest basal IFNL2/3 expression is mediated by detection of mitochondrial DNA by the cGAS/STING pathway that only occurs at high cellular density due to lack of inhibition by an active Hippo pathway that features cytosolic retention of phosphorylated YAP. They find this IFNL2/3 expression and subsequent pathway activation are required for the capacity of epithelial cells to form tight junctions and maintain barrier function through negative regulation of claudin-2, which impairs these processes. While the work was primarily conducted in an immortalized colorectal cell line, the work is comprehensive, mechanistic, and well-controlled and provides important novel insights into links between mucosal barrier function and innate immunity and is felt to be of high significance.

We want to thank the reviewers for their time and enthusiasm for our work examining how basal interferon lambda 2/3 expression maintains intestinal epithelial tight junction integrity. The reviewer accurately captured our investigation linking the cGAS/STING pathway to IFNL2/3 expression and barrier function through claudin-2 regulation, and we appreciate their recognition of the significance of these findings in connecting mucosal barrier function with innate immunity.

Major critiques:

1. There is limited data provided on how cell viability was monitored and quantitated, particularly during states of cellular confluence, which is the essential variable influencing the observed phenotypes, and whether there was an impact on viability of the multiple KO lines and chemical inhibitors that were used. It would be helpful to more explicitly include this information to help readers interpret the findings, in addition to what is provided in Supp Fig.6, which hints at a possible difference between lines but which presents proliferation data without commenting on viability. Likewise, comment on presence or absence of genes/pathways associated with cell viability, apoptosis, etc could be further commented on for the RNA-seq data provided in Supp. Fig. 5.

We appreciate the reviewer's suggestion and agree that this is a critical point missing in the previous version of the manuscript. To address this point, we performed the CytoTox 96® Non-Radioactive Cytotoxicity Assay, which measures lactate dehydrogenase (LDH) release as an indicator of membrane integrity and cell viability. We assessed cytotoxicity across WT and all knockout cell lines under both high and low cell density conditions. Our results demonstrate that there is no significant difference in intrinsic cytotoxicity between the cell lines, strongly suggesting that the observed phenotypes are not due to altered cell viability. We have added this data to Appendix Figure 5B. Interestingly, we observed a modest increase in cytotoxicity at high density compared to low density. However, to rule out the possibility that

this difference drives the phenotype, we treated WT cells with the MST1/2 inhibitor XMU-MP-1, which mimics Hippo pathway OFF signaling (a low-density-like state, as shown in Figure 4D–E). As indicated below, LDH levels in mock- and XMU-MP-1–treated cells were comparable, indicating that phenotype differences between densities are not due to cytotoxicity.

In addition, we conducted LDH assays for all pharmacological treatments used in the study and confirmed that none of the drugs induced cytotoxicity at the concentrations used in our study – as depicted below. These data have been added to the manuscript as Appendix Figure 3.

Finally, we revisited the RNA-seq dataset and analyzed expression of genes related to apoptosis and cell viability pathways. These analyses revealed no significant changes in the expression of viability- or apoptosis-associated genes across densities or between cell lines. Moreover, gene set enrichment analysis (GSEA) of our RNA-seq data revealed that apoptosis and cytotoxicity pathways are not differentially regulated between high and low cell density conditions, nor across different cell lines. These findings further support the conclusion that the observed phenotypes are not linked to differences in cell viability. These viability data have now been added to the Appendix Figure 5

2. TBP expression can be downregulated during epithelial cell differentiation and throughout the manuscript gene expression results are expressed relative to TBP. As much of the data and interpretation relies on this comparison, it would be helpful to include data or comment showing TBP expression itself is not influenced by the experimental conditions (i.e. confluency and the various knockouts and inhibitors) to improve confidence TBP can indeed serve as a reliable "housekeeping" gene to normalize the genes of interest.

To assess the reliability of TBP as a housekeeping gene under our experimental conditions, we analyzed RNA-seq data across all cell lines and densities. TBP expression remained stable, as did HPRT1, another commonly used housekeeping gene, indicating that TBP is a suitable reference for normalization. Furthermore, the expression of ISGs such as IFIT1, ISG15, Mx1, and OAS1 was consistently higher in WT cells at high density. These data have been added to the Appendix Figure 4.

Mx1 expression at high density showed a consistent 7–10 fold increase relative to low density, regardless of which housekeeping gene was used for normalization. We have quantified the relative expression of Mx1 in high vs. low cell density using multiple housekeeping genes—including TBP, HPRT1, HMBS, SDHA, GUSB, RPLP0, PPIA, TFRC, RPL13A, GAPDH, PGK1, and YWHAZ. These data are presented below and show that expression of Mx1 is always higher in high cell density independently of the housekeeping genes used for the normalization. These findings further validate the robustness of our gene expression data. We have chosen to not integrate the data presented below in the manuscript as we believe the expression of TBP and HPRT1 in our different cell lines (data above and now Appendix Figure 4) are sufficient.

To validate the use of TBP as a reliable housekeeping gene across our experimental conditions, we also checked TBP Cq values in mock-treated cells and cells treated with all compounds used in the study, including BX795, H151, IN2, G150, RU.521, EtBr, ddC, and XMU-MP-1. As shown below, Cq values remained stable and did not show significant variation across treatments. These results confirm that TBP expression is unaffected by the compounds used and thus serves as a suitable internal control for qRT-PCR normalization throughout our study.

Minor comments:

1. The introduction is long and includes information repeated in the discussion, could consider trimming (i.e. most of last 2 paragraphs recap the major findings).

We thank the reviewer for their comments and have shorten the intro by removing the last two summary paragraphs as suggested.

2. Consider an alternative way of expressing that claudin-2 expression is "bad" (page 15), with the assumption that this protein likely serves a physiologic purpose in some context.

We have revised the text (line 421-427) to better reflect this dual nature by emphasizing that while claudin-2 serves important physiological functions in fluid and ion transport under homeostatic conditions, its aberrant or sustained upregulation has been linked to disrupted barrier integrity and pathological leakiness, particularly in the context of inflammatory bowel diseases.

3. In Fig. 2C nuclear pIRF3 is evaluated by western blot and then quantitated to conclude there is an increase in high density cells. It would be helpful to provide similar graphical, quantitative data for total IRF3 in total and cytosolic fractions to help assure the reader there are no differences in overall levels using the same technique used to quantitate pIRF3.

Thanks for this suggestion. We have now included the quantification of total IRF3 levels in both total and cytosolic fractions as we previously showed for pIRF3. The results show that total and cytosolic IRF3 levels remain comparable between high and low cell density, supporting the interpretation that the observed increase in nuclear pIRF3 at high density is due to enhanced activation rather than differences in IRF3 total protein abundance. This quantification has been added to Figure 2C right panel.

4. Comment in the discussion refers to patients with mutations in IFNL2/3. No data or references are provided to provide context for this. Consider add data or references or indicate data not shown (or work in progress, etc.).

The statement in the discussion refers to the study by Ouahed et al. (<https://doi.org/10.1101/2022.03.17.22271929>), which we have now properly cited in the revised manuscript. Since this original preprint, we have worked with the authors to evaluate the antiviral functions of these interferons. Notably, the updated version of their work is currently under revision at the Journal of Experimental Medicine, highlighting its ongoing relevance and the shared interest in understanding the functional impact of IFNL2/3 mutations in early-onset IBD.

5. In the introduction it is noted that IFNL4 is only made by a subset of the population. It would be interesting to speculate in the discussion how this protein may or may not influence the observed phenotypes (at the least, could comment if T84 cell lines are capable of making IFNL4 or not).

Although many epithelial cell lines are presumed to lack functional IFNL4 due to the high prevalence of the TT/TT genotype at the rs368234815 locus, T84 cells have not, to our knowledge, been genotyped at this position. Therefore, we cannot definitively conclude whether they express functional IFNL4 or not. However, given that a substantial portion of the human population—especially individuals of African and

some Asian descent—retain the ability to express functional IFN λ 4, it would be interesting in future studies to explore how IFN λ 4 signaling might modulate epithelial barrier function and basal antiviral responses in these contexts. We have extended our discussion to include these points (Line 548-550).

6. There is a degree of residual MX1 and IFIT1 induction in high density conditions in spite of STING abrogation and use of cGas inhibitors (Fig.3C-D); it would be interesting to speculate the reason (i.e. incomplete inhibition vs. alternate pathway).

As STING expression is an interferon-stimulated gene (ISG), it is upregulated as T84 cells become confluent, which could lead to a situation where excessive STING levels make it difficult to fully inhibit the pathway. In this context, incomplete inhibition of the pathway could result in residual MX1 and IFIT1 induction despite the use of inhibitors.

7. Fig. 4C it would be helpful to show (or comment) that total YAP levels are similar at high and low densities.

We appreciate the reviewer's suggestion. We have included the total YAP levels and show that they remain comparable between high and low cell densities, while, phosphorylated YAP (pYAP) levels are low at low density and increase at high density, consistent with density-dependent regulation of YAP activity. The new Figure 4C now represents this.

8. For Fig. 5, it would be helpful to comment on the markedly different nature of the pattern of ZO-1 staining in IFNLR KO and IFNL2/3 lines. In addition, it would be helpful to speculate on why there is partial but not uniform ZO-1 staining noted in the IFNLR KO line and why exogenous IFNL2/3 treatment only partially rescues the phenotype in IFNL2/3 KO cells.

In IFNLR KO cells, we observe a partial but non-uniform ZO-1 localization at the membrane, suggesting that while some tight junctions can form, they may be unstable or improperly organized in the absence of IFNL signaling. In IFNLRKO cells none of the IFNLs can signal while in the IFNL2/3 KO cells IFNL1 is still present. While both IFNL1 and IFNL2/3 signal on the same receptor, we cannot exclude the possibility that IFNL1 and IFNL2/3 impact tight junction formation differently. While we believe that this could be a very interesting finding, it is outside the scope of this paper. We have added a line to the discussion that these differences are important, and more work is required to follow up on them (Line 544-548).

In the IFN λ 2/3 KO cells, exogenous IFN λ 2/3 treatment partially rescues ZO-1 localization, but does not fully restore a WT-like phenotype. This incomplete rescue may reflect the need for sustained or temporally coordinated IFN λ signaling during key stages of epithelial polarization that transient treatment cannot fully replicate. It is also possible that IFN λ signaling must act in concert with other cytokines, intrinsic differentiation programs, or density-dependent cues to fully support tight junction formation.

9. Fig 6E-F legend should specify how long after IFNL2/3 treatment RNA and protein were harvested, and if these experiments were performed, could comment on whether repeated or prolonged IFNL treatment led to more significant decreases in claudin2 transcript and protein.

T84 WT and IFN λ 2/3 KO cells were seeded and treated with recombinant IFN λ 2/3 for 3 days prior to RNA and protein harvest. These details have been added to the figure legend to help clarify this (lines 974). This treatment duration was selected based on prior optimization experiments, as it produced the most robust decrease in CLDN2 transcript and protein levels. Longer or repeated treatments did not further enhance this phenotype in our hands.

10. Although select results were validated in patient-derived organoids composed of primary cells, which enhances the work significantly, the discussion section should include a paragraph on the limitations of the immortalized cell model that was used for the bulk of the analyses (T84 colorectal cell line) with respect to interpreting/extrapolating the results.

We appreciate the reviewer's suggestion. This was also a concern shared by another reviewer and we have updated our revised version to include more data using primary organoids. Specifically, we tested the effect of key inhibitors—including the TBK1 inhibitor (BX795), STING inhibitors (H151, IN-2), and cGAS inhibitors (G150, RU.521)—in ileum-derived organoids cultured at high and low densities and assessed their impact on IFN λ 2/3 and ISG (Mx1) expression by qRT-PCR and included these in Figure 2F and 3E. Our results show that treatment with these inhibitors significantly reduced basal IFN λ 2/3 and Mx1 expression at high density, while expression levels at low density remained largely unaffected.

Referee #3:

The manuscript by Keser et al describes a role for basal interferon lambda expression in epithelial tight junction formation and barrier integrity, through the negative regulation of claudin-2, an epithelial protein that normally interferes with barrier integrity. Basal interferon lambda expression is induced by cGAS/STING-mediated mitochondrial DNA detection but inhibited at low cell density by the Hippo mechanotransduction pathway. This is an excellent study and well written manuscript, with clearly presented data and well supported conclusions. I have only minor comments.

We appreciate the reviewers enthusiasm for our manuscript and their suggestions.

1. Page 7-8 and Fig 2B - what was the effect of IFNAR KO on IFN lambda expression? Conversely, what was the effect of IFNLR KO on type I IFN expression? These data would provide a more complete picture of the relevant effects of these KOs.

We thank the reviewer for this important suggestion. To address this point, we analyzed the expression of IFNL2/3 and IFNB in T84 WT, IFNLR KO, IFNAR KO, and IFNAR/IFNLR double KO (dKO) cells at both high and low densities. We found that IFNL2/3 expression was consistently low at low density and strongly upregulated at high density across all cell lines, indicating that this density-dependent induction of IFN λ is independent of type I and type III IFN receptor signaling. Conversely, IFNB expression levels remained similar and comparable across cell lines, with no significant difference observed between high and low densities. These results suggest that cell density has a specific effect on IFN λ expression, while type I IFN expression remains largely unaffected. These data can now be found in EV Figure 2A and B.

2. Page 14, 2nd paragraph and Fig 5E - Looks like a significant reduction of FITC-dextran diffusion at day 5 by IFN lambda treatment

We thank the reviewer for this observation. Although we do observe a decrease in permeability in the IFN λ 2/3 KO cells treated with recombinant IFN λ 2/3 at day 5, the establishment of TEER and the impairment of FITC-dextran diffusion were not fully rescued upon exogenous treatment (Figure 5D–E). This suggests that endogenous IFN λ 2/3 signaling during earlier stages of polarization may be critical for optimal barrier formation. We have now revised the text accordingly.

3. Sup Fig 7A - I'm confused by this image - the ZO1 network does not seem to outline cells as marked by nuclei - is this just an effect of organoids?

While epithelial cells are often illustrated as growing strictly perpendicular (at a 90-degree angle) to the substrate, they frequently exhibit a tilt relative to the imaging plane. This tilt causes a spatial shift between apical junctional markers like ZO-1 and basal structures such as the nucleus. As a result, in top-down confocal images, ZO-1 staining may appear offset from DAPI-labeled nuclei. The accompanying schematic illustrates how this natural cell inclination can create the impression that ZO-1 does not accurately outline the cell borders when, in fact, it is correctly localized at the apical junctions.

Here are some examples of ZO-1 staining in other cells from different sources:

Nuclei and ZO-1 staining do not perfectly overlap in MDCK cells with the ZO-1 Monoclonal Antibody (Thermo Fisher Cat #33-9100).

Nuclei and ZO-1 staining do not perfectly overlap in MCF7 cells with the ZO-1 Polyclonal Antibody (Cat #CL594-21773)

Nuclei and ZO-1 staining do not perfectly overlap in CaCo-2 cells (Lehner et al, 2020)

4. Fig 6I - which day post seeding of cells were these images taken?

We thank the reviewer for pointing this out. The images in Figure 6I were taken 8 days post-transduction, corresponding to the endpoint of the transwell experiment. We have now clarified this information in the figure legend.

5. Fig 7 - In the top and bottom right panels, it is not clear what the transmembrane proteins/structures at the cell junctions are - you have CLDN2 right next to them but I don't think they are claudin-2 protein

We have redesigned the final schematic to remove the depiction of CLDN2 at the cell junctions, as we agree that its inclusion could be misleading. While our study focuses on the regulation of CLDN2 expression under different conditions, we do not present data specifically addressing change in its subcellular localization. To avoid overinterpretation, we now discuss CLDN2 regulation only in the text and have refrained from visually representing it in the revised Figure 7.

6. Discussion page 19, 2nd paragraph - is "basal" IFN lambda expression ever relevant if the intestinal tract is seeded with microbiota immediately after birth? How would the presence of microbiota affect your proposed mechanism?

While it is true that the intestinal tract is rapidly colonized by microbiota after birth, several studies have shown that epithelial cells can produce low levels of "basal" IFN even in the absence of overt infection, and that this expression can be further modulated by microbial signals. In our model, basal IFN λ refers to the low-level, homeostatic expression in the absence of strong pathogenic stimulation. The presence of microbiota likely contributes to maintaining or modulating this basal IFN λ expression through constant low-level stimulation of innate immune receptors. Therefore, rather than being mutually exclusive, microbiota presence may reinforce the proposed mechanism by sustaining basal IFN λ signaling, which in turn supports tight junction maturation and epithelial barrier function during homeostasis.

7. Page 12 first line - confirmed instead of controlled?

We thank the reviewer for catching this. We have replaced "controlled" with "confirmed" in the text on page 12, as suggested (Line 370).

Dear Steve,

Thank you for submitting your revised manuscript (EMBOJ-2025-120363R1) to The EMBO Journal for our consideration, and for your patience during peer review. Your manuscript has been sent back to the original referees who had previously reviewed the initial version of your manuscript, and we have now received their comments, which are included below.

I am very pleased to say that all referees are satisfied with the thorough revision, find the revised study compelling and significant for the field, recognize that their initially raised criticisms and concerns have been adequately and sufficiently addressed, and now support the publication of the manuscript in our journal with requests for only minor corrections (ref. #2 and #3), which we kindly ask you to address in a final version of your manuscript. Please include in your resubmission a brief point-by-point response explaining how these requests are addressed and detailing any changes to the manuscript.

From the editorial side, there are also a few changes and corrections we need you to make in the final version of your manuscript, before we can proceed with its formal acceptance and publication in The EMBO Journal:

- The funding information mentioned in the Acknowledgements section of your manuscript should be identical to that provided in our manuscript tracking system (eJP); currently, the following information is missing from eJP: UF College of Medicine start-up package funds. Please note that the funding information included in the Comments box could not be extracted by our production team, and therefore all funders should be added to the "More Funders" list.

- Please provide a list of up to 5 keywords (preferably broad terms that would enhance online search engine discoverability of your article) after the Abstract of your revised manuscript.

- Our journal's reference format is alphabetical; please revise the format of your references according to the instructions provided in our guide to authors: <https://www.embopress.org/page/journal/14602075/authorguide#referencesformat>.

- Please include in the Data availability statement of your revised manuscript the specific and permanent URL for your deposited RNA sequencing data.

- The "Conflict of Interest" heading should be changed to "Disclosure and competing interests statement".

- Regarding Figure and Table callouts, please note that:

1. There are callouts for Table 2, but no such table has been uploaded.

2. Callouts for Expanded View (EV) Figures should be renamed to "Figure EV1-EV4" instead of EV Figure 1E, EV Figure 2A etc.

3. Appendix Figures should be called out as "Appendix Figure S#" (e.g. Appendix Figure S2A)

- Please correct the name of the journal in your Author Checklist and re-upload the corrected version of the checklist.

- Thank you for uploading your Source Data. Please note that the Source Data checklist our staff has previously shared with you must also be completed and uploaded as a Related Manuscript File. Source data files need to be saved in a single folder per Figure and then uploaded as a .zip folder. For example, all Source data files for Figure 1 need to be saved in a single folder, which then needs to be zipped and uploaded as "SD Figure 1.zip" folder. For EV and/or Appendix figures, please ZIP together all Source Data.

- Please note that EMBO press papers are accompanied online by:

- A) a short (2 sentences) summary of the findings and their significance,

- B) 2-5 short bullet points highlighting the key results, and

- C) a synopsis image in .jpg or .png format that is exactly 550 pixels wide and 300-600 pixels high (the height is variable). Please note that all text needs to be legible at the final size.

Please upload this information along with your revised manuscript (the text for A and B should be provided in a separate Word file).

- During our routine pre-publication Figure checks, our team noticed that blots shown in the following Figures are over-contrasted:

Figure 6 D, F and G Actin

Figure EV1F Actin

Please check the blots and remake the figures.

- During our routine data checks, our data editors have raised the following queries regarding figures, data, and legends. Please make sure that all requests below are completely addressed in the final version of your manuscript (please highlight all changes in the manuscript):

1. Please provide the exact p values in the legends of Figures 1A, B, D, E, F, G; 2A, B, C, D, E, F; 3A, C, D, E, F, G; 4E, G, H;

5B, C, D, E, G, H; 6B, E, H; EV1 A, C, E, G; EV2B, E, G; EV4 C, D.

2. Please indicate the statistical test used for data analysis in the legends of Figures EV3 E, G.

3. Please note that the measure of center for the error bars needs to be defined in the legends of Figures 1A, B, D, E, F, G; 2A, B, C, D, E, F; 3A, C, D, E, F, G, H; 4E, G, H; 5B, C, D, E, G, H; 6B, E, H; EV1 A, C, E, G; EV2 B, C, D, E, G; EV4 C, D.

- The order of the manuscript sections must be corrected as follows: Title page - Abstract and Keywords - Introduction - Results - Discussion - Methods - Data Availability - Acknowledgements - Disclosure and Competing Interests Statement - References - Figure Legends - main Tables (if there are any) - Expanded View Figure Legends.

Please also note that as part of the EMBO publications' Transparent Editorial Process, The EMBO Journal publishes online a Peer Review File along with each accepted manuscript. This File will be published in conjunction with your paper and will include the referee reports, your point-by-point response and all pertinent correspondence relating to the manuscript. You can opt out of this by letting the editorial office know (contact@embojournal.org). If you do opt out, the Peer Review File link will point to the following statement: "No Peer Review File is available with this article, as the authors have chosen not to make the review process public in this case."

We look forward to seeing a final version of your manuscript as soon as possible. Please let us know if you have any questions and use this link to submit your revision: <https://emboj.msubmit.net/cgi-bin/main.plex>.

Best regards,

Ioannis

Referee #1:

The authors have thoroughly addressed all of our comments and concerns. This is a compelling and rigorous study, and will significantly impact the field.

Referee #2:

The extensive revisions and inclusion of additional new organoid data as well as requested control data to help interpret viability, gene expression, and protein expression results in this revised manuscript have adequately addressed the comments of all three reviewers. A final minor comment for consideration is that while the last 2 paragraphs of the introduction were removed to help with length, could consider re-introduction of 1-2 sentences at the end of the introduction outlining the premise and approach for the current work.

Referee #3:

This is an important study and the authors have addressed the previous reviewers' comments with thoughtful responses and new data that support their conclusions. Although it was difficult to match the authors' responses with modifications in the manuscript (because either the change was not marked/tracked or was not on the stated lines), I believe that the changes made have significantly improved the manuscript.

One minor point is that I do not see the data on IFNL and IFNB gene expression in response to reviewer 3 point 1 - the authors claim they are now in EV Fig 2A and B but that does not appear to be the case.

Dear Ioannis, EMBO editorial board, and reviewers,

We are happy that our revised manuscript (EMBOJ-2025-120363R1) was well received by the reviewers. We have gone through the suggested editorial changes and have updated our manuscript to include these requests. Please see below our editorial updates as well as our answers to the open reviewer comments. Editorial or reviewers comments are in **black** and our response is in **blue**. We hope that our changes have improved the manuscript and that it is now suitable for publication.

Best regards,

Steeve Bouland and Megan Stanifer

Editorial comments

- The funding information mentioned in the Acknowledgements section of your manuscript should be identical to that provided in our manuscript tracking system (eJP); currently, the following information is missing from eJP: UF College of Medicine start-up package funds. Please note that the funding information included in the Comments box could not be extracted by our production team, and therefore all funders should be added to the "More Funders" list.

We have now updated the funders list and ensured that it was identical between the Acknowledgements section and the eJP.

- Please provide a list of up to 5 keywords (preferably broad terms that would enhance online search engine discoverability of your article) after the Abstract of your revised manuscript.

We have included the following keywords in the main text, directly after the Abstract: Epithelial Barrier Function; Basal IFN λ Signaling; Tight Junctions; Hippo Signaling Pathway; cGAS/STING Pathway.

- Our journal's reference format is alphabetical; please revise the format of your references according to the instructions provided in our guide to authors: <https://www.embopress.org/page/journal/14602075/authorguide#referencesformat>.

We have revised the reference list according to the journal's alphabetical format, as outlined in the Guide to Authors.

- Please include in the Data availability statement of your revised manuscript the specific and permanent URL for your deposited RNA sequencing data.

We have now included the accession number GSE279172 at the following permanent URL: <https://www.ncbi.nlm.nih.gov/geo/query/acc.cgi?acc=GSE279172> in the Data Availability section.

- The "Conflict of Interest" heading should be changed to "Disclosure and competing interests statement".

We have now updated the heading.

- Regarding Figure and Table callouts, please note that:

1. There are callouts for Table 2, but no such table has been uploaded.

Sorry for this. The table is Appendix Table S2 and the text has been updated as such.

2. Callouts for Expanded View (EV) Figures should be renamed to "Figure EV1-EV4" instead of EV Figure 1E, EV Figure 2A etc.

All Expanded View Figures have been renamed.

3. Appendix Figures should be called out as "Appendix Figure S#" (e.g. Appendix Figure S2A)

All Appendix Figures are called as such.

- Please correct the name of the journal in your Author Checklist and re-upload the corrected version of the checklist.

We have updated the type to correctly state the name of the journal.

- Thank you for uploading your Source Data. Please note that the Source Data checklist our staff has previously shared with you must also be completed and uploaded as a Related Manuscript File. Source data files need to be saved in a single folder per Figure and then uploaded as a .zip folder. For example, all Source data files for Figure 1 need to be saved in a single folder, which then needs to be zipped and uploaded as "SD Figure 1.zip" folder. For EV and/or Appendix figures, please ZIP together all Source Data.

We have now completed the Source Data checklist and uploaded it as a Related Manuscript File. Additionally, all Source Data files have been organized into individual folders per figure, zipped accordingly (e.g., "SD Figure 1.zip"), and uploaded as

instructed. For the EV and Appendix figures, all Source Data files have been zipped together and uploaded as a single archive.

- Please note that EMBO press papers are accompanied online by:

A) a short (2 sentences) summary of the findings and their significance,

B) 2-5 short bullet points highlighting the key results, and

C) a synopsis image in .jpg or .png format that is exactly 550 pixels wide and 300-600 pixels high (the height is variable). Please note that all text needs to be legible at the final size.

Please upload this information along with your revised manuscript (the text for A and B should be provided in a separate Word file).

We have prepared the required materials accordingly. The short summary (A) and bullet-point highlights (B) have been compiled into a Word file, which has been uploaded with the new revision. In addition synopsis image has also be uploaded.

- During our routine pre-publication Figure checks, our team noticed that blots shown in the following Figures are over-contrasted:

Figure 6 D, F and G Actin

Figure EV1F Actin

Please check the blots and remake the figures.

We have now adjusted the images and have remade the figures accordingly. The updated versions have been uploaded with the revised manuscript.

- During our routine data checks, our data editors have raised the following queries regarding figures, data, and legends. Please make sure that all requests below are completely addressed in the final version of your manuscript (please highlight all changes in the manuscript):

1. Please provide the exact p values in the legends of Figures 1A, B, D, E, F, G; 2A, B, C, D, E, F; 3A, C, D, E, F, G; 4E, G, H; 5B, C, D, E, G, H; 6B, E, H; EV1 A, C, E, G; EV2B, E, G; EV4 C, D.

2. Please indicate the statistical test used for data analysis in the legends of Figures EV3 E, G.

3. Please note that the measure of center for the error bars needs to be defined in the legends of Figures 1A, B, D, E, F, G; 2A, B, C, D, E, F; 3A, C, D, E, F, G, H; 4E, G, H; 5B, C, D, E, G, H; 6B, E, H; EV1 A, C, E, G; EV2 B, C, D, E, G; EV4 C, D.

We have addressed all the requested revisions in the final version of the manuscript. Exact p-values, statistical tests, and measures of center for error bars have been added to the relevant figures and figure legends as indicated. All changes are highlighted in the revised manuscript.

- The order of the manuscript sections must be corrected as follows: Title page - Abstract and Keywords - Introduction - Results - Discussion - Methods - Data Availability - Acknowledgements - Disclosure and Competing Interests Statement - References - Figure Legends - main Tables (if there are any) - Expanded View Figure Legends.

We have revised the manuscript to follow the required section order.

Answers to reviewers:

Referee #1:

The authors have thoroughly addressed all of our comments and concerns. This is a compelling and rigorous study, and will significantly impact the field.

We sincerely thank the reviewer for their thoughtful and constructive comments. Your valuable insights greatly helped us improve the quality and clarity of our work. We are pleased to hear that you find the study compelling and rigorous, and we appreciate your recognition of its potential impact on the field. Your feedback has been instrumental in strengthening the manuscript.

Referee #2:

The extensive revisions and inclusion of additional new organoid data as well as requested control data to help interpret viability, gene expression, and protein expression results in this revised manuscript have adequately addressed the comments of all three reviewers. A final minor comment for consideration is that while the last 2 paragraphs of the introduction were removed to help with length, could consider re-introduction of 1-2 sentences at the end of the introduction outlining the premise and approach for the current work.

We greatly appreciate the reviewer's careful evaluation and thoughtful suggestion. The extensive revisions and inclusion of new organoid and control data were guided by the valuable feedback from all reviewers, which significantly improved our manuscript. In response to your suggestion, we have now added a brief summary at the end of the Introduction section to clearly outline the premise and approach of our study. Thank you again for your insightful and constructive comments.

Referee #3:

This is an important study and the authors have addressed the previous reviewers' comments with thoughtful responses and new data that support their conclusions. Although it was difficult to match the authors' responses with modifications in the manuscript (because either the change was not marked/tracked or was not on the

stated lines), I believe that the changes made have significantly improved the manuscript.

One minor point is that I do not see the data on IFNL and IFNB gene expression in response to reviewer 3 point 1 - the authors claim they are now in EV Fig 2A and B but that does not appear to be the case.

We thank the reviewer for their careful reading and constructive feedback. Regarding the data on IFNL and IFNB gene expression, EV Figure 2A and B present qRT-PCR analysis of IFNL2/3 and IFNB transcripts (gene expression) across multiple cell lines (WT, IFNAR KO, IFNLR KO, IFNR KO, and IFNL2/3 KO) at two different cell densities as suggested in reviewer 3 point 1. Specifically, the amount of IFNB was similar in WT and IFNLR KO cells, while IFNL expression was comparable between WT and IFNAR KO cells. These results support the conclusions drawn in the manuscript. We apologize if this was not clearly indicated in the figure legend or text.

Dear Steeve,

Congratulations on an excellent manuscript! I am very pleased to inform you that it has been accepted for publication in The EMBO Journal. Thank you for comprehensively addressing the initially raised referees' concerns and all editorial requests for changes and corrections.

If you have any questions, please do not hesitate to contact the Editorial Office. Thank you for your contribution to The EMBO Journal. Working with you has been a pleasure!

Best regards,

Ioannis
